# Differentially Private Stochastic Gradient Descent with Fixed-Size Minibatches: Tighter RDP Guarantees with or without Replacement

**Jeremiah Birrell**
Texas State University
jbirrell@txstate.edu

**Reza Ebrahimi**
University of South Florida
ebrahimim@usf.edu

**Rouzbeh Behnia**
University of South Florida
behnia@usf.edu

**Jason Pacheco**
University of Arizona
pachecoj@cs.arizona.edu

## Abstract

Differentially private stochastic gradient descent (DP-SGD) has been instrumental in privately training deep learning models by providing a framework to control and track the privacy loss incurred during training. At the core of this computation lies a subsampling method that uses a privacy amplification lemma to enhance the privacy guarantees provided by the additive noise. Fixed size subsampling is appealing for its constant memory usage, unlike the variable sized minibatches in Poisson subsampling. It is also of interest in addressing class imbalance and federated learning. Current computable guarantees for fixed-size subsampling are not tight and do not consider both add/remove and replace-one adjacency relationships. We present a new and holistic Rényi differential privacy (RDP) accountant for DP-SGD with fixed-size subsampling without replacement (FSwoR) and with replacement (FSwR). For FSwoR we consider both add/remove and replace-one adjacency, where we improve on the best current computable bound by a factor of $4$. We also show for the first time that the widely-used Poisson subsampling and FSwoR with replace-one adjacency have the same privacy to leading order in the sampling probability. Our work suggests that FSwoR is often preferable to Poisson subsampling due to constant memory usage. Our FSwR accountant includes explicit non-asymptotic upper and lower bounds and, to the authors' knowledge, is the first such RDP analysis of fixed-size subsampling with replacement for DP-SGD. We analytically and empirically compare fixed size and Poisson subsampling, and show that DP-SGD gradients in a fixed-size subsampling regime exhibit lower variance in practice in addition to memory usage benefits.

## 1 Introduction

Differentially private stochastic gradient descent (DP-SGD) (Abadi et al., 2016) (DP-SGD) has been one of the cornerstones of privacy preserving deep learning. DP-SGD allows a so-called moments accountant technique to sequentially track privacy leakage (Abadi et al., 2016). This technique is subsumed by Rényi differential privacy (RDP) (Mironov, 2017), a relaxation of standard differential privacy (DP) (Dwork and Rothblum, 2016) that is widely used in private deep learning and implemented in modern DP libraries such as Opacus (Meta Platforms, 2024) and autodp (Zhu and Wang, 2019; Zhu et al., 2022). RDP facilitates rigorous analysis when the dataset is accessed by a sequence of randomized mechanisms as in DP-SGD. While other privacy accountant frameworks such

38th Conference on Neural Information Processing Systems (NeurIPS 2024).

as $f$-DP (Dong et al., 2022), privacy loss distributions (PLD) (Koskela et al., 2020), Privacy Random Variable (PRV) (Gopi et al., 2021), Analytical Fourier Accountant (AFA) (Zhu et al., 2022), and Saddle-Point Accountant (SPA) (Alghamdi et al., 2023) have been proposed to improve conversion to privacy profiles, RDP is still one of the main privacy accountants used in popular deep learning tools and applications. Thus, we expect providing tighter bounds specific to RDP will have important practical implications for the community. Each iteration of DP-SGD can be viewed as a private release of information about a stochastic gradient. DP accounting methods bound the total privacy loss incurred by applying a sequence of DP-SGD mechanisms in training. Privacy of the overall mechanism is ensured through the application of two random processes: 1) a randomized mechanism that subsamples minibatches of the training data, 2) noise applied to each gradient calculation (e.g., Gaussian). Informally, if a mechanism $\mathcal{M}$ is $(\epsilon, \delta)$-DP then a mechanism that subsamples with probability $q \in (0, 1)$ ensures privacy $(O(q\epsilon), q\delta)$-DP; a result known as the "privacy amplification lemma" (Li et al., 2012; Zhu and Wang, 2019). Computing the privacy parameters of a mechanism requires an accounting procedure that can be nontrivial to design. Abadi et al. (2016) compute DP parameters for the special case of Gaussian noise, which was later extended to track RDP parameters by Mironov (2017). These analyses are limited to Poisson subsampling mechanisms which produce minibatches of variable size. This creates engineering challenges in modern machine learning pipelines and also does not allow for sampling with replacement. Fixed-size minibatches are preferable as they can be aligned with available GPU memory, leading to higher learning throughput. In practice, fixed size minibathces are of interest in practical applications such as private federated learning. To address the issue, Wang et al. (2019) and Zhu et al. (2022) propose the latest fixed-size accounting methods. Wang et al. (2019) provide computable RDP bounds for FSwoR that cover general mechanisms under replace-one adjacency, while Zhu et al. (2022) give computable bounds for FSwoR under add/remove adjacency. Specifically, the study in (Wang et al., 2019) provides a general formula to convert the RDP parameters of a mechanism $\mathcal{M}$ to RDP parameters of a subsampled (without replacement) mechanism, but the bounds they provide are only tight up to a constant factor at leading order (i.e., comparing dominant terms in the asymptotic expansion in $q$). Moreover, while the FSwoR results under add/remove adjacency relationship in (Zhu et al., 2022) are in parallel to our results, their accountant is not readily applicable to FSwoR under replace-one adjacency relationship.

In our work, we (1) present a new and holistic RDP accountant for DP-SGD with fixed-size subsampling without replacement (FSwoR) and with replacement (FSwR) that considers both add/remove and replace-one adjacency. We note that FSwoR is equivalent to shuffling the dataset at the start of every iteration and then taking the first $|B|$ elements, which is conveniently similar to the default behavior of common deep learning libraries (e.g. Pytorch's `DataLoader` iterator object) while FSwR latter is implemented, e.g., in Pytorch's `RandomSampler`, and is widely used to address class imbalance in classifiers (Mease et al., 2007) and biased client selection in federated learning applications (Cho et al., 2022). (2) Our FSwoR improves on the best current computable bound under replace-one adjacency (Wang et al., 2019) by a factor of $4$. (3) Our FSwR accountant includes non-asymptotic upper and lower bounds and, to the authors' knowledge, is the first such analysis of fixed-size RDP with replacement for DP-SGD. (4) For the first time, we show that FSwoR and the widely-used Poisson subsampling have the same privacy under replace-one adjacency to leading order in the sampling probability. This has important practical benefits given the memory management advantages of fixed-size subsampling. Thus our results suggest that FSwoR is often preferable to Poisson subsampling. The implementation of our accountant is included in Supplementary Materials and is also publicly available to the community at `https://github.com/star-ailab/FSRDP`.

## 2 Background and Related Work

Mironov (2017) proposed a differential privacy definition based on Rényi divergence, RDP, that subsumes the celebrated moments accountant (Abadi et al., 2016) used to track privacy loss in DP-SGD. RDP and many other flavors of DP rely on subsampling for *privacy amplification* to ensure that privacy guarantees hold when DP-SGD is applied on samples (i.e., minibatches) of training data (Bun et al., 2018; Balle et al., 2018). Later works such as (Mironov et al., 2019; Zhu and Wang, 2019) propose a generic refined bound for subsampled Gaussian Mechanism that improves the tightness of guarantees in RDP. However, these methods also produce variable-sized minibatches. Bun et al. (2018), Balle et al. (2018), and Wang et al. (2019) propose fixed-size subsampled mechanisms. Balle et al. (2018) provide a general fixed-size subsampling analysis with and without replacement but their bounds focus on tight $(\epsilon, \delta)$-DP bounds for a single step, and are not readily applicable to DP-SGD.

The AFA developed by Zhu et al. (2022) is an enhancement of the moments accountant offering a lossless conversion to $(\epsilon, \delta)$ guarantees and is designed for general mechanisms. However, in the context of DP-SGD, there are still technical and practical difficulties. For instance, in DP-SGD there is generally a large number of training steps, in which case the numerical integrations required by Algorithm 1 in Zhu et al. (2022) involve the sum of a large number of oscillatory terms. This can lead to numerical difficulties, especially in light of the lack of rigorous error bounds for the double (Gaussian) quadrature numerical integration approach as discussed in their Appendix E.1. In addition, the bounds in Zhu et al. (2022) are not readily applicable under replace-one adjacency. Due to these considerations, and the fact that RDP is still widely used, in this work, we focus on improving the RDP bounds from Wang et al. (2019).Hayes et al. (2024) uses the results in (Zhu et al., 2022) to provide a bound specific to DP-SGD when one is only concerned with training data reconstruction attacks rather than the membership inference attacks, which leads to a relaxation of DP to provide a bound that is only applicable to defending against data reconstruction and not membership inference. Our work can be viewed as a complimentary work to (Hayes et al., 2024) that is specific to DP-SGD and provides conservative bounds protecting against membership inference attacks (and thus any other type of model inversion attacks including training data reconstruction) in the RDP context.

In summary, to our knowledge, Wang et al. (2019) provide the best computable bounds in the fixed-size regime for RDP that are practical for application to DP-SGD. While the computations in (Wang et al., 2019) have the attractive property of applying to a general mechanism, in this work, we show that there is room for obtaining tighter bounds specific to DP-SGD with Gaussian noise. Our FS-RDP bounds address this theoretical gap by employing a novel Taylor expansion expansion approach, which precisely captures the leading order behavior in the sampling probability, while employing a computable upper bounds on the integral remainder term to prevent privacy leakage. For convenience, we provide the definition of RDP below.

**Definition 2.1** ($(\alpha, \epsilon)$-RDP (Mironov et al., 2019)). Let $\mathcal{M}$ be a randomized mechanism, i.e., $\mathcal{M}(D)$ is a probability distribution for any choice of allowed input (dataset) $D$. A randomized mechanism is said to have $(\alpha, \epsilon)$-RDP if for any two adjacent datasets $D, D'$ it holds that $D_\alpha(\mathcal{M}(D)\|\mathcal{M}(D')) \leq \epsilon$, where the Rényi divergence of order $\alpha > 1$ is defined by

$$D_\alpha(Q\|P) := \frac{1}{\alpha - 1} \log \left[ \int \left( \frac{Q(x)}{P(x)} \right)^\alpha P(x) dx \right] . \tag{1}$$

The definition of dataset adjacency varies among applications. In this work we consider two such relations: 1) The add/remove adjacency relation, where datasets $D$ and $D'$ to be adjacent if one can be obtained from the other by adding or removing a single element; we denote this by $D \simeq_{a/r} D'$. 2) The replace-one adjacency definition, denoted $D \simeq_{r\text{-}o} D'$, wherein $D$ is obtained from $D'$ by replacing a single element. In the next section we derive new RDP bounds when $\mathcal{M}$ is DP-SGD using fixed-size minibatches, with or without replacement.

## 3 Rényi-DP Bounds for SGD with Fixed-size Subsampling

In this section we present our main theoretical results, leading up to Rényi-DP bounds for SGD for fixed-size subsampling done without replacement (Thm. 3.3 for add/remove adjacency and Thm. 3.4 for replace-one adjacency) and with replacement (Thm. 3.7). In Sec. 3.4, we also compare our results with the analysis in (Wang et al., 2019) and show that our analysis yields tighter Rényi bounds by a factor of approximately 4.

### 3.1 FS-RDP: Definition and Initial Bounds

Given a loss function $\mathcal{L}$, a training dataset $D$ with $|D|$ elements, and a fixed minibatch size, we consider the DP-SGD NN parameter updates with fixed-size minibatches,

$$\Theta_{t+1}^D = \Theta_t^D - \eta_t G_t , \qquad G_t = \frac{1}{|B|} \left( \sum_{i \in B_t^D} \text{Clip}(\nabla_\theta \mathcal{L}(\Theta_t, D_i)) + Z_t \right) , \tag{2}$$

where $t$ is the iteration number, the initial condition $\Theta_0^D$ is independent of $D$, $\eta_t$ are the learning rates, the noises $Z_t$ are Gaussians with mean 0 and covariance $C^2 \sigma_t^2 I$, and the $B_t^D$ are random minibatches

with fixed size; we denote the fixed value of $|B_t^D|$ by $|B|$ for brevity. We will consider fixed-size subsampling both without and with replacement; in the former, $B_t^D$ is a uniformly selected subset of $\{0, ..., |D| - 1\}$ of fixed size $|B|$ for every $t$ and in the latter, $B_t^D$ is a uniformly selected element of $\{0, ..., |D| - 1\}^{|B|}$ for every $t$. We assume the Gaussian noises and minibatches are all independent jointly for all steps. Here Clip denotes the clipping operation on vectors, with $\ell^2$-norm bound $C > 0$, i.e., $\text{Clip}(x) \coloneqq x/\max\{1, \|x\|_2/C\}$. We derive a RDP bound (Mironov, 2017) for the mechanism consisting of $T$ steps of DP-SGD (2), denoted:

$$\mathcal{M}_{[0,T]}^{FS}(D) \coloneqq (\Theta_0^D, ..., \Theta_T^D). \tag{3}$$

Specifically, we consider the cases without replacement, $\mathcal{M}_{[0,T]}^{FS_{woR}}(D)$, and with replacement, $\mathcal{M}_{[0,T]}^{FS_{wR}}(D)$. To obtain these FS-RDP accountants we bound $D_\alpha(\mathcal{M}_{[0,T]}^{FS}(D) \| \mathcal{M}_{[0,T]}^{FS}(D'))$, where $D'$ is any dataset that is adjacent to $D$; as previously stated, we will consider both add/remove and replace-one adjacency. By taking worst-case bounds over the input state at each step, as done in Theorem 2.1 in (Abadi et al., 2016), we decompose the problem into a sum over steps

$$D_\alpha(\mathcal{M}_{[0,T]}^{FS}(D) \| \mathcal{M}_{[0,T]}^{FS}(D')) \leq \sum_{t=0}^{T-1} \sup_{\theta_t} D_\alpha(p_t(\theta_{t+1}|\theta_t, D) \| p_t(\theta_{t+1}|\theta_t, D')), \tag{4}$$

where the time-inhomogeneous transition probabilities are,

$$p_t(\theta_{t+1}|\theta_t, D) = \frac{1}{Z_{|B|,|D|}} \sum_b N_{\mu_t(\theta_t, b, D), \widetilde{\sigma}_t^2}(\theta_{t+1}), \tag{5}$$

$$\mu_t(\theta_t, b, D) \coloneqq \theta_t - \eta_t \frac{1}{|B|} \sum_{i \in b} \text{Clip}(\nabla_\theta \mathcal{L}(\theta_t, D_i)).$$

Here $N_{\mu, \widetilde{\sigma}^2}(\theta)$ is the Gaussian density with mean $\mu$ and covariance $\widetilde{\sigma}^2 I$, $\widetilde{\sigma}_t^2 \coloneqq C^2 \eta_t^2 \sigma_t^2 / |B|^2$, and the summation is over the set of allowed index minibatches, $b$, with normalization constant $Z_{|B|,|D|} = \binom{|D|}{|B|}$ for FS$_{woR}$-RDP and $Z_{|B|,|D|} = |D|^{|B|}$ for FS$_{wR}$-RDP. When additional clarity is needed we use the notation $p_t^{FS_{woR}}$ and $p_t^{FS_{wR}}$ to distinguish the transition probabilities for these respective cases.

## 3.2 FS$_{woR}$-RDP Upper Bounds

The computations thus far mimic those in (Abadi et al., 2016; Mironov et al., 2019), with the choice of subsampling method and adjacency relation playing no essential role. That changes in this section, where we specialize to the case of subsampling without replacement. First, in Section 3.2.1 we consider the add/remove adjacency relation; the proof in this case contains many of the essential ideas of our method but is simpler from a computational perspective. The more difficult case of replace-one adjacency will then be studied in Section 3.2.2

### 3.2.1 Add/remove Adjacency

In Appendix A we derive the following Rényi divergence bound for FS$_{woR}$-subsampled DP-SGD under add/remove adjacency.

**Theorem 3.1.** *Let $D \simeq_{a/r} D'$ be adjacent datasets. With transition probabilities defined as in (5) and letting $q = |B|/|D|$ we have*

$$\sup_{\theta_t} D_\alpha(p_t^{FS_{woR}}(\theta_{t+1}|\theta_t, D) \| p_t^{FS_{woR}}(\theta_{t+1}|\theta_t, D')) \leq D_\alpha(q N_{1, \sigma_t^2/4} + (1-q) N_{0, \sigma_t^2/4} \| N_{0, \sigma_t^2/4}). \tag{6}$$

The key step in the proof consists of the decomposition of the mechanism given in Lemma A.1. We note that the r.h.s. of (6) differs by a factor of $1/4$ in the variances from the corresponding result for Poisson subsampling in Mironov et al. (2019). This is due to the inherent sensitivity difference between fixed-size and Poisson subsampling under add/remove adjacency; see (28) - (29). To bound the r.h.s. of (3.1) we will employ a Taylor expansion computation with explicit remainder bound. As the r.h.s. of (6) has the same mathematical form as the Poisson subsampling result of Mironov et al. (2019), and therefore can be bounded by the same methods employed there, the Taylor expansion

method is not strictly necessary in the add/remove case. However, we find it useful to illustrate the Taylor expansion method in this simpler case before proceeding to the significantly more complicated replace-one case in Section 3.2.2, where the method in Mironov et al. (2019) does not apply.

The Rényi divergences on the r.h.s. of (6) can be written

$$D_\alpha(qN_{1,\sigma_t^2/4} + (1-q)N_{0,\sigma_t^2/4} \| N_{0,\sigma_t^2/4}) = \frac{1}{\alpha - 1} \log[H_{\alpha,\sigma_t}(q)] \,, \tag{7}$$

where for any $\alpha > 1$, $\sigma > 0$ we define

$$H_{\alpha,\sigma}(q) := \int \left( \frac{qN_{1,\sigma^2/4}(\theta) + (1-q)N_{0,\sigma^2/4}(\theta)}{N_{0,\sigma^2/4}(\theta)} \right)^\alpha N_{0,\sigma^2/4}(\theta)d\theta \,. \tag{8}$$

Eq. (8) does not generally have a closed form expression. We upper bound it via Taylor expansion with integral formula for the remainder at order $m$ (c.f. Thm. 1.2, (Stewart, 2022)):

$$H_{\alpha,\sigma}(q) = \sum_{k=0}^{m-1} \frac{q^k}{k!} \frac{d^k}{dq^k} H_{\alpha,\sigma}(0) + R_{\alpha,\sigma,m}(q) \,, \tag{9}$$

where the remainder term is given by

$$R_{\alpha,\sigma,m}(q) = q^m \int_0^1 \frac{(1-s)^{m-1}}{(m-1)!} \frac{d^m}{dq^m} H_{\alpha,\sigma}(sq)ds \,. \tag{10}$$

Note that we do not take $m \to \infty$ and so $H_{\alpha,\sigma}$ is not required to be analytic in order to make use of (9). Also note that (9) is an equality, therefore if we can compute/upper-bound each of the terms then we will arrive at a computable non-asymptotic upper bound on $H_{\alpha,\sigma}(q)$, without needing to employ non-rigorous stopping criteria for an infinite series, thus avoiding privacy leakage. The order $m$ is a parameter that can be freely chosen by the user. To implement (9) one must compute the derivatives $\frac{d^k}{dq^k} H_{\alpha,\sigma}(0)$ and bound the remainder (10). We show how to do both of those steps for general $m$ in Appendix B; in the following theorem we summarize the results for $m = 3$, which we find provides sufficient accuracy in our experiments.

**Theorem 3.2** (Taylor Expansion Upper Bound). *For $q < 1$ we have*

$$H_{\alpha,\sigma}(q) = 1 + \frac{q^2}{2} \alpha(\alpha - 1) M_{\sigma,2} + R_{\alpha,\sigma,3}(q) \,, \tag{11}$$

*where the remainder has the bound*

$$R_{\alpha,\sigma,3}(q) \leq q^3 \alpha(\alpha-1)|\alpha - 2| \begin{cases} \sum_{\ell=0}^{\lceil\alpha\rceil-3} q^\ell \frac{(\lceil\alpha\rceil-3)!}{(\lceil\alpha\rceil-3-\ell)!(3+\ell)!} \widetilde{B}_{\sigma,\ell+3} + \frac{1}{6}\widetilde{B}_{\sigma,3} & \text{if } \alpha - 3 > 0 \\ \frac{1}{6}(1-q)^{\alpha-3}\widetilde{B}_{\sigma,3} & \text{if } \alpha - 3 \leq 0 \end{cases}$$

$$=: \widetilde{R}_{\alpha,\sigma,3}(q) \,, \tag{12}$$

*and the $M$ and $\widetilde{B}$ parameters are given by (41) and (44) respectively.*

The reason for the complexity of the formulas (11) - (12) is the need to obtain a rigorous upper bound, and not simply an asymptotic result. Results for other choices of $m$ are given in Appendix B.

Combining Theorem 3.2 with equations (4) - (8) we now arrive at a computable $T$-step FS$_{\text{woR}}$-RDP guarantee.

**Theorem 3.3** ($T$-step FS$_{\text{woR}}$-RDP Upper Bound: Add/Remove Adjacency). *Assuming $q < 1$, the mechanism $\mathcal{M}_{[0,T]}^{FS_{woR}}(D)$, defined in (3), has $(\alpha, \epsilon_{[0,T]}^{FS_{woR}}(\alpha))$-RDP under add/remove adjacency, where*

$$\epsilon_{[0,T]}^{FS_{woR}}(\alpha) \leq \sum_{t=0}^{T-1} \frac{1}{\alpha - 1} \log \left[ 1 + \frac{q^2}{2} \alpha(\alpha-1)M_{\sigma,2} + \widetilde{R}_{\alpha,\sigma,3}(q) \right] \,, \tag{13}$$

*where $M$ and $\widetilde{R}$ are given by (41) and (12) respectively.*

More generally, using the calculations in Appendix B one obtains RDP bounds for any choice of the number of terms, $m$. The result (13) corresponds to $m = 3$, which we find to be sufficiently accurate in practice; see Figure 6.

### 3.2.2 Replace-one Adjacency

Theorem 3.1 applies to add/remove adjacency, but our method of proof can also be used in the replace-one adjacency case, where it yields the following $FS_{woR}$-RDP upper bounds.

**Theorem 3.4** ($FS_{woR}$-RDP Upper Bounds for Replace-one Adjacency). *Let $D \simeq_{r\text{-}o} D'$ be adjacent datasets. Assuming $q := |B|/|D| < 1$, for any integer $m \geq 3$ we have*

$$\sup_{\theta_t} D_\alpha(p_t^{FS_{woR}}(\theta_{t+1}|\theta_t, D) \| p_t^{FS_{woR}}(\theta_{t+1}|\theta_t, D')) \tag{14}$$

$$\leq \frac{1}{\alpha - 1} \log \left[ 1 + q^2 \alpha(\alpha - 1) \left( e^{4/\sigma_t^2} - e^{2/\sigma_t^2} \right) + \sum_{k=3}^{m-1} \frac{q^k}{k!} \widetilde{F}_{\alpha, \sigma_t, k} + \widetilde{E}_{\alpha, \sigma_t, m}(q) \right],$$

*where $\widetilde{F}_{\alpha, \sigma_t, k}$ is given by (80) and $\widetilde{E}_{\alpha, \sigma_t, m}(q)$ by (87).*

The derivation, found in Appendix C, follows many of the same steps as the add/remove-adjacency case from Theorem 3.1, though deriving bounds on the terms in the Taylor expansion is significantly more involved and the resulting formulas are more complicated. The decomposition from Lemma A.1 again constitutes a key step in the proof. For comparison purposes, in Appendix C.1 we derive the analogous result for Poisson-subsampling under replace-one adjacency; see Theorem C.9.

A corresponding $T$-step RDP bound, analogous to Theorem 3.3, for replace-one adjacency can similarly be obtained by combining Theorem 3.4 with Eq. (4).

**Theorem 3.5** ($T$-step $FS_{woR}$-RDP Upper Bound: Replace-one Adjacency). *Assuming $q < 1$ and for any integer $m \geq 3$, the mechanism $\mathcal{M}_{[0,T]}^{FS_{woR}}(D)$, defined in (3), has $(\alpha, \epsilon_{[0,T]}^{FS_{woR}}(\alpha))$-RDP under replace-one adjacency, where*

$$\epsilon_{[0,T]}^{FS_{woR}}(\alpha) \leq \sum_{t=0}^{T-1} \frac{1}{\alpha - 1} \log \left[ 1 + q^2 \alpha(\alpha - 1) \left( e^{4/\sigma_t^2} - e^{2/\sigma_t^2} \right) + \sum_{k=3}^{m-1} \frac{q^k}{k!} \widetilde{F}_{\alpha, \sigma_t, k} + \widetilde{E}_{\alpha, \sigma_t, m}(q) \right]. \tag{15}$$

*Here $\widetilde{F}_{\alpha, \sigma_t, k}$ is given by (80) and $\widetilde{E}_{\alpha, \sigma_t, m}(q)$ by (87).*

We emphasize that one strength of our approach is that it provides a unified method for deriving computable bounds in both the add/remove and replace-one adjacency cases. We also note that Theorem 11(b) of Zhu et al. (2022), Theorem 6 of Balle et al. (2018), and Theorem 5 in Mironov et al. (2019) combine to provide an alternative method for deriving the add/remove result of Theorem 3.1 but not the replace-one result of Theorem 3.4. Unlike in the corresponding add/remove theorems, using $m > 3$ in Theorems 3.4 and 3.5 is often required to obtain acceptable accuracy, though we generally find that $m = 4$ is sufficient; see Figure 2.

### 3.3 $FS_{wR}$-RDP Upper and Lower Bounds

We similarly obtain RDP bounds for DP-SGD using fixed-size minibatches with replacement, i.e., (2) with minibatches $B_t^D$ that are iid uniformly random samples from $\{0, ..., |D| - 1\}^{|B|}$. First we present upper bounds on the RDP of $\mathcal{M}_{[0,T]}^{FS_{wR}}(D)$. The derivation follows a similar pattern to that of our results for $FS_{woR}$-subsampling, first using a probabilistic decomposition of the mechanism (see Lemma D.1) and then expanding (see Appendix D.1).

**Theorem 3.6** (Fixed-size RDP with Replacement Upper Bound). *Assuming $|D| > |B|$ and under add/remove adjacency, the mechanism $\mathcal{M}_{[0,T]}^{FS_{wR}}(D)$ has $(\alpha, \epsilon_{[0,T]}^{FS_{wR}}(\alpha))$-RDP with ($H_{\alpha,\sigma}$ given by (8))*

$$\epsilon_{[0,T]}^{FS_{wR}}(\alpha) \leq \sum_{t=0}^{T-1} \frac{1}{\alpha - 1} \log \left[ \sum_{n=1}^{|B|} \widetilde{a}_n H_{\alpha, \sigma_t/n}(\widetilde{q}) \right], \tag{16}$$

$$\widetilde{q} := 1 - (1 - |D|^{-1})^{|B|}, \quad \widetilde{a}_n := \widetilde{q}^{-1} \binom{|B|}{n} |D|^{-n} (1 - |D|^{-1})^{|B|-n}.$$

The functions $H_{\alpha,\sigma}$ were previously bounded above in Theorem 3.2 (for order $m = 3$), and more generally in Appendix B (for general $m$). It is straightforward to show that $\widetilde{q} = O(q)$ and $\widetilde{a}_n =$

$O(q^{n-1})$. Therefore, recalling the asymptotics (11) of $H_{\alpha,\sigma}$, the behavior of $\epsilon_{[0,T]}^{FS_{wR}}(\alpha)$ and $\epsilon_{[0,T]}^{FS_{woR}}(\alpha)$ is the same at leading order in $q$. We also present the following novel lower bounds on the Rényi divergence for one step of $FS_{wR}$ DP-SGD; see Appendix D.2 for the derivation.

**Theorem 3.7** (Fixed-size RDP with Replacement Lower Bound). *Let the mechanism $\mathcal{M}_t^{FS_{wR}}(D, \theta_t)$ denote the $t$'th step of $FS_{wR}$ DP-SGD (2) when using the training dataset $D$ and starting from the NN parameters $\theta_t$, i.e., the transition probabilities (5) where the average is over minibatches $b$ that are elements of $\{0, ..., |D| - 1\}^{|B|}$. In this result we consider the choice of $|B|$ to be fixed but will allow $D$ and $\theta_t$ to vary. Suppose there exists $\theta_t$, $d$, and $d'$ such that*

$$Clip(\nabla_\theta \mathcal{L}(\theta_t, d)) = -Clip(\nabla_\theta \mathcal{L}(\theta_t, d')) \text{ and } \|Clip(\nabla_\theta \mathcal{L}(\theta_t, d))\| = \|Clip(\nabla_\theta \mathcal{L}(\theta_t, d'))\| = C,$$

*i.e., when using the NN parameters $\theta_t$, the clipped gradients at the samples $d$ and $d'$ are anti-parallel and saturate the clipping threshold. Then for any integer $\alpha \geq 2$ and any choice of dataset size $N$ we have the following worst-case Rényi divergence lower bound*

$$\sup_{\substack{(\theta_t, D, D'): |D| = N, \\ D' \simeq_{a/r} D}} D_\alpha\left(\mathcal{M}_t^{FS_{wR}}(D, \theta_t) \| \mathcal{M}_t^{FS_{wR}}(D', \theta_t)\right) \geq \frac{1}{\alpha - 1} \log\left[\sum_{n_1, ..., n_\alpha = 0}^{|B|} a_{n_1} ... a_{n_\alpha} e^{\frac{4}{\sigma_t^2} \sum_{i < j} n_i n_j}\right],$$

$$(17)$$

*where $a_n := \binom{|B|}{n} N^{-n} (1 - N^{-1})^{|B| - n}$.*

The lower bound (17) provides a complement to one step of the upper bound from Theorem 3.6. We demonstrate the behavior of the lower bound (17) in Figure 1, where we employed the computational method discussed in Appendix D.2.1. Specifically, we show the $FS_{wR}$-RDP lower bound as a function of $\alpha$ for fixed $q$ and several choices of $|B|$. For small $\alpha$, the lower bounds for different $|B|$'s are approximately equal. This holds up until a critical $|B|$-dependent threshold where there is a "phase transition" to a regime with non-trivial $|B|$ dependence (even for fixed $q$) wherein the RDP bound increases quickly. This is in contrast to $FS_{woR}$-RDP which depends on $|B|$ only

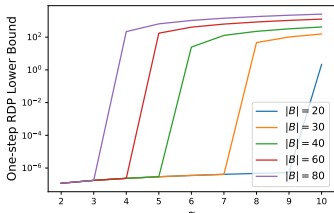

Figure 1: $FS_{wR}$-RDP lower bounds from Theorem 3.7 as a function of $\alpha$, with $\sigma = 6$ and $q = 0.001$.

through the ratio $q$. The critical $\alpha$ is smaller for larger $|B|$, which implies worse privacy guarantees for larger $|B|$, with all else being equal. Intuitively, a larger minibatch size provides a greater chance for the element that distinguished $D$ from $D'$ to be selected multiple times in a single minibatch. If it is selected too many times then it can overwhelm the noise and become "noticeable", thus substantially degrading privacy; this is the key property that distinguishes $FS_{wR}$ from $FS_{woR}$. In practice, this behavior implies that $FS_{wR}$-RDP requires much smaller minibatch sizes than $FS_{woR}$-RDP in order to avoid this critical threshold and maintain privacy.

### 3.4 Comparison with (Wang et al., 2019)

We compare our fixed-size RDP upper bounds with the upper and lower bounds obtained by applying the general-purpose method from Wang et al. (2019) to DP-SGD with Gaussian noise. First we compare the results asymptotically to leading order in $q$ and $1/\sigma_t^2$, which is the domain relevant to applications, and then we will compare them numerically using the full non-asymptotic bounds. Unless stated otherwise, from here on the phrase leading order implicitly refers to the parameters $q$ and $1/\sigma_t^2$.

The method from Wang et al. (2019) applied to DP-SGD (App. E.1) gives the one-step Rényi bounds

$$\epsilon'_{Wang,t}(\alpha) \leq 2q^2 \alpha (e^{4/\sigma_t^2} - 1) + O(q^3) = 8q^2 \alpha / \sigma_t^2 + O(q^2/\sigma_t^4) + O(q^3). \quad (18)$$

In contrast, to second order in $q$, one step of our FS-RDP results under add/remove adjacency from Theorem 3.3 or Theorem 3.6 gives

$$\epsilon'_t(\alpha) \leq \frac{q^2}{2} \alpha (e^{4/\sigma_t^2} - 1) + O(q^3) = 2q^2 \alpha / \sigma_t^2 + O(q^2/\sigma_t^4) + O(q^3),$$

while our FS$_{\text{woR}}$-RDP bound under replace-one adjacency from Theorem 3.4 gives

$$\epsilon'_t(\alpha) \leq q^2\alpha\left(e^{4/\sigma_t^2} - e^{2/\sigma_t^2}\right) + O(q^3) \qquad (19)$$
$$= 2q^2\alpha/\sigma_t^2 + O(q^2/\sigma_t^4) + O(q^3).$$

Therefore, to leading order, our methods all have the same behavior and give tighter RDP bounds by a factor of $4$ than that of Wang et al. (2019). We emphasize that the adjacency relation used by Wang et al. (2019) is replace-one and therefore, strictly speaking, should be compared only with our Theorem 3.4, which also uses replace-one adjacency. Figure 2 shows a non-asymptotic comparison between these RDP bounds, with ours being tighter significantly tighter than that of Wang et al. (2019). In particular, our bound from Theorem 3.4 with $m = 4$ remains close to the theoretical lower bound from Wang et al. (2019) over the entire range of $\alpha$'s pictured, which is a sufficiently large range for practical application to DP, e.g., larger than the default range used in Meta Platforms (2024).

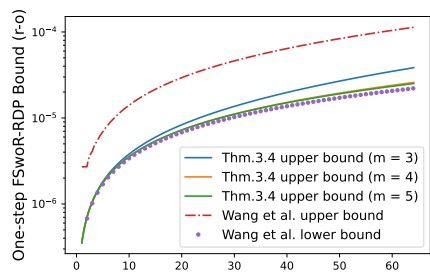

Figure 2: Comparison of FS$_{\text{woR}}$-RDP bounds under replace-one adjacency from Theorem 3.4 for various choices of $m$ with the upper and lower bounds from Wang et al. (2019). $\sigma_t = 6$, $|B| = 120$, $|D| = 50,000$.

Following the setting in (Abadi et al., 2016), we provide corresponding $(\epsilon, \delta)$-DP guarantees in Figure 3 shows corresponding $\epsilon$ for $\delta \in \{1e-4, 1e-5, 1e-6, 1e-7, 1e-8, 1e-9, 1e-10\}$ after 250 training epochs in DP-SGD. To translate RDP bounds to $(\epsilon, \delta)$-DP, we used Theorem 21 in (Balle et al., 2020), which is also used in the Opacus library. As shown in Figure 3, our FS$_{\text{woR}}$ bounds with $m \in \{3, 4, 5\}$ (solid lines) are close to the lower-bound provided in (Wang et al., 2019) (circles) over a range of $\delta$'s that depend on the value of $m$. These guarantees are significantly tighter than the upper bound from Wang et al. (2019) (dashed line). In our experiments, we observe that $m = 4$ and $m = 5$ yields essentially the same results and we find that $m = 4$ suffices to obtain good results in practice while outperforming (Wang et al., 2019).

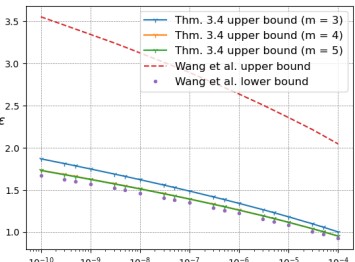

Figure 3: FS$_{\text{woR}}$ $(\epsilon, \delta)$-DP guarantees under replace-one adjacency; comparison of Wang et al. (2019) upper and lower bounds with our Theorem 3.4 for various choices of $m$. $\sigma_t = 6$, $|B| = 120$, $|D| = 50,000$.

## 4 Comparing Fixed-size and Poisson Subsampling

In Sec. 3.4 we demonstrated that our Rényi-DP-SGD bounds with fixed-size subsampling in Thm. 3.3 are tighter than the general-purpose fixed-size subsampling method from Wang et al. (2019). However, the most commonly used implementations of DP-SGD rely on Poisson subsampling, not fixed-size, and there the comparison is more nuanced. In this section we compare Poisson and fixed-size subsampling and point out advantages and disadvantages of each.

### 4.1 Privacy Comparison

In terms of privacy guarantees, all else being equal, when using replace-one adjacency we find that DP-SGD with fixed-size subsampling yields the same privacy guarantees as Poisson subsampling to leading order; specifically, when moving from add/remove to replace one-adjacency, our fixed-size subsampling DP bounds do not change at leading order but the Poisson subsampling DP bounds change by a factor of 2 to match the fixed-size results. In contrast, when using add/remove adjacency we find that Poisson subsampling has a natural advantage over fixed-size subsampling by approximately a factor of 2. Here we outline the reason for this intuitively.

The derivation of Rényi-DP bounds when using Poisson subsampling, (see Abadi et al. (2016); Mironov et al. (2019)) shares many steps with our derivation for fixed-size subsampling without replacement in Appendix A. In both cases, to leading order the bound on the Rényi divergence is proportional to $\|\mu - \mu'\|^2$, where $\mu$ and $\mu'$ are the sum of the gradients under the adjacent datasets $D$ and $D'$. When using Poisson subsampling, under add/remove adjacency, $\mu'$ differs from $\mu$ by the addition or deletion of a single term in the sum with probability $q$, while under replace-one adjacency, $\mu'$ differs from $\mu$ by the replacement of one element with probability $q$. Therefore in the former case, $\|\mu - \mu'\|$ is the norm of a single clipped vector, hence it scales with the clipping threshold, $C$, while

in the latter, $\|\mu - \mu'\|$ is the norm of the difference between two clipped vector, hence it scales with $2C$, as in the worst case the two vectors are anti-parallel. In fixed-size subsampling and under both adjacency relations, $\mu'$ differs from $\mu$ by replacing one element with another (due to the fixed-size minibatch constraint) and therefore $\|\mu - \mu'\|$ scales like $2C$. After squaring and at leading order, this leads to a factor of $4$ difference in the Rényi divergences between Poisson and fixed-size cases under add/remove adjacency while under replace-one adjacency the Poisson and fixed-size results agree; see Appendix C.1 for a more precise analysis of the latter comparison.

Translating this to $(\epsilon, \delta)$-DP then leads to an approximate factor of $2$ difference in $\epsilon$ for the same $\delta$ under add/remove adjacency, as shown in Appendix G, while under replace-one adjacency the different subsampling methods lead to the same DP guarantees at leading order. When the higher order terms are taken into account, Poisson subsampling regains a slight privacy advantage under replace-one adjacency; see Figures 4 and 7.

### 4.2  Comparison on CIFAR10

Our numerical comparisons in Section 3.4 showed that $\text{FS}_\text{woR}$-RDP yields significantly tighter guarantees than its counterpart from Wang et al. (2019). It is also useful to compare the empirical privacy guarantees of fixed-size versus the Poisson subsampling RDP commonly used in DP implementations such as Opacus. To this end, we use CIFAR10 and a simple convolutional neural network (CNN) to compare the canonical Poisson subsampling RDP with FS-RDP. Our setup closely follows the procedure in (Abadi et al., 2016). Note that the goal of this comparison is not to achieve the state-of-the-art privacy guarantees on CIFAR10 with any specific neural network. Instead we aim to compare the performance and guarantees with the alternative fixed-size and Poisson subsampling methods. Following Abadi et al. (2016), we used a simple CNN network that is pre-trained on CIFAR100 dataset non-privately. Our CNN has six convolutional layers 32, 32, 64, 128, 128, and 256 square filters of size $3 \times 3$, in each layer respectively. Since using Batch Normalization layers leads to privacy violation, we used group normalization instead as a common practice that is also used the Opacus library (Meta Platforms, 2024). Our network has 3 group normalization layers of size 64, 128, and 256 after the second, fourth, and sixth convolution layer. Similar to Abadi et al. (2016), this model reaches the accuracy of 81.33% after 250 epochs of training in a non-private setting. For the Poisson subsampling RDP, we simply used the one in Opacus.

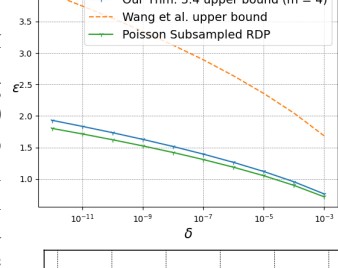

Fig. 4 compares the privacy guarantees $(\epsilon, \delta)$ for $\text{FS}_\text{woR}$-RDP and the method proposed by Wang et al. (2019) on the above CNN model after 250 training epochs with $\sigma_t = 6$, $|B| = 120$, clipping threshold $C = 3.0$, and learning rate $lr = 1e-3$ on CIFAR10 following the same setting in Abadi et al. (2016). Fig. 4 also compares the testing accuracy of Poisson subsampled RDP in Opacus with our $\text{FS}_\text{woR}$-RDP. We ran each method 5 times with different random seeds for sampling: $0, 1, 364, 2, 560, 3, 000$, and $4, 111$. Shaded areas show 3-sigma standard deviations around the mean. Both logical and physical minibatch sizes were set to 120 in Opacus. Experiments were run on a single work station with an NVIDIA RTX 3090 with 24GB memory. The runtime for all experiments was under 12 hours. As shown in Figure 4, $\text{FS}_\text{woR}$-RDP yields substantially tighter bounds than (Wang et al., 2019) (top panel) and reaches the average accuracy of 63.87% after 250 epochs (vs. 61.57% for Poisson subsampled RDP) (bottom panel). Figure 4 indicates that for a fixed $\sigma$, $\text{FS}_\text{woR}$-RDP surpasses the accuracy of Poisson subsampled RDP in Opacus, but with slightly higher epsilon guarantees, as shown in the top panel. This behavior matches the approximate calculation from Eq. (150) as well as the discussion in Section 4.1; specifically, Poisson and $\text{FS}_\text{woR}$-RDP agree to leading order but the higher order contributions give Poisson subsampling a slight privacy advantage at the same level

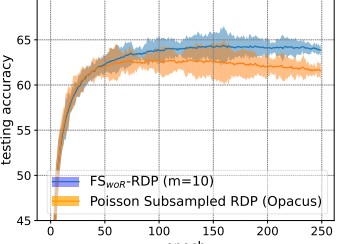

Figure 4: Privacy guarantees of $\text{FS}_\text{woR}$-RDP, Wang et al.'s, and Poisson Subsampled RDP (**top**). Comparing $\text{FS}_\text{woR}$-RDP performance against Poisson subsampled RDP (**bottom**). $\sigma_t = 6$, $C = 3$, $|B| = 120$, $|D| = 50,000$, $lr = 1e-3$.

of added noise. We emphasize that these results all apply to the replace-one notion of adjacency, as used in (Wang et al., 2019) and also our Theorem 3.4; we did not find existing computable RDP bounds for Poisson-subsampling under replace-one adjacency and so we used the new result in Theorem C.9. We conjecture that the performance difference observed in the bottom panel of Figure 4

is due to the difference in variance between Poisson and fixed-size subsampling, which we derive in Appendix F. Specifically, there we show that fixed-size subsampling has reduced variance compared to Poisson when the parameters are away from a minimizer. Additional experiments, gauging the sensitivity of $FS_{woR}$-RDP to key parameters including $\sigma$ and $|B|$, are given in Appendix H.

### 4.3 Memory Usage Comparison

The RDP accountant used in common DP libraries assumes a Poisson subsampling mechanism that leads to variable-sized minibatches during the training. That is, the size of each mini-batch cannot be determined in advance. Allowing the minibatches to have variable sizes creates an engineering challenge: an uncharacteristically large minibatch can cause an out-of-memory error, and depending on how fine tuned the minibatch size is to the available GPU memory this could be a frequent occurrence. To tackle this issue, the Opacus implementation requires the additional complication of wrapping the Pytorch `DataLoader` object into a `BatchMemoryManager` object, to alleviate the memory intensive nature of Poisson subsampling. `BatchMemoryManager` requires privacy practitioners to define two minibatch sizes: one is a *logical* minibatch size for Poisson sampling and the other is a *physical* minibatch size that determines the actual space allocated in memory and is determined by the practitioner via a variable named `max_physical_batch_size` (Meta Platforms, 2024). This design follows the distinction between 'lots' and minibatches described in the moments accountant (Abadi et al., 2016).

Our proposed FS-RDP does not require this memory management and is much less memory intensive. Figure 5 depicts the GPU memory footprints during 100 epochs of privately training the above CNN on CIFAR10 with $FS_{woR}$-RDP and three of the most common DP accountants: RDP (Mironov et al., 2019), $f$-DP (Dong et al., 2022), and PRV (Gopi et al., 2021). As shown in Figure 5, the memory usage of FS-RDP remains constant at 14,826 MB. However, the memory usage varies from 7,984 MB to 23,280 MB (almost twice larger than FS-RDP) for Poisson subsampled RDP in Opacus. Similar memory usage is observed for other Opacus accountants that use Poisson subsampling mechanism (i.e., $f$-DP and PRV). This fluctuating memory usage could result in undesirable outcomes in settings where resource-constrained devices are considered. For instance, in federated learning environments,

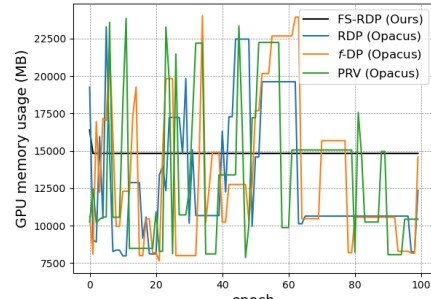

Figure 5: Comparing memory usage of FS-RDP with other Opacus privacy accountants in each training epoch. We used $|B| = 120$, and $|D| = 50,000$. Unlike other methods, FS-RDP's memory usage remains constant.

erratic memory usage could result in a higher dropout rate and delay the global model's convergence (Liu et al., 2021). It could also undermine model's accuracy and fairness, preventing lower-end devices from participating in the training process (Imteaj et al., 2021).

## 5 Conclusion

Differentially private stochastic gradient descent with fixed-size minibatches has attractive properties including reduced gradient estimator variance and simplified memory management. Our work presents a holistic RDP accountant for DP-SGD with fixed-size subsampling without replacement (FSwoR) and with replacement (FSwR) and, in the FSwoR case, consider both add/remove and replace-one adjacency. As we showed theoretically and empirically, since FSwoR under replace-one adjacency leads to the same leading-order privacy guarantees as the widely-used Poisson subsampling, we suggest using the former over the latter to benefit from the memory management and reduced variance properties. For subsampling without replacement under replace-one adjacency, we obtained significantly tighter RDP bounds (4 times improvement) over the most recent computable results (Wang et al., 2019). For subsampling with replacement we obtained the first non-asymptotic upper and lower RDP bounds for DP-SGD. We also provided the first comparison of gradient estimator's variance and privacy guarantees between FS-RDP and Poisson subsampled RDP commonly used in DP libraries. Our analysis revealed that FS-RDP reduces the variance of the gradient estimator. We highlighted the memory usage advantages of FS-RDP over Poisson subsampled RDP, which makes it a practical choice for privately training large deep learning models. We made FS-RDP's implementation and its evaluations available online such that it can be added to common DP libraries.

## Acknowledgements

Behnia acknowledges support from the University of South Florida Sarasota-Manatee Office of Research via the Interdisciplinary Research Grant program.

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

# A Proof of Theorem 3.1: FS$_{\text{woR}}$-RDP Bound under Add/Remove Adjacency

In this appendix we prove a one-step Rényi bound for DP-SGD using subsampling without replacement under the add/remove adjacency relation, as stated in Theorem 3.1. To do this, we will show that the sum over minibatches $b$ in (5) can be rewritten as an average of Gaussian mixtures with parameter $q = |B|/|D|$. This will then allow us to mimic several steps from the derivation in Mironov et al. (2019), which applies to Poisson subsampling. In the Poisson subsampling case the Gaussian mixture structure is more or less apparent from the start, but here it requires some additional work to isolate. We first consider the case where $D'$ is obtained from $D$ by removing one element. Without loss of generality we can let $D' = (D_0, ..., D_{|D|-2})$. Later we will show that the bound we derive in this case also bounds the case where $D'$ has one additional element.

Noting that $\frac{1}{N_D} \sum_b$ is simply the expectation with respect to the distribution of $B_t^D$, we start by showing that the distribution of $B_t^D$ can be obtained from the distribution of $B_t^{D'}$ through the introduction of two auxiliary random variables $J$ and $\widetilde{B}$ as follows: Let $J$ be a Bernoulli($q$) random variable where $q = |B|/|D|$ and let $(B', \widetilde{B})$ be random variables independent of $J$ such that $B'$ is a uniformly selected subset of $\{0, ..., |D|-2\}$ of size $|B|$ (so $B' \sim B_t^{D'}$) and $P(\widetilde{B} \in \cdot | B' = b')$ is the uniform distribution on subsets of $b'$ of size $|B|-1$. Defining the random variable $B$ by

$$B = \begin{cases} B' \text{ if } J = 0, \\ \widetilde{B} \cup \{|D|-1\} \text{ if } J = 1 \end{cases} \tag{20}$$

we have the following:

**Lemma A.1.** *The distribution of $B$, defined in (20), equals that of $B_t^D$.*

*Proof.* It is clear that $B$ is valued in the subsets of $\{0, ..., |D|-1\}$ having size $|B|$. Given $b \subset \{0, ..., |D|-1\}$ of size $|B|$ we have

$$P(B = b) = P(B = b, J = 0) + P(B = b, J = 1) \tag{21}$$
$$= P(B' = b)P(J = 0) + P(\widetilde{B} \cup \{|D|-1\} = b)P(J = 1).$$

If $|D| - 1 \notin b$ then $b \subset \{0, ..., |D|-2\}$ and the second term in (21) is zero, hence

$$P(B = b) = P(B' = b)(1 - q) = \frac{1 - q}{\binom{|D|-1}{|B|}} = \frac{(|D| - |B|)/|D|}{(|D|-1)!/(|B|!(|D|-1-|B|)!)} \tag{22}$$

$$= \frac{1}{|D|!/(|B|!(|D|-|B|)!)} = \frac{1}{\binom{|D|}{|B|}} = P(B_t^D = b)$$

as claimed. Now suppose $|D| - 1 \in b$. Then $b \setminus \{|D|-1\}$ is a subset of $\{0, ..., |D|-2\}$ of size $|B| - 1$, hence

$$P(B = b) = qP(\widetilde{B} \cup \{|D|-1\} = b) = \frac{|B|}{|D|} P(\widetilde{B} = b \setminus \{|D|-1\}) \tag{23}$$

$$= \frac{|B|}{|D|} E_{b' \sim B'} \left[ P(\widetilde{B} = b \setminus \{|D|-1\} | B' = b') \right]$$

$$= \frac{|B|}{|D|} E_{b' \sim B'} \left[ 1_{b \setminus \{|D|-1\} \subset b'} P(\widetilde{B} = b \setminus \{|D|-1\} | B' = b') \right]$$

$$= \frac{1}{|D|} P(b \setminus \{|D|-1\} \subset B') = \frac{1}{|D|} \frac{|D| - 1 - (|B|-1)}{\binom{|D|-1}{|B|}}$$

$$= \frac{1}{\binom{|D|}{|B|}} = P(B_t^D = b).$$

This completes the proof. $\square$

Using Lemma A.1, we can express the expectation with respect to $B_t^D$ in terms of $J$, $B'$, and $\widetilde{B}$. This will provide us with the desired decomposition of the sum in (5). The definition (20) is reminiscent

of the constructions in Proposition 23 in Appendix C of Wang et al. (2019), however our analysis will also result in tighter bounds for DP-SGD than one gets from the general-purpose method in Wang et al. (2019).

Using Lemma A.1 in the manner discussed above, the transition probabilities (5) for one step of DP-SGD with $\text{FS}_{\text{woR}}$-subsampling can be rewritten as follows:

$$
\begin{aligned}
p_t(\theta_{t+1}|\theta_t, D) =& E_{b \sim B_t^D} \left[ N_{\mu_t(\theta_t, b, D), \widetilde{\sigma}_t^2}(\theta_{t+1}) \right] \tag{24} \\
=& E_{(b', \widetilde{b}) \sim (B', \widetilde{B})} \left[ q N_{\mu_t(\theta_t, \widetilde{b} \cup \{|D|-1\}, D), \widetilde{\sigma}_t^2}(\theta_{t+1}) + (1-q) N_{\mu_t(\theta_t, b', D), \widetilde{\sigma}_t^2}(\theta_{t+1}) \right] .
\end{aligned}
$$

Recalling that $B' \sim B_t^{D'}$ and noting that $\mu_t(\theta_t, B', D') = \mu_t(\theta_t, B', D)$ (as $D'$ agrees with $D$ except for the last missing entry, which by definition cannot be contained in $B'$), the transition probabilities $p_t(\theta_{t+1}|\theta_t, D')$ can similarly be written

$$
\begin{aligned}
p_t(\theta_{t+1}|\theta_t, D') =& E_{b' \sim \widetilde{B}_t^{D'}} \left[ N_{\mu_t(\theta_t, b', D')}(\theta_{t+1}) \right] \tag{25} \\
=& E_{(b', \widetilde{b}) \sim (B', \widetilde{B})} \left[ N_{\mu_t(\theta_t, b', D), \widetilde{\sigma}_t^2}(\theta_{t+1}) \right] .
\end{aligned}
$$

The outer expectations in the expressions (24) and (25) are now both with respect to the same distribution, with the key difference being captured by the parameter $q$. The construction (20) was introduced precisely so that we might obtain such a decomposition. It allows the next step to proceed as in DP-SGD with Poisson subsampling Abadi et al. (2016); Mironov et al. (2019). Namely, we use quasiconvexity of the Rényi divergences (i.e., the data processing inequality) Van Erven and Harremos (2014) to obtain

$$
\begin{aligned}
& D_\alpha(p(\theta_{t+1}|\theta_t, D) \| p_t(\theta_{t+1}|\theta_t, D')) \tag{26} \\
\leq & \max_{(b', \widetilde{b})} D_\alpha \left( q N_{\mu_t(\theta_t, \widetilde{b} \cup \{|D|-1\}, D), \widetilde{\sigma}_t^2} + (1-q) N_{\mu_t(\theta_t, b', D), \widetilde{\sigma}_t^2} \| N_{\mu_t(\theta_t, b', D), \widetilde{\sigma}_t^2} \right) \\
= & \max_{(b', \widetilde{b})} D_\alpha \left( q N_{\|\mu_t(\theta_t, \widetilde{b} \cup \{|D|-1\}, D) - \mu_t(\theta_t, b', D)\|, \widetilde{\sigma}_t^2} + (1-q) N_{0, \widetilde{\sigma}_t^2} \| N_{0, \widetilde{\sigma}_t^2} \right) ,
\end{aligned}
$$

where, as in Abadi et al. (2016); Mironov et al. (2019), we used a change of variables to reduce the computation to 1-dimension in the last line.

*Remark* A.2. We note that the above analysis can be applied to any divergence, $\mathcal{D}$, that is quasiconvex in its arguments, not just the Rényi divergences. In the general case one similarly finds

$$
\begin{aligned}
& \mathcal{D}(p(\theta_{t+1}|\theta_t, D) \| p_t(\theta_{t+1}|\theta_t, D') \tag{27} \\
\leq & \max_{(b', \widetilde{b})} \mathcal{D} \left( q N_{\mu_t(\theta_t, \widetilde{b} \cup \{|D|-1\}, D), \widetilde{\sigma}_t^2} + (1-q) N_{\mu_t(\theta_t, b', D), \widetilde{\sigma}_t^2} \| N_{\mu_t(\theta_t, b', D), \widetilde{\sigma}_t^2} \right) .
\end{aligned}
$$

For instance, this holds for all $f$ divergences, including the important class of hockey-stick divergences that are important for other DP frameworks such as in Zhu et al. (2022).

Returning to (26), recalling that $C$ is the clipping bound and that $\widetilde{b} \subset b'$ and $|b' \setminus \widetilde{b}| = 1$, we can compute

$$
\begin{aligned}
& \|\mu_t(\theta_t, \widetilde{b} \cup \{|D|-1\}, D) - \mu_t(\theta_t, b', D)\|^2 \tag{28} \\
= & \frac{\eta_t^2}{|B|^2} \|\text{Clip}(\nabla_\theta \mathcal{L}(\theta_t, D_{|D|-1})) - \sum_{i \in b' \setminus \widetilde{b}} \text{Clip}(\nabla_\theta \mathcal{L}(\theta_t, D_i))\|^2 \\
\leq & \frac{4 C^2 \eta_t^2}{|B|^2} := r_t^2 . \tag{29}
\end{aligned}
$$

We note that this result is 4 times as large as the corresponding calculation in the case of Poisson subsampling, i.e., the sensitivity in the case of fixed-sized subsampling is inherently twice that of Poisson subsampling when using add/remove adjacency. This is because in Poisson subsampling when the minibatch from $D$ differs from that of $D'$ it is due to the inclusion of a single additional element. However, in fixed-size subsampling, when the minibatches are not identical then they differ by a replacement; this contributes more to the difference in means by a factor of 2.

Combining (29) with the fact that scaling the means in Gaussian mixtures by the same factor $c \geq 1$ can only increase the Rényi divergence, see Section 2 in Mironov et al. (2019), we obtain the following uniform Rényi bound.

$$\sup_{\theta_t} D_\alpha(p(\theta_{t+1}\|\theta_t, D)|p_t(\theta_{t+1}|\theta_t, D')) \leq D_\alpha\left(qN_{r_t,\widetilde{\sigma}_t^2} + (1-q)N_{0,\widetilde{\sigma}_t^2}\|N_{0,\widetilde{\sigma}_t^2}\right). \quad (30)$$

Now we show that the right-hand-side of (30) also bounds the case where $D'$ has one additional element. Repeating the steps in the above derivation, in this case we obtain

$$\sup_{\theta_t} D_\alpha(p(\theta_{t+1}|\theta_t, D)\|p_t(\theta_{t+1}|\theta_t, D')) \leq D_\alpha(N_{0,\widetilde{\sigma}_t^2}\|\widetilde{q}N_{r_t,\widetilde{\sigma}_t^2} + (1-\widetilde{q})N_{0,\widetilde{\sigma}_t^2}), \quad (31)$$

where $\widetilde{q} = |B|/(|D|+1)$; the appearance of $|D'| = |D| + 1$ here is due to the interchange of the roles of $D$ and $D'$ in the decomposition (20) for this case. Next use Theorem 5 in Mironov et al. (2019) to obtain

$$D_\alpha(N_{0,\widetilde{\sigma}_t^2}\|\widetilde{q}N_{r_t,\widetilde{\sigma}_t^2} + (1-\widetilde{q})N_{0,\widetilde{\sigma}_t^2}) \leq D_\alpha(\widetilde{q}N_{r_t,\widetilde{\sigma}_t^2} + (1-\widetilde{q})N_{0,\widetilde{\sigma}_t^2}\|N_{0,\widetilde{\sigma}_t^2}). \quad (32)$$

Quasiconvexity of the Rényi divergences implies the right-hand-side of (32) is non-decreasing in $\widetilde{q}$. Therefore, as $\widetilde{q} \leq q$, we obtain

$$\sup_{\theta_t} D_\alpha(p(\theta_{t+1}|\theta_t, D)\|p_t(\theta_{t+1}|\theta_t, D')) \leq D_\alpha(qN_{r_t,\widetilde{\sigma}_t^2} + (1-q)N_{0,\widetilde{\sigma}_t^2}\|N_{0,\widetilde{\sigma}_t^2}). \quad (33)$$

The upper bound here is the same as the one we obtained in (30) when $D'$ had one fewer element than $D$. Finally, by changing variables we can rescale the mean to 1 and therefore, noting that

$$\widetilde{\sigma}_t/r_t = \sigma_t/2, \quad (34)$$

we obtain

$$\sup_{\theta_t} D_\alpha(p(\theta_{t+1}|\theta_t, D)\|p_t(\theta_{t+1}|\theta_t, D')) \leq D_\alpha(qN_{1,\sigma_t^2/4} + (1-q)N_{0,\sigma_t^2/4}\|N_{0,\sigma_t^2/4}). \quad (35)$$

This completes the proof.

# B    Taylor Expansion of $H_{\alpha,\sigma}(q)$

In this appendix we provide details regarding the Taylor expansion (9) of $H_{\alpha,\sigma}(q)$ (Eq. (8)), which can be written

$$H_{\alpha,\sigma}(q) := \int \left(q\frac{N_{1,\sigma^2/4}(\theta)}{N_{0,\sigma^2/4}(\theta)} + (1-q)\right)^\alpha N_{0,\sigma^2/4}(\theta)d\theta.$$

In Section 3.2.1 we obtained one-step RDP bounds in terms of the Rényi divergence

$$D_\alpha(qN_{1,\sigma_t^2/4} + (1-q)N_{0,\sigma_t^2/4}\|N_{0,\sigma_t^2/4}) = \frac{1}{\alpha-1}\log[H_{\alpha,\sigma_t}(q)]. \quad (36)$$

To obtain the computable RDP bounds in Theorem 3.3 and also in Theorem 3.6 we must bound $H_{\alpha,\sigma}(q)$. We proceed by computing the coefficients in its Taylor expansion, including an explicit upper bound on the remainder term (10). First note that using the dominated convergence theorem (see, e.g., Theorem 2.27 in Folland (1999)), it is straightforward to see that $H_{\alpha,\sigma}(q)$ is smooth in $q$ and can be differentiated under the integral any number of times:

$$\frac{d^k}{dq^k}H_{\alpha,\sigma}(q) = \prod_{j=0}^{k-1}(\alpha-j)\int\left(q\frac{N_{1,\sigma^2/4}(\theta)}{N_{0,\sigma^2/4}(\theta)} + (1-q)\right)^{\alpha-k}\left(\frac{N_{1,\sigma^2/4}(\theta)}{N_{0,\sigma^2/4}(\theta)} - 1\right)^k N_{0,\sigma^2/4}(\theta)d\theta. \quad (37)$$

This justifies the use of Taylor's theorem with integral remainder to $H_{\alpha,\sigma}$ for any order $m \in \mathbb{Z}^+$ to arrive at

$$H_{\alpha,\sigma}(q) = 1 + \sum_{k=2}^{m-1}\frac{q^k}{k!}\left(\prod_{j=0}^{k-1}(\alpha-j)\right)M_{\sigma,k} + R_{\alpha,\sigma,m}(q), \quad (38)$$

where

$$M_{\sigma,k} := \int \left( \frac{N_{1,\sigma^2/4}(\theta)}{N_{0,\sigma^2/4}(\theta)} - 1 \right)^k N_{0,\sigma^2/4}(\theta)d\theta \tag{39}$$

(note that the $M_{\sigma,1} = 0$, which is why the summation in Eq. (38) starts at $k = 2$) and the remainder term is given by

$$R_{\alpha,\sigma,m}(q) = q^m \int_0^1 \frac{(1-s)^{m-1}}{(m-1)!} \frac{d^m}{dq^m} H_{\alpha,\sigma}(sq)ds \,. \tag{40}$$

For integer $k \geq 0$, the $M_{\sigma,k}$'s can be computed using the binomial theorem together with the formula for the moment generating function (MGF) of a Gaussian:

$$
\begin{aligned}
M_{\sigma,k} &= \sum_{\ell=0}^{k} (-1)^{k-\ell} \binom{k}{\ell} \int \left( \frac{N_{1,\sigma^2/4}(\theta)}{N_{0,\sigma^2/4}(\theta)} \right)^\ell N_{0,\sigma^2/4}(\theta)d\theta \\
&= \sum_{\ell=0}^{k} (-1)^{k-\ell} \binom{k}{\ell} e^{-\ell/(2\sigma^2/4)} \int e^{\ell\theta/(\sigma^2/4)} N_{0,\sigma^2/4}(\theta)d\theta \\
&= \sum_{\ell=0}^{k} (-1)^{k-\ell} \binom{k}{\ell} e^{\ell(\ell-1)/(\sigma^2/2)} \\
&= \sum_{\ell=2}^{k} (-1)^{k-\ell} \binom{k}{\ell} e^{2\ell(\ell-1)/\sigma^2} + (-1)^{k-1}(k-1) \,.
\end{aligned}
\tag{41}
$$

In particular, $M_{\sigma,0} = 1$ and $M_{\sigma,1} = 0$. If $\alpha$ is an integer then the expansion (38) truncates at a finite number of terms (terms with $k > \alpha$ are zero), but for non-integer $\alpha$ one must bound the remainder term. To enhance numerical stability of the computations, in practice we add terms starting from the highest order and proceeding to the lowest order.

**Bounding the Taylor Expansion Remainder Term:**
To bound the remainder term (40) we need to bound $\frac{d^m}{dq^m} H_{\alpha,\sigma}(sq)$ for $t, q \in (0,1)$. We will break the calculation into two cases, where different methods are appropriate. Here we assume $q < 1$.

In these bounds, it will be useful to employ the following definition for integer $j \geq 0$:

$$B_{\sigma,j} := \int \left| \frac{N_{1,\sigma^2/4}(\theta)}{N_{0,\sigma^2/4}(\theta)} - 1 \right|^j N_{0,\sigma^2/4}(\theta)d\theta \,. \tag{42}$$

For $j$ even we have $B_{\sigma,j} = M_{\sigma,j}$ and for $j$ odd we can use the Cauchy-Schwarz inequality to bound

$$B_{\sigma,j} \leq B_{\sigma,j-1}^{1/2} B_{\sigma,j+1}^{1/2} = M_{\sigma,j-1}^{1/2} M_{\sigma,j+1}^{1/2} \,, \tag{43}$$

as $j \pm 1$ are both even. Therefore an explicitly computable upper bound for all integer $j \geq 1$ is given by

$$B_{\sigma,j} \leq \begin{cases} M_{\sigma,j} & \text{if } j \text{ even} \\ M_{\sigma,j-1}^{1/2} M_{\sigma,j+1}^{1/2} & \text{if } j \text{ odd} \end{cases} =: \widetilde{B}_{\sigma,j} \,, \tag{44}$$

where $M_{\sigma,k}$ is given by (41).

**Case 1:** $\alpha - m > 0$

Using the bound $x^{\alpha-m} \leq 1 + x^{\lceil\alpha\rceil-m}$ for all $x > 0$ we can compute

$$\left|\frac{d^m}{dq^m}H_{\alpha,\sigma}(sq)\right| \tag{45}$$

$$\leq \prod_{j=0}^{m-1}|\alpha-j|\int\left(1+\left(sq\frac{N_{1,\sigma^2/4}(\theta)}{N_{0,\sigma^2/4}(\theta)}+(1-sq)\right)^{\lceil\alpha\rceil-m}\right)\left|\frac{N_{1,\sigma^2/4}(\theta)}{N_{0,\sigma^2/4}(\theta)}-1\right|^m N_{0,\sigma^2/4}(\theta)d\theta$$

$$= \prod_{j=0}^{m-1}|\alpha-j|\left[\int\left(sq\left(\frac{N_{1,\sigma^2/4}(\theta)}{N_{0,\sigma^2/4}(\theta)}-1\right)+1\right)^{\lceil\alpha\rceil-m}\left|\frac{N_{1,\sigma^2/4}(\theta)}{N_{0,\sigma^2/4}(\theta)}-1\right|^m N_{0,\sigma^2/4}(\theta)d\theta + B_{\sigma,m}\right]$$

$$= \prod_{j=0}^{m-1}|\alpha-j|\left[\sum_{\ell=0}^{\lceil\alpha\rceil-m}\binom{\lceil\alpha\rceil-m}{\ell}(sq)^\ell\int\left(\frac{N_{1,\sigma^2/4}(\theta)}{N_{0,\sigma^2/4}(\theta)}-1\right)^\ell\right. \tag{46}$$

$$\left. \times \left|\frac{N_{1,\sigma^2/4}(\theta)}{N_{0,\sigma^2/4}(\theta)}-1\right|^m N_{0,\sigma^2/4}(\theta)d\theta + B_{\sigma,m}\right]$$

$$\leq \prod_{j=0}^{m-1}|\alpha-j|\left[\sum_{\ell=0}^{\lceil\alpha\rceil-m}\binom{\lceil\alpha\rceil-m}{\ell}(sq)^\ell\widetilde{B}_{\sigma,\ell+m}+\widetilde{B}_{\sigma,m}\right].$$

Therefore

$$|R_{\alpha,\sigma,m}(q)| \tag{47}$$

$$\leq q^m\prod_{j=0}^{m-1}|\alpha-j|\left[\sum_{\ell=0}^{\lceil\alpha\rceil-m}\binom{\lceil\alpha\rceil-m}{\ell}q^\ell\widetilde{B}_{\sigma,\ell+m}\int_0^1\frac{(1-s)^{m-1}}{(m-1)!}s^\ell ds+\frac{1}{m!}\widetilde{B}_{\sigma,m}\right]$$

$$= q^m\prod_{j=0}^{m-1}|\alpha-j|\left[\sum_{\ell=0}^{\lceil\alpha\rceil-m}q^\ell\frac{(\lceil\alpha\rceil-m)!}{(\lceil\alpha\rceil-m-\ell)!(m+\ell)!}\widetilde{B}_{\sigma,\ell+m}+\frac{1}{m!}\widetilde{B}_{\sigma,m}\right].$$

**Case 2:** $\alpha - m \leq 0$

In this case we have

$$\left(sq\frac{N_{1,\sigma^2/4}(\theta)}{N_{0,\sigma^2/4}(\theta)}+(1-sq)\right)^{\alpha-m}\leq(1-q)^{\alpha-m} \tag{48}$$

and therefore

$$\left|\frac{d^m}{dq^m}H_{\alpha,\sigma}(sq)\right|\leq(1-q)^{\alpha-m}\prod_{j=0}^{m-1}|\alpha-j|B_{\sigma,m}. \tag{49}$$

This leads to the following bound on the remainder

$$|R_{\alpha,\sigma,m}(q)|\leq q^m\int_0^1\frac{(1-s)^{m-1}}{(m-1)!}(1-q)^{\alpha-m}\prod_{j=0}^{m-1}|\alpha-j|B_{\sigma,m}ds \tag{50}$$

$$= \frac{q^m}{m!}(1-q)^{\alpha-m}\prod_{j=0}^{m-1}|\alpha-j|B_{\sigma,m}, \tag{51}$$

where $\widetilde{B}_{\sigma,m}$ is defined by (44).

**Explicit bounds for $m=3$:** We find that $m=3$ generally yields sufficiently accurate bounds. Here we specialize the above results to this case. For $q < 1$ we have

$$H_{\alpha,\sigma}(q) = 1 + \frac{q^2}{2}\alpha(\alpha-1)M_{\sigma,2} + R_{\alpha,\sigma,3}(q) \tag{52}$$

with

$$R_{\alpha,\sigma,3}(q) \le q^3\alpha(\alpha-1)|\alpha-2| \begin{cases} \sum_{\ell=0}^{\lceil\alpha\rceil-3} q^\ell \frac{(\lceil\alpha\rceil-3)!}{(\lceil\alpha\rceil-3-\ell)!(3+\ell)!}\widetilde{B}_{\sigma,\ell+3} + \frac{1}{6}\widetilde{B}_{\sigma,3} & \text{if } \alpha-3>0 \\ \frac{1}{6}(1-q)^{\alpha-3}\widetilde{B}_{\sigma,3} & \text{if } \alpha-3\le 0 \end{cases}. \tag{53}$$

This completes the proof of Theorem 3.2.

To illustrate the use of these bounds, and the fact that $m=3$ is often sufficient in practice, in Figure 6 we plot the $\text{FS}_{\text{woR}}$-RDP bound from Theorem 3.3 (which uses the expansion and bound (52) - (53)) for one step (i.e, $T=1$) and with the parameter values $\sigma_t=6$, $|B|=120$, and $|D|=50,000$. Note that the difference between the bound for $m=3$ and $m=4$ is negligible over a wide range of $\alpha$ (sufficient to cover the default $\alpha$ range needed by Meta Platforms (2024)). The value of $q$ in this example is far from being unrealistically small; smaller values of $q$ will only make the effect of $m$ even less important. Therefore $m=3$ is generally sufficient.

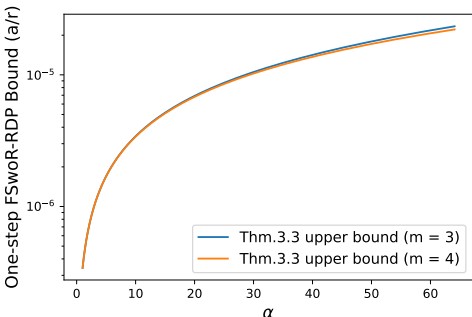

Figure 6: One step of Theorem 3.3 for $m=3,4$, using the parameter values $\sigma_t=6$, $|B|=120$, and $|D|=50,000$.

## C  Proof of Theorem 3.4: $\text{FS}_{\text{woR}}$-RDP Bound under Replace-One Adjacency

We now prove the $\text{FS}_{\text{woR}}$-RDP bound under replace-one adjacency as stated in Theorem 3.4 above. Without loss of generality, we can assume that $D$ and $D'$ share all elements except the last. The derivation shares several initial steps with the add/remove case from Appendix A. Specifically, we apply Lemma A.1 to the transition probabilities for one step of DP-SGD with $\text{FS}_{\text{woR}}$-subsampling, both for $D$ and for $D'$ (as they now have the same number of elements), to obtain

$$p_t(\theta_{t+1}|\theta_t, D) = E_{(b',\widetilde{b})\sim(B',\widetilde{B})}\left[qN_{\mu_t(\theta_t,\widetilde{b}\cup\{|D|-1\},D),\widetilde{\sigma}_t^2}(\theta_{t+1}) + (1-q)N_{\mu_t(\theta_t,b',D),\widetilde{\sigma}_t^2}(\theta_{t+1})\right],$$
$$p_t(\theta_{t+1}|\theta_t, D') = E_{(b',\widetilde{b})\sim(B',\widetilde{B})}\left[qN_{\mu_t(\theta_t,\widetilde{b}\cup\{|D|-1\},D'),\widetilde{\sigma}_t^2}(\theta_{t+1}) + (1-q)N_{\mu_t(\theta_t,b',D'),\widetilde{\sigma}_t^2}(\theta_{t+1})\right], \tag{54}$$

similarly to (24). Now we again use quasiconvexity, this time in both arguments simultaneously, to bound the effect of the mixture (expectation) over $(b',\widetilde{b})$ by the worst case, yielding the one-step Rényi bound

$$D_\alpha\left(p(\theta_{t+1}|\theta_t, D)\|p_t(\theta_{t+1}|\theta_t, D')\right) \le \max_{(b',\widetilde{b})} D_\alpha\left(qQ_{\widetilde{b}} + (1-q)P_{b'}\|qQ'_{\widetilde{b}} + (1-q)P'_{b'}\right), \tag{55}$$

$$Q_{\widetilde{b}} := N_{\mu_t(\theta_t,\widetilde{b}\cup\{|D|-1\},D),\widetilde{\sigma}_t^2}, \quad Q'_{\widetilde{b}} := N_{\mu_t(\theta_t,\widetilde{b}\cup\{|D'|-1\},D'),\widetilde{\sigma}_t^2},$$
$$P_{b'} := N_{\mu_t(\theta_t,b',D),\widetilde{\sigma}_t^2}, \quad P'_{b'} := N_{\mu_t(\theta_t,b',D'),\widetilde{\sigma}_t^2}.$$

Using the fact $D$ and $D'$ agree on the index minibatch $b'$, and hence $\mu_t(\theta_t, b', D) = \mu_t(\theta_t, b', D')$, we see that $P_{b'} = P'_{b'}$. We can therefore change variables to center both of their means at zero, giving

$$D_\alpha(p(\theta_{t+1}|\theta_t, D)\|p_t(\theta_{t+1}|\theta_t, D')) \tag{56}$$

$$\leq \max_{(b',\widetilde{b})} D_\alpha\left(qN_{\Delta\mu_t(b',\widetilde{b}),\widetilde{\sigma}_t^2} + (1-q)N_{0,\widetilde{\sigma}_t^2}\|qN_{\Delta\mu'_t(b',\widetilde{b}),\widetilde{\sigma}_t^2} + (1-q)N_{0,\widetilde{\sigma}_t^2}\right),$$

$$\Delta\mu_t(b',\widetilde{b}) := \mu_t(\theta_t, \widetilde{b} \cup \{|D| - 1\}, D) - \mu_t(\theta_t, b', D),$$

$$\Delta\mu'_t(b',\widetilde{b}) := \mu_t(\theta_t, \widetilde{b} \cup \{|D| - 1\}, D') - \mu_t(\theta_t, b', D).$$

The means of the Gaussians satisfy the bounds

$$\|\Delta\mu_t(b',\widetilde{b})\| \leq r_t, \quad \|\Delta\mu'_t(b',\widetilde{b})\| \leq r_t, \quad \|\Delta\mu_t(b',\widetilde{b}) - \Delta\mu'_t(b',\widetilde{b})\| \leq r_t, \tag{57}$$

where $r_t$ was defined in (29); the first two inequalities follow directly from (28) while the proof of the third is almost identical due to $D$ and $D'$ differing only in the last element.

*Remark* C.1. Note that, similarly to Remark A.2, Eq. (56) remains true when the Rényi divergences $D_\alpha$ are replaced by any other divergence $\mathcal{D}$ that is quasiconvex in both of its arguments, e.g., the hockey-stick divergences. This fact is useful for other differential privacy paradigms, though using the quasiconvexity bound does not always lead the tightest possible bounding pair of distributions; see Lebeda et al. (2024).

The key difference between the bound (56) and the corresponding result (26) in the add/remove adajcency case is that both arguments of the Rényi divergence on the right-hand side of (56) are now Gaussian mixtures with mixing parameter $q$; in the add/remove adjacency case only one argument is a Gaussian mixture. This difference makes the following computations more involved, though the same Taylor expansion technique can be applied.

Using the definition of Rényi divergences we can write

$$D_\alpha\left(qN_{\Delta\mu_t(b',\widetilde{b}),\widetilde{\sigma}_t^2} + (1-q)N_{0,\widetilde{\sigma}_t^2}\|qN_{\Delta\mu'_t(b',\widetilde{b}),\widetilde{\sigma}_t^2} + (1-q)N_{0,\widetilde{\sigma}_t^2}\right) = \frac{1}{\alpha-1}\log\left[F_\alpha(q)\right],$$

$$F_\alpha(q) := \int \frac{\left(qN_{\Delta\mu_t(b',\widetilde{b}),\widetilde{\sigma}_t^2}(\theta) + (1-q)N_{0,\widetilde{\sigma}_t^2}(\theta)\right)^\alpha}{\left(qN_{\Delta\mu'_t(b',\widetilde{b}),\widetilde{\sigma}_t^2}(\theta) + (1-q)N_{0,\widetilde{\sigma}_t^2}(\theta)\right)^{\alpha-1}}d\theta. \tag{58}$$

Next we Taylor expand the argument of the logarithm:

$$F_\alpha(q) = \sum_{k=0}^{m-1} \frac{q^k}{k!}\frac{d^k}{dq^k}F_\alpha(0) + E_{\alpha,m}(q), \tag{59}$$

where here the remainder term is given by

$$E_{\alpha,m}(q) := q^m \int_0^1 \frac{(1-s)^{m-1}}{(m-1)!}\frac{d^m}{dq^m}F_\alpha(sq)ds. \tag{60}$$

Using the formula $\frac{d^k}{dx^k}(fg) = \sum_{j=0}^k \binom{k}{j}\frac{d^j}{dx^j}f\frac{d^{k-j}}{dx^{k-j}}g$ along the dominated convergence theorem to justify differentiating under the integral, the derivatives can be computed as follows:

$$\frac{d^k}{dq^k}F_\alpha(q) \tag{61}$$

$$= \sum_{j=0}^k \binom{k}{j}\left(\prod_{\ell=0}^{j-1}(\alpha-\ell)\right)\left(\prod_{\ell=0}^{k-j-1}(1-\alpha-\ell)\right)\int \frac{\left(qN_{\Delta\mu_t(b',\widetilde{b}),\widetilde{\sigma}_t^2}(\theta) + (1-q)N_{0,\widetilde{\sigma}_t^2}(\theta)\right)^{\alpha-j}}{\left(qN_{\Delta\mu'_t(b',\widetilde{b}),\widetilde{\sigma}_t^2}(\theta) + (1-q)N_{0,\widetilde{\sigma}_t^2}(\theta)\right)^{\alpha+k-j-1}}$$

$$\times \left(N_{\Delta\mu_t(b',\widetilde{b}),\widetilde{\sigma}_t^2}(\theta) - N_{0,\widetilde{\sigma}_t^2}(\theta)\right)^j \left(N_{\Delta\mu'_t(b',\widetilde{b}),\widetilde{\sigma}_t^2}(\theta) - N_{0,\widetilde{\sigma}_t^2}(\theta)\right)^{k-j}d\theta.$$

**Taylor expansion coefficients for $k = 0, 1, 2$:** Evaluating (61) at $q = 0$ we have

$$\frac{d^k}{dq^k} F_\alpha(0) \tag{62}$$

$$= \sum_{j=0}^{k} (-1)^{k-j} \binom{k}{j} \left( \prod_{\ell=0}^{j-1} (\alpha - \ell) \right) \left( \prod_{\ell=0}^{k-j-1} (\alpha + \ell - 1) \right)$$

$$\times \int \left( N_{\Delta\mu_t(b',\widetilde{b}),\widetilde{\sigma}_t^2}(\theta) / N_{0,\widetilde{\sigma}_t^2}(\theta) - 1 \right)^j \left( N_{\Delta\mu_t'(b',\widetilde{b}),\widetilde{\sigma}_t^2}(\theta) / N_{0,\widetilde{\sigma}_t^2}(\theta) - 1 \right)^{k-j} N_{0,\widetilde{\sigma}_t^2}(\theta) d\theta \,.$$

Note that we have not yet been able to maximize over $b', \widetilde{b}$; doing so while incorporating the constraints (57) is nontrivial and constitutes one of the the primary difficulties in obtaining sufficiently tight bounds. We will make particular attention to the first few terms and then use looser (but still sufficiently tight in practice) approximations for the higher order terms. First note that $F_\alpha(0) = 1$ and

$$\frac{d}{dq} F_\alpha(0) = -(\alpha - 1) \int \left( N_{\Delta\mu_t'(b',\widetilde{b}),\widetilde{\sigma}_t^2}(\theta) - N_{0,\widetilde{\sigma}_t^2}(\theta) \right) d\theta \tag{63}$$

$$+ \alpha \int \left( N_{\Delta\mu_t(b',\widetilde{b}),\widetilde{\sigma}_t^2}(\theta) - N_{0,\widetilde{\sigma}_t^2}(\theta) \right) d\theta = 0 \,.$$

For $k = 2$ we can expand (62) and then evaluate the integrals by using the formula for the MGF of a Gaussian to obtain

$$\frac{d^2}{dq^2} F_\alpha(0) \tag{64}$$

$$= \alpha(\alpha - 1) \left( \exp\left( \frac{\|\Delta\mu_t(b',\widetilde{b})\|^2}{\widetilde{\sigma}_t^2} \right) + \exp\left( \frac{\|\Delta\mu_t'(b',\widetilde{b})\|^2}{\widetilde{\sigma}_t^2} \right) - 2\exp\left( \frac{\Delta\mu_t(b',\widetilde{b}) \cdot \Delta\mu_t'(b',\widetilde{b})}{\widetilde{\sigma}_t^2} \right) \right) \,.$$

Using the third constraints in Eq. (57) we obtain the bound

$$\exp\left( \frac{\Delta\mu_t(b',\widetilde{b}) \cdot \Delta\mu_t'(b',\widetilde{b})}{\widetilde{\sigma}_t^2} \right) \tag{65}$$

$$= \exp\left( \frac{\|\Delta\mu_t(b',\widetilde{b})\|^2 + \|\Delta\mu_t'(b',\widetilde{b})\|^2 - \|\Delta\mu_t(b',\widetilde{b}) - \Delta\mu_t'(b',\widetilde{b})\|^2}{2\widetilde{\sigma}_t^2} \right)$$

$$\geq \exp\left( \frac{\|\Delta\mu_t(b',\widetilde{b})\|^2 + \|\Delta\mu_t'(b',\widetilde{b})\|^2 - r_t^2}{2\widetilde{\sigma}_t^2} \right) \,.$$

Elementary calculus along with the bounds $\|\Delta\mu_t(b',\widetilde{b})\| \leq r_t$, $\|\Delta\mu_t'(b',\widetilde{b})\| \leq r_t$ then imply

$$\frac{d^2}{dq^2} F_\alpha(0) \leq 2\alpha(\alpha - 1) \left( e^{r_t^2/\widetilde{\sigma}_t^2} - e^{r_t^2/(2\widetilde{\sigma}_t^2)} \right) = 2\alpha(\alpha - 1) \left( e^{4/\sigma_t^2} - e^{2/\sigma_t^2} \right) \,, \tag{66}$$

where we used (34). These terms are sufficient to precisely capture the leading-order behavior.

**Bounding the coefficients for $k \geq 3$:** For the higher-order terms we obtain an upper bound which relies on the following lemmas. It will be useful to phrase various bounds in terms of the $\chi^\beta$ divergences.

**Definition C.2.** For $\beta > 0$ define the $\chi^\beta$ divergences

$$\mathcal{D}_{\chi^\beta}(Q\|P) = E_P[|dQ/dP - 1|^\beta] \,. \tag{67}$$

It will also be convenient to adopt the notation $\mathcal{D}_{\chi^0} := 1$, despite it not defining a divergence.

The following property is a imply consequence of Hölder's inequality.

**Lemma C.3.** *Let $\gamma \geq \beta > 0$. Then*

$$\mathcal{D}_{\chi^\beta}(Q\|P) \leq \mathcal{D}_{\chi^\gamma}(Q\|P)^{\beta/\gamma} \,. \tag{68}$$

The next lemma will be used to obtain a uniform bound, i.e., not dependent on $b'$, $\widetilde{b}$, or $\theta_t$.

**Lemma C.4.** *For $k \geq 0$ an integer we have*

$$\max_{b',\widetilde{b}} \mathcal{D}_{\chi^k}\left(N_{\Delta\mu_t(b',\widetilde{b}),\widetilde{\sigma}_t^2} \| N_{0,\widetilde{\sigma}_t^2}\right) \leq \widetilde{B}_{\sigma_t,k}\,, \tag{69}$$

$$\max_{b',\widetilde{b}} \mathcal{D}_{\chi^k}\left(N_{\Delta\mu_t(b',\widetilde{b}),\widetilde{\sigma}_t^2} \| N_{\Delta\mu_t'(b',\widetilde{b}),\widetilde{\sigma}_t^2}\right) \leq \widetilde{B}_{\sigma_t,k}\,,$$

*where $\widetilde{B}$ was defined in* (44). *These bounds also hold with $\Delta\mu_t$ and $\Delta\mu_t'$ interchanged.*

*Proof.* $\mathcal{D}_{\chi^k}$ is the $f$ divergence corresponding to $f_\beta(y) = |y - 1|^k$, therefore it is jointly convex in $(Q, P)$. Therefore the same argument as in Section 2 of Mironov et al. (2019) (which only relies on the convexity of the divergence) implies

$$\mathcal{D}_{\chi^k}\left(N_{\Delta\mu_t(b',\widetilde{b}),\widetilde{\sigma}_t^2} \| N_{0,\widetilde{\sigma}_t^2}\right) \leq \mathcal{D}_{\chi^k}\left(N_{r_t,\widetilde{\sigma}_t^2} \| N_{0,\widetilde{\sigma}_t^2}\right) \tag{70}$$

$$= \mathcal{D}_{\chi^k}\left(N_{1,\sigma_t^2/4} \| N_{0,\sigma_t^2/4}\right) = B_{\sigma_t,k} \leq \widetilde{B}_{\sigma_t,k}\,,$$

where we used (57) and (44). The second result in (69) is proven in a similar manner, where we must also change variables to center the second Gaussian at zero and then employ the third bound in (57). $\qquad\square$

The second lemma we require is a slight improvement on Lemma 24 of (Wang et al., 2019).

**Lemma C.5.** *Let $p, q, r$ be strictly positive probability densities and $k \geq 1$ be an integer. Then*

$$\int \left(\frac{p(\theta) - q(\theta)}{r(\theta)}\right)^k r(\theta) d\theta \tag{71}$$

$$\leq \mathcal{D}_{\chi^k}(q\|p) + \mathcal{D}_{\chi^k}(p\|q) + \mathcal{D}_{\chi^k}(p\|r) + \begin{cases} \mathcal{D}_{\chi^k}(q\|r) & \text{if } k \text{ is even} \\ 0 & \text{if } k \text{ is odd} \end{cases}.$$

*Remark* C.6. Note that a key difference between our technique and that of Wang et al. (2019) is that we only apply Lemma C.5 to the higher order terms (i.e., when $k \geq 2$) while we bounded the leading contribution (66) more precisely. In contrast, Wang et al. (2019) used Lemma C.5 (their Lemma 24) on all terms.

*Proof.* We proceed by breaking the domain of integration into four regions, depending on the sizes of $p, q, r$ and then use the appropriate bounds to replace one of the densities with another:

$$\int \left(\frac{p(\theta) - q(\theta)}{r(\theta)}\right)^k r(\theta) d\theta \tag{72}$$

$$= \int_{r \geq p} \frac{(p(\theta) - q(\theta))^k}{r(\theta)^{k-1}} d\theta + \int_{r < p, r \geq q} \frac{(p(\theta) - q(\theta))^k}{r(\theta)^{k-1}} d\theta$$

$$+ \int_{r < p, r < q, q \geq p} \frac{(p(\theta) - q(\theta))^k}{r(\theta)^{k-1}} d\theta + \int_{r < p, r < q, q < p} \frac{(p(\theta) - q(\theta))^k}{r(\theta)^{k-1}} d\theta$$

$$\leq \int_{r \geq p} \frac{|p(\theta) - q(\theta)|^k}{p(\theta)^{k-1}} d\theta + \int_{r < p, r \geq q} \frac{|p(\theta) - q(\theta)|^k}{q(\theta)^{k-1}} d\theta$$

$$+ \begin{cases} \int_{r < p, r < q, q \geq p} \frac{(q(\theta) - r(\theta))^k}{r(\theta)^{k-1}} d\theta & \text{if } k \text{ is even} \\ 0 & \text{if } k \text{ is odd} \end{cases} + \int_{r < p, r < q, q < p} \frac{(p(\theta) - r(\theta))^k}{r(\theta)^{k-1}} d\theta$$

$$\leq \int \frac{|p(\theta) - q(\theta)|^k}{p(\theta)^{k-1}} d\theta + \int \frac{|p(\theta) - q(\theta)|^k}{q(\theta)^{k-1}} d\theta$$

$$+ \begin{cases} \int \frac{|q(\theta) - r(\theta)|^k}{r(\theta)^{k-1}} d\theta & \text{if } k \text{ is even} \\ 0 & \text{if } k \text{ is odd} \end{cases} + \int \frac{(p(\theta) - r(\theta))^k}{r(\theta)^{k-1}} d\theta\,.$$

This completes the proof. $\qquad\square$

**Lemma C.7.** *Let $\beta_1, \beta_2 > 0$ with $\beta_1 + \beta_2$ and integer. Then*

$$\max_{b',\widetilde{b}} \int \left| N_{\Delta\mu_t(b',\widetilde{b}),\widetilde{\sigma}_t^2}(\theta)/N_{0,\widetilde{\sigma}_t^2}(\theta) - 1 \right|^{\beta_1} \left| N_{\Delta\mu_t'(b',\widetilde{b}),\widetilde{\sigma}_t^2}(\theta)/N_{0,\widetilde{\sigma}_t^2}(\theta) - 1 \right|^{\beta_2} N_{0,\widetilde{\sigma}_t^2}(\theta)d\theta \quad (73)$$

$$\leq \widetilde{B}_{\sigma_t, \beta_1+\beta_2},$$

*where $\widetilde{B}_{\sigma,k}$ was defined in* (44).

*Proof.* Let $p, q > 1$ be conjugate exponents, i.e., they satisfy $1/p + 1/q = 1$. Using Hölder's inequality we can compute

$$\int \left| N_{\Delta\mu_t(b',\widetilde{b}),\widetilde{\sigma}_t^2}(\theta)/N_{0,\widetilde{\sigma}_t^2}(\theta) - 1 \right|^{\beta_1} \left| N_{\Delta\mu_t'(b',\widetilde{b}),\widetilde{\sigma}_t^2}(\theta)/N_{0,\widetilde{\sigma}_t^2}(\theta) - 1 \right|^{\beta_2} N_{0,\widetilde{\sigma}_t^2}(\theta)d\theta \quad (74)$$

$$\leq \mathcal{D}_{\chi^{\beta_1 p}}\left( N_{\Delta\mu_t(b',\widetilde{b}),\widetilde{\sigma}_t^2} \| N_{0,\widetilde{\sigma}_t^2} \right)^{1/p} \mathcal{D}_{\chi^{\beta_2 q}}\left( N_{\Delta\mu_t'(b',\widetilde{b}),\widetilde{\sigma}_t^2} \| N_{0,\widetilde{\sigma}_t^2} \right)^{1/q}.$$

We now use a simple heuristic that allow for $p, q$ to be chosen so as to minimize the right-hand side of this upper bound when $\beta_1, \beta_2$ are large. Lemma C.4 together with (44) and (41) suggest that the two factors on the right-hand side approximately behave like $\exp(2p\beta_1^2/\sigma_t^2)$ and $\exp(2q\beta_2^2/\sigma_t^2)$ respectively. Therefore, to make the upper bound as tight as possible, we want to to minimize $p\beta_1^2 + q\beta_2^2$ subject to $p > 1$, $q = p/(p-1)$. Simple calculus shows that the minimum occurs at $p := 1 + \beta_2/\beta_1$, $q = 1 + \beta_1/\beta_2$. Making this choice, using Lemma C.3 to convert to integer orders, and then employing Lemma 69 we find

$$\int \left| N_{\Delta\mu_t(b',\widetilde{b}),\widetilde{\sigma}_t^2}(\theta)/N_{0,\widetilde{\sigma}_t^2}(\theta) - 1 \right|^{\beta_1} \left| N_{\Delta\mu_t'(b',\widetilde{b}),\widetilde{\sigma}_t^2}(\theta)/N_{0,\widetilde{\sigma}_t^2}(\theta) - 1 \right|^{\beta_2} N_{0,\widetilde{\sigma}_t^2}(\theta)d\theta \quad (75)$$

$$\leq \mathcal{D}_{\chi^{\lceil\beta_1 p\rceil}}\left( N_{\Delta\mu_t(b',\widetilde{b}),\widetilde{\sigma}_t^2} \| N_{0,\widetilde{\sigma}_t^2} \right)^{\beta_1/\lceil\beta_1 p\rceil} \mathcal{D}_{\chi^{\lceil\beta_2 q\rceil}}\left( N_{\Delta\mu_t'(b',\widetilde{b}),\widetilde{\sigma}_t^2} \| N_{0,\widetilde{\sigma}_t^2} \right)^{\beta_2/\lceil\beta_2 q\rceil}$$

$$\leq \left( \widetilde{B}_{\sigma_t,\lceil\beta_1 p\rceil} \right)^{\beta_1/\lceil\beta_1 p\rceil} \left( \widetilde{B}_{\sigma_t,\lceil\beta_2 q\rceil} \right)^{\beta_2/\lceil\beta_2 q\rceil}.$$

In the case where $\beta_1 + \beta_2$ is an integer the above can be simplified via

$$\left( \widetilde{B}_{\sigma_t,\lceil\beta_1 p\rceil} \right)^{\beta_1/\lceil\beta_1 p\rceil} \left( \widetilde{B}_{\sigma_t,\lceil\beta_2 q\rceil} \right)^{\beta_2/\lceil\beta_2 q\rceil} = \widetilde{B}_{\sigma_t,\beta_1+\beta_2}.$$

$\square$

We use these lemmas to bound the higher order derivatives as follows. First rewrite (62) as

$$\frac{d^k}{dq^k}F_\alpha(0) \quad (76)$$

$$=(\alpha-1)\alpha^{k-1}\sum_{j=0}^{k}(-1)^{k-j}\binom{k}{j}$$

$$\times \int \left( N_{\Delta\mu_t(b',\widetilde{b}),\widetilde{\sigma}_t^2}(\theta)/N_{0,\widetilde{\sigma}_t^2}(\theta) - 1 \right)^{j} \left( N_{\Delta\mu_t'(b',\widetilde{b}),\widetilde{\sigma}_t^2}(\theta)/N_{0,\widetilde{\sigma}_t^2}(\theta) - 1 \right)^{k-j} N_{0,\widetilde{\sigma}_t^2}(\theta)d\theta$$

$$+ (\alpha-1)\alpha^{k-1}\sum_{j=0}^{k}(-1)^{k-j}\binom{k}{j}\left[ \frac{\alpha}{\alpha-1}\left( \prod_{\ell=0}^{j-1}(1-\ell/\alpha) \right)\left( \prod_{\ell=0}^{k-j-1}(1+(\ell-1)/\alpha) \right) - 1 \right]$$

$$\times \int \left( N_{\Delta\mu_t(b',\widetilde{b}),\widetilde{\sigma}_t^2}(\theta)/N_{0,\widetilde{\sigma}_t^2}(\theta) - 1 \right)^{j} \left( N_{\Delta\mu_t'(b',\widetilde{b}),\widetilde{\sigma}_t^2}(\theta)/N_{0,\widetilde{\sigma}_t^2}(\theta) - 1 \right)^{k-j} N_{0,\widetilde{\sigma}_t^2}(\theta)d\theta$$

$$=(\alpha-1)\alpha^{k-1}\int \left( N_{\Delta\mu_t(b',\widetilde{b}),\widetilde{\sigma}_t^2}(\theta)/N_{0,\widetilde{\sigma}_t^2}(\theta) - N_{\Delta\mu_t'(b',\widetilde{b}),\widetilde{\sigma}_t^2}(\theta)/N_{0,\widetilde{\sigma}_t^2}(\theta) \right)^{k} N_{0,\widetilde{\sigma}_t^2}(\theta)d\theta$$

$$+ (\alpha-1)\alpha^{k-1}\sum_{j=0}^{k}(-1)^{k-j}\binom{k}{j}\left[ \frac{\alpha}{\alpha-1}\left( \prod_{\ell=0}^{j-1}(1-\ell/\alpha) \right)\left( \prod_{\ell=0}^{k-j-1}(1+(\ell-1)/\alpha) \right) - 1 \right]$$

$$\times \int \left( N_{\Delta\mu_t(b',\widetilde{b}),\widetilde{\sigma}_t^2}(\theta)/N_{0,\widetilde{\sigma}_t^2}(\theta) - 1 \right)^{j} \left( N_{\Delta\mu_t'(b',\widetilde{b}),\widetilde{\sigma}_t^2}(\theta)/N_{0,\widetilde{\sigma}_t^2}(\theta) - 1 \right)^{k-j} N_{0,\widetilde{\sigma}_t^2}(\theta)d\theta.$$

The first family of integrals can be bounded using Lemma C.5 and then Lemma C.4 to obtain

$$\int \left( N_{\Delta\mu_t(b',\widetilde{b}),\widetilde{\sigma}_t^2}(\theta)/N_{0,\widetilde{\sigma}_t^2}(\theta) - N_{\Delta\mu_t'(b',\widetilde{b}),\widetilde{\sigma}_t^2}(\theta)/N_{0,\widetilde{\sigma}_t^2}(\theta) \right)^k N_{0,\widetilde{\sigma}_t^2}(\theta)d\theta \tag{77}$$

$$\leq \begin{cases} 4M_{\sigma_t,k} & \text{if } k \text{ is even} \\ 3M_{\sigma_t,k-1}^{1/2}M_{\sigma_t,k+1}^{1/2} & \text{if } k \text{ is odd} \end{cases} .$$

The second family of integrals can be bounded by using Lemma C.7 (when $j, k-j > 0$) and Lemma C.4 (when $j = 0$ or $j = k$), which yields

$$\left| \int \left( N_{\Delta\mu_t(b',\widetilde{b}),\widetilde{\sigma}_t^2}(\theta)/N_{0,\widetilde{\sigma}_t^2}(\theta) - 1 \right)^j \left( N_{\Delta\mu_t'(b',\widetilde{b}),\widetilde{\sigma}_t^2}(\theta)/N_{0,\widetilde{\sigma}_t^2}(\theta) - 1 \right)^{k-j} N_{0,\widetilde{\sigma}_t^2}(\theta)d\theta \right| \tag{78}$$

$$\leq \widetilde{B}_{\sigma_t,k} ,$$

Therefore we find

$$\frac{d^k}{dq^k}F_\alpha(0) \tag{79}$$

$$\leq (\alpha-1)\alpha^{k-1} \begin{cases} 4M_{\sigma_t,k} & \text{if } k \text{ is even} \\ 3M_{\sigma_t,k-1}^{1/2}M_{\sigma_t,k+1}^{1/2} & \text{if } k \text{ is odd} \end{cases}$$

$$+ (\alpha-1)\alpha^{k-1} \sum_{j=0}^{k} \binom{k}{j} \left| \frac{\alpha}{\alpha-1} \left( \prod_{\ell=0}^{j-1}(1-\ell/\alpha) \right) \left( \prod_{\ell=0}^{k-j-1}(1+(\ell-1)/\alpha) \right) - 1 \right| \widetilde{B}_{\sigma_t,k}$$

$$:= \widetilde{F}_{\alpha,\sigma_t,k} , \tag{80}$$

where $M$ and $\widetilde{B}$ are given by (41) and (44) respectively.

**Bounding the remainder term:** We now proceed to bound the remainder term (60). Using (61), for $q < 1$ we can compute

$$|E_{\alpha,m}(q)| \leq q^m \int_0^1 \frac{(1-s)^{m-1}}{(m-1)!} \left| \frac{d^m}{dq^m}F_\alpha(sq) \right| ds \tag{81}$$

$$\leq q^m \sum_{j=0}^{m} \binom{m}{j} \left( \prod_{\ell=0}^{j-1}|\alpha-\ell| \right) \left( \prod_{\ell=0}^{m-j-1}(\alpha+\ell-1) \right)$$

$$\times \int_0^1 \frac{(1-s)^{m-1}}{(m-1)!} \int \frac{\left( sq N_{\Delta\mu_t(b',\widetilde{b}),\widetilde{\sigma}_t^2}(\theta)/N_{0,\widetilde{\sigma}_t^2}(\theta) + (1-sq) \right)^{\alpha-j}}{\left( sq N_{\Delta\mu_t'(b',\widetilde{b}),\widetilde{\sigma}_t^2}(\theta)/N_{0,\widetilde{\sigma}_t^2}(\theta) + (1-sq) \right)^{\alpha+m-j-1}}$$

$$\times \left| N_{\Delta\mu_t(b',\widetilde{b}),\widetilde{\sigma}_t^2}(\theta)/N_{0,\widetilde{\sigma}_t^2}(\theta) - 1 \right|^j \left| N_{\Delta\mu_t'(b',\widetilde{b}),\widetilde{\sigma}_t^2}(\theta)/N_{0,\widetilde{\sigma}_t^2}(\theta) - 1 \right|^{m-j} N_{0,\widetilde{\sigma}_t^2}(\theta)d\theta ds$$

$$\leq \frac{q^m}{m!} \sum_{j=0}^{m} (1-q)^{-(\alpha+m-j-1)} \binom{m}{j} \left( \prod_{\ell=0}^{j-1}|\alpha-\ell| \right) \left( \prod_{\ell=0}^{m-j-1}(\alpha+\ell-1) \right) K_{\alpha,m,j}(q) ,$$

$$K_{\alpha,m,j}(q) := m \int_0^1 (1-s)^{m-1} \int \left( sq N_{\Delta\mu_t(b',\widetilde{b}),\widetilde{\sigma}_t^2}(\theta)/N_{0,\widetilde{\sigma}_t^2}(\theta) + (1-sq) \right)^{\alpha-j}$$

$$\times \left| N_{\Delta\mu_t(b',\widetilde{b}),\widetilde{\sigma}_t^2}(\theta)/N_{0,\widetilde{\sigma}_t^2}(\theta) - 1 \right|^j \left| N_{\Delta\mu_t'(b',\widetilde{b}),\widetilde{\sigma}_t^2}(\theta)/N_{0,\widetilde{\sigma}_t^2}(\theta) - 1 \right|^{m-j} N_{0,\widetilde{\sigma}_t^2}(\theta)d\theta ds ,$$

where to obtain the final inequality we used the fact that $\alpha + m - j - 1 > 0$ and

$$sq N_{\Delta\mu_t'(b',\widetilde{b}),\widetilde{\sigma}_t^2}(\theta)/N_{0,\widetilde{\sigma}_t^2}(\theta) + (1-sq) \geq 1 - q , \tag{82}$$

to upper bound the reciprocal of the denominator.

To bound the integrals $K_{\alpha,m,j}$ we consider the following two cases.

**Case 1: $\alpha - j \leq 0$:**
In this case we use

$$\left(sqN_{\Delta\mu_t(b',\widetilde{b}),\widetilde{\sigma}_t^2}(\theta)/N_{0,\widetilde{\sigma}_t^2}(\theta) + (1-sq)\right)^{\alpha-j} \leq (1-q)^{\alpha-j} \tag{83}$$

to bound the first factor in the integrand. Then, using Lemma C.7 and Lemma C.4, we can compute

$$K_{\alpha,m,j}(q) \tag{84}$$

$$\leq (1-q)^{\alpha-j}\int \left|N_{\Delta\mu_t(b',\widetilde{b}),\widetilde{\sigma}_t^2}(\theta)/N_{0,\widetilde{\sigma}_t^2}(\theta) - 1\right|^j \left|N_{\Delta\mu_t'(b',\widetilde{b}),\widetilde{\sigma}_t^2}(\theta)/N_{0,\widetilde{\sigma}_t^2}(\theta) - 1\right|^{m-j} N_{0,\widetilde{\sigma}_t^2}(\theta)d\theta$$

$$\leq (1-q)^{\alpha-j}\widetilde{B}_{\sigma_t,m} \,.$$

**Case 2: $\alpha - j > 0$:**
In this case use the bound $x^{\alpha-j} \leq 1 + x^{\lceil\alpha\rceil-j}$ for all $x > 0$, followed by the binomial formula and Lemma C.7 to compute

$$K_{\alpha,m,j}(q) \leq m\int_0^1 (1-s)^{m-1}\int\left(1 + \left(sqN_{\Delta\mu_t(b',\widetilde{b}),\widetilde{\sigma}_t^2}(\theta)/N_{0,\widetilde{\sigma}_t^2}(\theta) + (1-sq)\right)^{\lceil\alpha\rceil-j}\right) \tag{85}$$

$$\times \left|N_{\Delta\mu_t(b',\widetilde{b}),\widetilde{\sigma}_t^2}(\theta)/N_{0,\widetilde{\sigma}_t^2}(\theta) - 1\right|^j \left|N_{\Delta\mu_t'(b',\widetilde{b}),\widetilde{\sigma}_t^2}(\theta)/N_{0,\widetilde{\sigma}_t^2}(\theta) - 1\right|^{m-j} N_{0,\widetilde{\sigma}_t^2}(\theta)d\theta ds$$

$$= \int \left|N_{\Delta\mu_t(b',\widetilde{b}),\widetilde{\sigma}_t^2}(\theta)/N_{0,\widetilde{\sigma}_t^2}(\theta) - 1\right|^j \left|N_{\Delta\mu_t'(b',\widetilde{b}),\widetilde{\sigma}_t^2}(\theta)/N_{0,\widetilde{\sigma}_t^2}(\theta) - 1\right|^{m-j} N_{0,\widetilde{\sigma}_t^2}(\theta)d\theta$$

$$+ m\sum_{\ell=0}^{\lceil\alpha\rceil-j} q^\ell\binom{\lceil\alpha\rceil-j}{\ell}\int_0^1 (1-s)^{m-1}s^\ell\int\left(N_{\Delta\mu_t(b',\widetilde{b}),\widetilde{\sigma}_t^2}(\theta)/N_{0,\widetilde{\sigma}_t^2}(\theta) - 1\right)^\ell$$

$$\times \left|N_{\Delta\mu_t(b',\widetilde{b}),\widetilde{\sigma}_t^2}(\theta)/N_{0,\widetilde{\sigma}_t^2}(\theta) - 1\right|^j \left|N_{\Delta\mu_t'(b',\widetilde{b}),\widetilde{\sigma}_t^2}(\theta)/N_{0,\widetilde{\sigma}_t^2}(\theta) - 1\right|^{m-j} N_{0,\widetilde{\sigma}_t^2}(\theta)d\theta ds$$

$$\leq \widetilde{B}_{\sigma_t,m} + \sum_{\ell=0}^{\lceil\alpha\rceil-j} q^\ell\frac{(\lceil\alpha\rceil-j)!m!}{(\lceil\alpha\rceil-j-\ell)!(m+\ell)!}\widetilde{B}_{\sigma_t,\ell+m} \,.$$

Putting the above two cases together we arrive at the remainder bound

$$|E_{\alpha,m}(q)| \leq q^m\int_0^1 \frac{(1-s)^{m-1}}{(m-1)!}\left|\frac{d^m}{dq^m}F_\alpha(sq)\right|ds \tag{86}$$

$$\leq \frac{q^m}{m!}\sum_{j=0}^m (1-q)^{-(\alpha+m-j-1)}\binom{m}{j}\left(\prod_{\ell=0}^{j-1}|\alpha-\ell|\right)\left(\prod_{\ell=0}^{m-j-1}(\alpha+\ell-1)\right)$$

$$\times \begin{cases} (1-q)^{\alpha-j}\widetilde{B}_{\sigma_t,m} & \text{if } \alpha - j \leq 0 \\ \widetilde{B}_{\sigma_t,m} + \sum_{\ell=0}^{\lceil\alpha\rceil-j} q^\ell\frac{(\lceil\alpha\rceil-j)!m!}{(\lceil\alpha\rceil-j-\ell)!(m+\ell)!}\widetilde{B}_{\sigma_t,\ell+m} & \text{if } \alpha - j > 0 \end{cases}$$

$$:= \widetilde{E}_{\alpha,\sigma_t,m}(q) \,. \tag{87}$$

Combining the above results, we obtain the following $\text{FS}_{\text{woR}}$-RDP bounds under replace-one adjacency:

$$\sup_{\theta_t} D_\alpha(p(\theta_{t+1}|\theta_t, D)\|p_t(\theta_{t+1}|\theta_t), D') \tag{88}$$

$$\leq \frac{1}{\alpha-1}\log\left[1 + q^2\alpha(\alpha-1)\left(e^{4/\sigma_t^2} - e^{2/\sigma_t^2}\right) + \sum_{k=3}^{m-1}\frac{q^k}{k!}\widetilde{F}_{\alpha,\sigma_t,k} + \widetilde{E}_{\alpha,\sigma_t,m}(q)\right],$$

where $m \geq 3$ is an integer, $\widetilde{F}_{\alpha,\sigma_t,k}$ is given by (80), and $\widetilde{E}_{\alpha,\sigma_t,m}(q)$ is given by (87); both of these quantities are expressed in terms of $M$ and $\widetilde{B}$, as given by (41) and (44) respectively. This completes the proof of Theorem 3.4.

*Remark* C.8. The same $M$'s appear many times during in the formulas for $\widetilde{F}_{\alpha,\sigma_t,k}$ and $\widetilde{E}_{\alpha,\sigma_t,m}(q)$. Therefore, in practice, we compute all required values once and recall them as needed.

### C.1 Comparison with Poisson Subsampling under Replace-one Adjacency

In this appendix we provide a precise comparison between RDP for fixed-size and Poisson subsampling under replace-one adjacency, complimenting the intuitive discussion in Section 4.1.

Here we let $p_t^{\text{Poisson}}(\theta_{t+1}|\theta_t, D)$ denote the transition probabilities for one step of DP-SGD with Poisson subsampling. By Mirroring the analysis in Section 2 of Mironov et al. (2019), but now under the assumption that $D$ and $D'$ have the same size and differ in only their last elements, we arrive at the one-step Rényi divergence bound

$$D_\alpha \left( p_t^{\text{Poisson}}(\theta_{t+1}|\theta_t, D) \| p_t^{\text{Poisson}}(\theta_{t+1}|\theta_t, D') \right) \tag{89}$$
$$\leq \max_T D_\alpha \left( q N_{\Delta\mu_t(T), \widetilde{\sigma}_t^2} + (1-q) N_{0, \widetilde{\sigma}_t^2} \| q N_{\Delta\mu_t'(T), \widetilde{\sigma}_t^2} + (1-q) N_{0, \widetilde{\sigma}_t^2} \right),$$
$$\Delta\mu_t(T) := \mu_t(\theta_t, T \cup \{|D|-1\}, D) - \mu_t(\theta_t, T, D),$$
$$\Delta\mu_t'(T) := \mu_t(\theta_t, T \cup \{|D|-1\}, D') - \mu_t(\theta_t, T, D),$$

where $T$ denotes the collection of indexes from $\{0, ..., |D|-2\}$ that were randomly selected during Poisson sampling.

This is similar to the result (56) for $\text{FS}_{\text{woR}}$-subsampling, except here the means satisfy the constraints

$$\|\Delta\mu_t(T)\| \leq r_t/2, \quad \|\Delta\mu_t'(T)\| \leq r_t/2, \quad \|\Delta\mu_t(T) - \Delta\mu_t'(T)\| \leq r_t, \tag{90}$$

and these bounds are achieved when the differing elements from $D$ and $D'$ are anti-parallel and saturate the clipping threshold $C$. As we will show, this difference only contributes at higher order.

We now mirror the analysis from earlier in this section, taking the new constraints (90) into account. We again have a Taylor expansion of the form

$$D_\alpha \left( q N_{\Delta\mu_t(T), \widetilde{\sigma}_t^2} + (1-q) N_{0, \widetilde{\sigma}_t^2} \| q N_{\Delta\mu_t'(T), \widetilde{\sigma}_t^2} + (1-q) N_{0, \widetilde{\sigma}_t^2} \right) \tag{91}$$
$$= \frac{1}{\alpha - 1} \log \left[ 1 + \sum_{k=2}^{m-1} \frac{q^k}{k!} \frac{d^k}{dq^k} \widehat{F}_\alpha(0) + \widehat{E}_{\alpha,m}(q) \right]. \tag{92}$$

The coefficient at $k = 2$ can be computed as in (62), which gives

$$\frac{d^2}{dq^2} \widehat{F}_\alpha(0) \tag{93}$$
$$= \alpha(\alpha-1) \left( \exp\left( \frac{\|\Delta\mu_t(T)\|^2}{\widetilde{\sigma}_t^2} \right) + \exp\left( \frac{\|\Delta\mu_t'(T)\|^2}{\widetilde{\sigma}_t^2} \right) - 2\exp\left( \frac{\Delta\mu_t(T) \cdot \Delta\mu_t'(T)}{\widetilde{\sigma}_t^2} \right) \right)$$
$$\leq 2\alpha(\alpha-1) \left( e^{1/\sigma_t^2} - e^{-1/\sigma_t^2} \right).$$

This bound is optimal in that equality is achieved when the distinct elements in $D$ and $D'$ are anti-parallel and saturate the clipping bound. To leading order in $1/\sigma_t^2$ we therefore have

$$\frac{d^2}{dq^2} \widehat{F}_\alpha(0) \leq 4\alpha(\alpha-1)/\sigma_t^2 + O(1/\sigma_t^4), \tag{94}$$

which agrees with the leading order behavior of the corresponding term in the $\text{FS}_{\text{woR}}$-subsampling case (66). As the second order coefficient in the Taylor expansion controls the dominant behavior of the RDP bound, we can therefore conclude that Poisson and $\text{FS}_{\text{woR}}$ subsampling give the same DP bounds under replace-one adjacency to leading order.

Our method for bounding the higher order terms, as in (80), and remainder, as in (86), only uses bounds on the norms of $\Delta\mu'$ and $\mu'$; it does not incorporate the bound that couples these two quantities. Therefore the corresponding bounds on the higher order terms for Poisson subsampling are obtained by simply replacing $r_t$ with $r_t/2$ in our prior calculations; this is equivalent to replacing $\sigma_t$ with $2\sigma_t$ in (80) and (86). Therefore we arrive at the following non-asymptotic bound for Poisson subsampling under replace-one adjacency.

**Theorem C.9** (Poisson-RDP Upper Bounds for Replace-one Adjacency). *Let $D \simeq_{r\text{-}o} D'$ be adjacent datasets. Assuming $q := |B|/|D| < 1$, for any integer $m \geq 3$ we have*

$$\sup_{\theta_t} D_\alpha(p_t^{Poisson}(\theta_{t+1}|\theta_t, D) \| p_t^{Poisson}(\theta_{t+1}|\theta_t, D')) \tag{95}$$

$$\leq \frac{1}{\alpha - 1} \log \left[ 1 + q^2 \alpha(\alpha - 1) \left( e^{1/\sigma_t^2} - e^{-1/\sigma_t^2} \right) + \sum_{k=3}^{m-1} \frac{q^k}{k!} \widetilde{F}_{\alpha, 2\sigma_t, k} + \widetilde{E}_{\alpha, 2\sigma_t, m}(q) \right],$$

*where $\widetilde{F}_{\alpha, 2\sigma_t, k}$ and $\widetilde{E}_{\alpha, 2\sigma_t, m}(q)$ are computed via (80) and (87) respectively (with $\sigma_t$ replaced by $2\sigma_t$).*

While our FS$_{\text{woR}}$ and Poisson-subsampled RDP bounds agree to leading order, the effect of the higher order terms leads to Poisson subsampling having a slight privacy advantage over FS$_{\text{woR}}$-subsampling in practice, especially when larger $\alpha$'s are required. This proven in Figure 7, where we see that the upper bound on Poisson-subsampled RDP (dashed line) is slightly below the theoretical lower bound for FS$_{\text{woR}}$-subsampled RDP derived in Wang et al. (2019) (circles) while our FS$_{\text{woR}}$-RDP bound from Theorem 3.4 (solid line) is slightly above the theoretical lower bound. All three are significantly below the upper bound from Wang et al. (2019) (dot-dashed line). We emphasize that the Poisson result does not contradict the lower bound from Wang et al. (2019), which only applies to FS$_{\text{woR}}$-subsampled RDP.

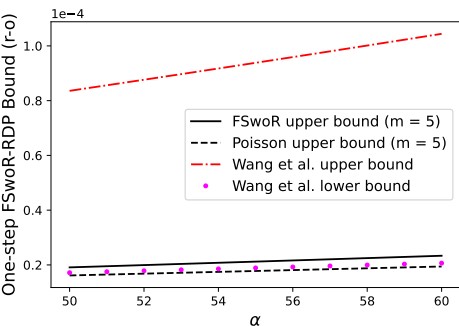

Figure 7: Comparison of our FS$_{\text{woR}}$-RDP and Poisson-RDP upper bounds under replace-one adjacency, Theorems 3.4 and C.9 respectively, with the upper and lower bounds on FS$_{\text{woR}}$-RDP from Wang et al. (2019). We used $\sigma_t = 6$, $|B| = 120$, and $|D| = 50,000$.

## D  Rényi DP-SGD Bounds for Fixed-size Subsampling with Replacement

In this Appendix we provide a detailed derivation of Rényi DP-SGD bounds when using fixed-size subsampling with replacement (FSwR). In addition to the lower bound stated in Theorem 3.7, we also derive an upper bound.

As in case without replacement in Appendix A, a key step is a decomposition of the sampling distribution into a baseline (where $D$ and $D'$ agree) and a perturbation where they disagree. The derivation for subsampling with replacement is a non-trivial generalization of sampling without replacement, with Lemma D.1 below requiring a several key new ingredients, as compared to Lemma A.1.

First let $D'$ be obtained from $D$ by removing one element. Without loss of generality, assume it is the last (i.e., at index $|D| - 1$) and call that element $d$. Let $B' = (B'_1, ..., B'_{|B|})$ where the components are iid Uniform($\{0, ..., |D|' - 1\}$) random variables, so that $B'$ uniformly samples from $\{0, ..., |D|' - 1\}$ with replacement. Now let $N \sim \text{Binomial}(|B|, |D|^{-1})$ and $\Pi$ be a uniformly random permutation of $(1, ..., |B|)$. Assume that $B'$, $N$, and $\Pi$ are independent. In the following lemma we show how sampling from $D$ with replacement can be obtained from $B'$, $N$, and $\Pi$. In short, $N$ will determine the number of entries of $B'$ are replaced by $|D| - 1$ and $\Pi$ will determine the indices of the entries that are to be replaced.

**Lemma D.1.** *For $i = 1, ..., |B|$ define*

$$B_i = \begin{cases} |D| - 1 & \text{if } \Pi(j) = i \text{ for some } j \leq N \\ B_i' & \text{otherwise} \end{cases} := \Phi_i(B', N, \Pi). \tag{96}$$

*Then the $B_i$ are iid $\mathrm{Uniform}(\{0, ..., |D| - 1\})$ random variables, i.e., $B := (B_1, ..., B_{|B|})$ uniformly samples from $\{0, ..., |D| - 1\}$ with replacement.*

*Proof.* Let $b \in \{0, ..., |D| - 1\}^{|B|}$ and define $n = |\{i : b_i = |D| - 1\}|$. Let $i_1, ..., i_n$ be the unique indices with $b_{i_j} = |D| - 1$. If $B = b$ then we must have $N = n$, therefore if we let $S_n$ denote the set of permutations on $\{1, ..., n\}$ we have

$$P(B = b) = P(B = b, N = n, \{\Pi(1), ..., \Pi(n)\} = \{i_1, ..., i_n\}) \tag{97}$$

$$= \sum_{\tau \in S_n} P\left(B_i = b_i \text{ for } i \notin \{i_1, ..., i_n\}, N = n, (\Pi(1), ..., \Pi(n)) = (i_{\tau(1)}, ..., i_{\tau(n)})\right)$$

$$= \sum_{\tau \in S_n} P\left(B_i' = b_i \text{ for } i \notin \{i_1, ..., i_n\}, N = n, (\Pi(1), ..., \Pi(n)) = (i_{\tau(1)}, ..., i_{\tau(n)})\right).$$

Using independence and that $|D'| = |D| - 1$ and $|S_n| = n!$ we can then compute

$$P(B = b) \tag{98}$$

$$= \sum_{\tau \in S_n} \left(\prod_{i \notin \{i_1, ..., i_n\}} P(B_i' = b_i)\right) P(N = n) P\left((\Pi(1), ..., \Pi(n)) = (i_{\tau(1)}, ..., i_{\tau(n)})\right)$$

$$= \sum_{\tau \in S_n} |D'|^{-(|B| - n)} \binom{|B|}{n} |D|^{-n} (1 - |D|^{-1})^{|B| - n} \frac{(|B| - n)!}{|B|!}$$

$$= n! |D|^{-|B|} \binom{|B|}{n} \frac{(|B| - n)!}{|B|!}$$

$$= |D|^{-|B|}.$$

This holds for all $b$, therefore we can conclude that the distribution of $B$ is the uniform distribution on $\{0, ..., |D| - 1\}^{|B|}$. All the remaining claims then immediately follow from this fact. $\square$

Using Lemma D.1 we can decompose the transition probabilities for one step of DP-SGD with $\mathrm{FS_{wR}}$-subsampling as follows

$$p_t(d\theta_{t+1} | \theta_t, D') = \int N_{\mu_t(\theta, b', D'), \widetilde{\sigma}_t^2}(d\theta_{t+1}) P_{B', \Pi}(db' d\pi), \tag{99}$$

$$p_t(d\theta_{t+1} | \theta_t, D) = \int N_{\mu_t(\theta_t, b, D), \widetilde{\sigma}_t^2}(d\theta_{t+1}) P_B(db) \tag{100}$$

$$= \int \sum_{n=0}^{|B|} \binom{|B|}{n} |D|^{-n} (1 - |D|^{-1})^{|B| - n} N_{\mu_t(\theta_t, \Phi(b', n, \pi), D), \widetilde{\sigma}_t^2}(d\theta_{t+1}) P_{B', \Pi}(db' d\pi).$$

Noting that $\mu_t(\theta, b', D') = \mu_t(\theta, b', D)$, as $D'$ differs from $D$ only by removal of the last element, we now use quasiconvexity to compute

$$D_\alpha(p_t(d\theta_{t+1} | \theta_t, D) \| p_t(d\theta_{t+1} | \theta_t, D')) \tag{101}$$

$$\leq \max_{b', \pi} D_\alpha \left(\sum_{n=0}^{|B|} \binom{|B|}{n} |D|^{-n} (1 - |D|^{-1})^{|B| - n} N_{\mu_t(\theta_t, \Phi(b', n, \pi), D), \widetilde{\sigma}_t^2} \left\| N_{\mu_t(\theta_t, b', D), \widetilde{\sigma}_t^2}\right)\right).$$

*Remark D.2.* As in Remarks A.2 and C.1, Eq. (101) remains true when the Rényi divergences $D_\alpha$ are replaced by any other divergence $\mathcal{D}$ that is quasiconvex in both of its arguments, e.g., the hockey-stick divergences. This fact is useful for other differential privacy paradigms.

## D.1 Upper Bound on $FS_{wR}$-RDP

The next stages in the calculation are more complex than in the case of sampling without replacement, as the first argument of the above Rényi divergence is a mixture of $|B| + 1$ Gaussians, as opposed to simply two as in the case without replacement. This is because the binomial random variable $N$ in (D.1) is playing the role that the Bernoulli random variable $J$ did in Lemma A.1.

Start by rewriting Eq. (101) as

$$D_\alpha(p_t(d\theta_{t+1}|\theta_t, D)\|p_t(d\theta_{t+1}|\theta_t, D')) \leq \max_{b', \pi} D_\alpha \left( \sum_{n=0}^{|B|} a_n N_{\Delta\mu_n, \widetilde{\sigma}_t^2} \middle\| N_{0, \widetilde{\sigma}_t^2} \right) ,$$

$$a_n := \binom{|B|}{n} |D|^{-n} (1 - |D|^{-1})^{|B|-n} , \quad \Delta\mu_n := \mu_t(\theta_t, \Phi(b', n, \pi), D) - \mu_t(\theta_t, b', D). \quad (102)$$

Let $1 < K \leq |B|$, choose $\widetilde{q}$ satisfying

$$1 \geq \widetilde{q} \geq \left( 1 + \frac{a_0}{\sum_{n=1}^K a_n} \right)^{-1} \quad (103)$$

and decompose the mixture of distributions as follows

$$\sum_{n=0}^{|B|} a_n N_{\Delta\mu_n, \widetilde{\sigma}_t^2} = a_0 N_{0, \widetilde{\sigma}_t^2} + \sum_{n=1}^K a_n N_{\Delta\mu_n, \widetilde{\sigma}_t^2} + \sum_{n=K+1}^{|B|} a_n N_{\Delta\mu_n, \widetilde{\sigma}_t^2} \quad (104)$$

$$= \widetilde{a}_0 N_{0, \widetilde{\sigma}_t^2} + \sum_{n=1}^K \widetilde{a}_n \left( \widetilde{q} N_{\Delta\mu_n, \widetilde{\sigma}_t^2} + (1 - \widetilde{q}) N_{0, \widetilde{\sigma}_t^2} \right) + \sum_{n=K+1}^{|B|} a_n N_{\Delta\mu_n, \widetilde{\sigma}_t^2} ,$$

$$\widetilde{a}_0 := a_0 - (1/\widetilde{q} - 1) \sum_{n=1}^K a_n , \quad \widetilde{a}_n := a_n / \widetilde{q} \text{ for } n \geq 1 .$$

The assumption (103) implies $\widetilde{a}_n \in [0, 1]$ and $\widetilde{a}_0 + \sum_{n=1}^K \widetilde{a}_n + \sum_{n=K+1}^{|B|} a_n = 1$ and therefore we can use convexity to bound (102) as follows:

$$D_\alpha(p_t(d\theta_{t+1}|\theta_t, D)\|p_t(d\theta_{t+1}|\theta_t, D')) \quad (105)$$

$$\leq \max_{b', \pi} \frac{1}{\alpha - 1} \log \left[ \widetilde{a}_0 + \sum_{n=1}^K \widetilde{a}_n \int \left( \widetilde{q} \frac{N_{\Delta\mu_n, \widetilde{\sigma}_t^2}(\theta)}{N_{0, \widetilde{\sigma}_t^2}(\theta)} + (1 - \widetilde{q}) \right)^\alpha N_{0, \widetilde{\sigma}_t^2}(\theta) d\theta \right.$$

$$\left. + \sum_{n=K+1}^{|B|} a_n \int \left( \frac{N_{\Delta\mu_n, \widetilde{\sigma}_t^2}(\theta)}{N_{0, \widetilde{\sigma}_t^2}(\theta)} \right)^\alpha N_{0, \widetilde{\sigma}_t^2}(\theta) d\theta \right] ,$$

$$\leq \frac{1}{\alpha - 1} \log \left[ \widetilde{a}_0 + \sum_{n=1}^K \widetilde{a}_n \int \left( \widetilde{q} \frac{N_{1, \sigma_t^2/(4n^2)}(\theta)}{N_{0, \sigma_t^2/(4n^2)}(\theta)} + (1 - \widetilde{q}) \right)^\alpha N_{0, \sigma_t^2/(4n^2)}(\theta) d\theta \right.$$

$$\left. + \sum_{n=K+1}^{|B|} a_n e^{2\alpha(\alpha-1)n^2/\sigma_t^2} \right]$$

$$= \frac{1}{\alpha - 1} \log \left[ \widetilde{a}_0 + \sum_{n=1}^K \widetilde{a}_n H_{\alpha, \sigma_t/n}(\widetilde{q}) + \sum_{n=K+1}^{|B|} a_n e^{2\alpha(\alpha-1)n^2/\sigma_t^2} \right] ,$$

where $H$ was defined in Eq. (8) and can be bounded using the techniques from Appendix B.

Now suppose $D'$ is obtained from $D$ by adding one new element. By reversing the role of $D$ and $D'$ in Lemma D.1 and then mirroring the above derivation while also employing Theorem 5 in Mironov

et al. (2019) we arrive at

$$D_\alpha(p_t(d\theta_{t+1}|\theta_t, D)\|p_t(d\theta_{t+1}|\theta_t, D')) \tag{106}$$

$$\leq \frac{1}{\alpha-1} \log \left[ \widetilde{a}_0' + \sum_{n=1}^{K} \widetilde{a}_n' H_{\alpha,\sigma_t/n}(\widetilde{q}) + \sum_{n=K+1}^{|B|} a_n' e^{2\alpha(\alpha-1)n^2/\sigma_t^2} \right],$$

$$a_n' := \binom{|B|}{n} |D'|^{-n} (1-|D'|^{-1})^{|B|-n},$$

$$\widetilde{a}_0' := a_0' - (1/\widetilde{q} - 1) \sum_{n=1}^{K} a_n', \quad \widetilde{a}_n' := a_n'/\widetilde{q} \text{ for } n \geq 1,$$

whenever $\widetilde{q}$ satisfies (103). Note that (103) also implies

$$q \geq \left( 1 + \frac{a_0'}{\sum_{n=1}^{K} a_n'} \right)^{-1} \tag{107}$$

due to the fact that $a_0$ is increasing in $|D|$ and $a_n$ is decreasing in $|D|$ for $n \geq 1$. These properties also imply

$$\widetilde{a}_0' + \sum_{n=1}^{K} \widetilde{a}_n' H_{\alpha,\sigma_t/n}(\widetilde{q}) + \sum_{n=K+1}^{|B|} a_n' e^{2\alpha(\alpha-1)n^2/\sigma_t^2} \tag{108}$$

$$\leq \widetilde{a}_0 + \sum_{n=1}^{K} \widetilde{a}_n H_{\alpha,\sigma_t/n}(\widetilde{q}) + \sum_{n=K+1}^{|B|} a_n e^{2\alpha(\alpha-1)n^2/\sigma_t^2},$$

and therefore we can conclude that

$$D_\alpha(p_t(d\theta_{t+1}|\theta_t, D)\|p_t(d\theta_{t+1}|\theta_t, D')) \tag{109}$$

$$\leq \frac{1}{\alpha-1} \log \left[ \widetilde{a}_0 + \sum_{n=1}^{K} \widetilde{a}_n H_{\alpha,\sigma_t/n}(\widetilde{q}) + \sum_{n=K+1}^{|B|} a_n e^{2\alpha(\alpha-1)n^2/\sigma_t^2} \right]$$

whenever $D' \simeq_{a/r} D$ and any choice of $\widetilde{q}$ that satisfies

$$1 \geq \widetilde{q} \geq \left( 1 + \frac{a_0}{\sum_{n=1}^{K} a_n} \right)^{-1}. \tag{110}$$

Note that if one chooses $\widetilde{q} = \left( 1 + \frac{a_0}{\sum_{n=1}^{K} a_n} \right)^{-1}$ then $\widetilde{a}_0 = 0$. If one also takes $K = |B|$ then $\widetilde{q} = 1 - (1 - |D|^{-1})^{|B|}$. This is the version of the result we present above in Theorem 3.6.

## D.2 Lower Bound on FS$_{wR}$-RDP

Here we derive a lower bound on the one-step RDP of SGD with fixed-size subsampling with replacement. To do this, suppose that all entries of $D'$ are the same, equal to $d'$, and that the extra entry $d$ in $D$ satisfies $d \neq d'$. Moreover, suppose there exists $\theta_t$ such that the clipped gradients at $d$ and $d'$ are anti-parallel with both having norm equal to the clipping bound, $C$. In this case, $\mu_t(\theta_t, b', D)$ and $\mu_t(\theta_t, \Phi(b', n, \pi), D)$ are independent of both $\pi$ and $b'$ and

$$\|\mu_t(\theta_t, b', D) - \mu(\theta_t, \Phi(b', n, \pi), D)\| = nr_t. \tag{111}$$

This also implies that (101) is an equality (and the argument of the divergence is independent of $b'$ and $\pi$). By changing variables and using (111) we can therefore write

$$D_\alpha(p_t(d\theta_{t+1}|\theta_t, D)\|p_t(d\theta_{t+1}|\theta_t, D')) \tag{112}$$

$$= \max_{b',\pi} D_\alpha\left(\sum_{n=0}^{|B|}\binom{|B|}{n}|D|^{-n}(1-|D|^{-1})^{|B|-n}N_{\|\mu_t(\theta_t,\Phi(b',n,\pi),D)-\mu_t(\theta_t,b',D)\|,\tilde\sigma_t^2}\Big\|N_{0,\tilde\sigma_t^2}\right)$$

$$= D_\alpha\left(\sum_{n=0}^{|B|}\binom{|B|}{n}|D|^{-n}(1-|D|^{-1})^{|B|-n}N_{nr_t,\tilde\sigma_t^2}\Big\|N_{0,\tilde\sigma_t^2}\right)$$

$$= D_\alpha\left(\sum_{n=0}^{|B|}\binom{|B|}{n}|D|^{-n}(1-|D|^{-1})^{|B|-n}N_{n,\sigma_t^2/4}\Big\|N_{0,\sigma_t^2/4}\right).$$

This gives the one-step RDP lower bound

$$\epsilon_t'(\alpha) \geq \frac{1}{\alpha-1}\log\left[\int\left(\sum_{n=0}^{|B|}\binom{|B|}{n}|D|^{-n}(1-|D|^{-1})^{|B|-n}\frac{N_{n,\sigma_t^2/4}(\theta)}{N_{0,\sigma_t^2/4}(\theta)}\right)^\alpha N_{0,\sigma_t^2/4}(\theta)d\theta\right]. \tag{113}$$

The integral over $\theta$ can be computed as follows. First note that summation in (113) is the average of $N_{n,\sigma_t^2/4}(\theta)/N_{0,\sigma_t^2/4}(\theta)$ under the Binomial distribution $\mathrm{Binomial}(|B|,|D|^{-1})$. As a convenient shorthand we write this average as $\int ...dn$. With this notation, for integer $\alpha$ we can rewrite the argument of the logarithm in the lower bound (113) an iterated integral, then exchange the order of the integrals and evaluate the integral over $\theta$ using the formula for the MGF of a Gaussian as follows

$$\int\left(\sum_{n=0}^{|B|}\binom{|B|}{n}|D|^{-n}(1-|D|^{-1})^{|B|-n}\frac{N_{n,\sigma_t^2/4}(\theta)}{N_{0,\sigma_t^2/4}(\theta)}\right)^\alpha N_{0,\sigma_t^2/4}(\theta)d\theta \tag{114}$$

$$= \int\int...\int\prod_{i=1}^\alpha\frac{N_{n_i,\sigma_t^2/4}(\theta)}{N_{0,\sigma_t^2/4}(\theta)}dn_1...dn_\alpha N_{0,\sigma_t^2/4}(\theta)d\theta$$

$$= \int...\int\int e^{-\sum_i n_i^2/(\sigma^2/2)+\theta\sum_i n_i/(\sigma^2/4)}N_{0,\sigma_t^2/4}(\theta)d\theta dn_1...dn_\alpha$$

$$= \int...\int e^{\frac{4}{\sigma_t^2}\sum_{i<j}n_i n_j}dn_1...dn_\alpha. \tag{115}$$

Therefore we arrive at the lower bound

$$\epsilon_t'(\alpha) \geq \frac{1}{\alpha-1}\log\left[\int...\int e^{\frac{4}{\sigma_t^2}\sum_{i<j}n_i n_j}dn_1...dn_\alpha\right]. \tag{116}$$

Recalling the definition of the $dn_i$'s, this completes the proof of Theorem 3.7.

The lower bound (115) behaves somewhat differently from the one derived for subsampling without replacement in Wang et al. (2019) and restated in Appendix E.2. In particular, note that the explicit $|B|$ and $|D|$ dependence in the lower bound (113) cannot be removed simply by reparameterizing the expression in terms of $q = |B|/|D|$. In contrast, the $\mathrm{FS_{woR}}$-RDP lower bound in Appendix E.2 depends on $|B|$ and $|D|$ only through $q$. This fact can be further illustrated through a simple loosening of the bound (113), where we bound the sum below by the term at $n = |B|$ to obtain

$$\epsilon_t'(\alpha) \geq \frac{1}{\alpha-1}\log\left[|D|^{-\alpha|B|}\int\left(\frac{N_{|B|,\sigma_t^2/4}(\theta)}{N_{0,\sigma_t^2/4}(\theta)}\right)^\alpha N_{0,\sigma_t^2/4}(\theta)d\theta\right] \tag{117}$$

$$= \frac{\alpha|B|}{\alpha-1}\left(2|B|(\alpha-1)/\sigma_t^2 - \log|B| - \log(1/q)\right).$$

For fixed ratio $q = |B|/|D|$, the right-hand-side of (117) approaches $\infty$ as $|B| \to \infty$. Therefore we can conclude that $\epsilon_t'(\alpha)$ has nontrivial $|B|$ dependence, even for fixed $q$ as claimed. This accounts for

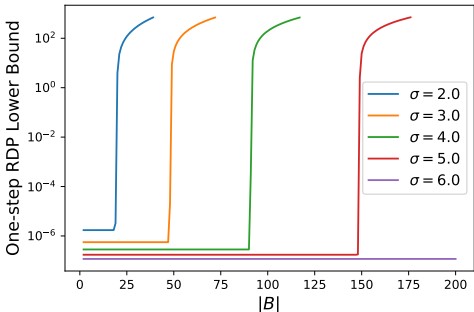

Figure 8: $FS_{wR}$-RDP lower bounds as a function of $|B|$, with $\alpha = 2$ and $q = 0.001$.

some of the difference in behavior between fixed-size subsampling with and without replacement. This behavior is also illustrated numerically in Figure 8, where we show the exact value of the lower bound (113) as a function of $|B|$ for $\alpha = 2$ (this case is straightforward to compute via the binomial theorem and MGF of a Gaussian) and for fixed $q = 0.001$; we show results for several values of $\sigma$. Note that initially each curve starts out horizontal, mirroring the case of $FS_{woR}$-RDP, which has no $|B|$ dependence outside of $q$. However, for each $\sigma$ there is a threshold value of $|B|$ as which there is a "phase transition", i.e., after which the explicit $|B|$ dependence becomes important and the RDP value increases rapidly.

### D.2.1    Comptutable $FS_{wR}$-RDP Lower Bounds

When $\alpha$ is an integer the lower bound (116) can, in principle, be computed exactly by recalling that the integrals with respect to $dn_i$ are shorthand for the expectation with respect to the distribution $\mathrm{Binomial}(B, |D|^{-1})$. Therefore the lower bound (116) consists of $\alpha$ nested sums of $|B| + 1$ terms each. Clearly, when $\alpha$ is larger than 2 and $|B|$ is not small iterated summation this quickly becomes computationally impractical. The looser lower bound (117) is easy to compute, but is much too inaccurate for most purposes. Here, for integer $\alpha$, we will show how obtain lower bounds that are both practical and significantly more accurate.

Recall that the integrals with respect to the $dn$'s are actually finite sums, each over $|B| + 1$ elements, and so in principle the above expression can be computed exactly but it is still not practical. We next show how to obtain recursively defined lower bounds that are significantly less computationally expensive than the $O(|B|^{\alpha})$ computations that the more naive formula (116) requires. To begin, for any integer $k \geq 2$ define

$$F_k(c, d) = \int ... \int e^{c \sum_{i<j} n_i n_j + d \sum_i n_i} dn_k ... dn_1 . \tag{118}$$

For $k = 2$ we can reduce the computation to a single summation by using the formula for the MGF of the Binomial distribution:

$$F_2(c, d) = \sum_{n=0}^{|B|} \binom{|B|}{n} |D|^{-n} (1 - |D|^{-1})^{|B|-n} e^{dn} (1 - |D|^{-1} + |D|^{-1} e^{cn+d})^{|B|} . \tag{119}$$

For $k > 2$ we explicitly write out the integral over $n_k$ as a summation and then simplify to obtain

$$F_k(c,d) \tag{120}$$

$$= \int ... \int e^{c\sum_{j=2}^{k-1}\sum_{i=1}^{j-1} n_i n_j + d\sum_{i=1}^{k-1} n_i} \int e^{(c\sum_{i=1}^{k-1} n_i + d)n_k} dn_k dn_{k-1}...dn_1$$

$$= \sum_{n=0}^{|B|} \binom{|B|}{n} |D|^{-n}(1-|D|^{-1})^{|B|-n} \int ... \int e^{c\sum_{j=2}^{k-1}\sum_{i=1}^{j-1} n_i n_j + d\sum_{i=1}^{k-1} n_i} e^{(c\sum_{i=1}^{k-1} n_i + d)n} dn_{k-1}...dn_1$$

$$= \sum_{n=0}^{|B|} \binom{|B|}{n} |D|^{-n}(1-|D|^{-1})^{|B|-n} e^{dn} \int ... \int e^{c\sum_{j=2}^{k-1}\sum_{i=1}^{j-1} n_i n_j + (d+cn)\sum_{i=1}^{k-1} n_i} dn_{k-1}...dn_1$$

$$= \sum_{n=0}^{|B|} \binom{|B|}{n} |D|^{-n}(1-|D|^{-1})^{|B|-n} e^{dn} F_{k-1}(c, d+cn).$$

As stated, the recursive formula provided by (119) and (120) is still too computationally expensive due to each recursive call requiring the summations over $|B| + 1$ elements. However, noting that terms in the summations are all positive, we can obtain a practical lower bound by retaining only a small number of terms. More specifically, given choices of $T_k \subset \{0, ..., |B|\}$ for each $k$, if we recursively define

$$F_{T,k}(c,d) := \sum_{n \in T_k} \binom{|B|}{n} |D|^{-n}(1-|D|^{-1})^{|B|-n} e^{dn} F_{k-1}(c, d+cn, T_{k-1}) \tag{121}$$

with

$$F_{T,2}(c,d) := \sum_{n \in T_2} \binom{|B|}{n} |D|^{-n}(1-|D|^{-1})^{|B|-n} e^{dn}(1-|D|^{-1}+|D|^{-1}e^{cn+d})^{|B|}. \tag{122}$$

Then, by induction, we have $F_k(c,d) \geq F_{T,k}(c,d)$ for all $k, T, c, d$ and therefore

$$\epsilon'_t(\alpha) \geq \frac{1}{\alpha-1} F_{T,\alpha}(4\sigma_t^2, 0) \tag{123}$$

for all $\alpha$ and all choices of $T_k$, $k = 2, ..., \alpha$. By appropriately choosing the number of terms $T_k$ for each $k$ one trades between accuracy and computational cost. In particular, if each $|T_k| = 1$ then the computation of (123) only requires $O(\alpha)$ computations. If $|T_k| = \ell > 2$ for all $k$ then the computation of (123) required $O(\ell^{\alpha-1})$ computations. In contrast, using the exact recursive formula (120) requires $O(|B|^{\alpha-1})$ computations, which is intractable unless $\alpha$ or $|B|$ is sufficiently small. However, despite this improvement, we should note that computational difficulties persist for cases where the number of nontrivial terms is too large.

In our experiments in Figure 9 we find that that using $T_k = \{|B|\}$ for $k > 2$ and $T_2 = \{0, ..., |B|\}$ gives an accurate and computationally tractable lower bound. In the example in Figure 1 more terms were required; we show results obtained by using $T_k = \{0, 1, 2, |B|\}$ for $k > 2$ and $T_2 = \{0, ..., |B|\}$.

### D.3 Comparison of Upper and Lower FS$_{wR}$-RDP Bounds

Now we will compare our lower bound for FS$_{wR}$-RDP from Theorem 3.7 with the upper bound derived in Appendix D.1. In Figure 9 we show the FS$_{wR}$-RDP upper bound (solid line) and lower bound (circles) and, for comparison, we also show our bound on FS$_{woR}$-RDP (dashed line). Note that all three are very similar when $\alpha$ is close to 1; this is to be expected, due to them having the same leading order behavior in $q$, given by (19). However, as $\alpha$ increases the non-leading order behavior becomes more important and we observe that, after some threshold $\alpha$, both the FS$_{wR}$-RDP upper and lower bounds increase rapidly. A similar "phase transition" was observed earlier in Wang et al. (2019) for subsampling without replacement, though for these parameters it occurs at much higher values of $\alpha$. A key difference between these two cases is that, for FS$_{wR}$-RDP, the phase transition can be brought on by increasing $|B|$ while leaving $q = |B|/|D|$ fixed, as demonstrated in Appendix D.2; see (117) and Figure 8. In contrast, the lower bound for FS$_{woR}$-RDP from Wang et al. (2019) depends on $|B|$ and $|D|$ only through their ratio $q$. We anticipate that the gap between the upper and lower bounds on FS$_{wR}$-RDP in Figure 9 could be reduced by using a higher order expansion in $q$, but we leave that to future work. We also emphasize that the dashed FS$_{woR}$-RDP upper bounds is only for comparison purposes as it does not apply to the FS$_{wR}$ case considered in this section.

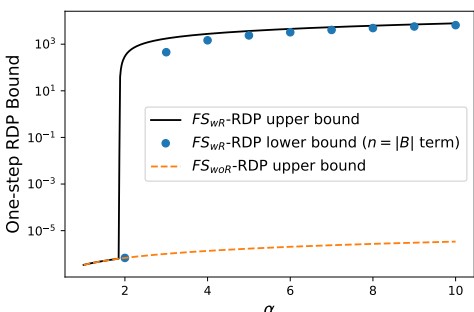

Figure 9: Comparison of our upper and lower bounds on $FS_{wR}$-RDP from Theorems 3.6 and 3.7. We used $\sigma_t = 6$, $|B| = 120$, and $|D| = 50,000$.

# E  Comparison with Wang et al. (2019)

In Wang et al. (2019), Rényi-DP enhancement of fixed-size subsampling was analyzed in a framework that applies to arbitrary mechanisms. In this appendix we specialize their results to fixed-size subsampling DP-SGD with Gaussian noise. This will facilitate comparison with our results. Specifically, we will show that our specialized method results in a factor tighter bounds by approximately a factor of 4.

## E.1  Upper Bound from Wang et al. (2019), Theorem 9

For integer $\alpha \geq 2$, the following the general fixed-size subsampling Rényi-DP enhancement bound (for one step) was derived in Wang et al. (2019):

$$\epsilon'_t(\alpha) \leq \frac{1}{\alpha - 1} K_t(\alpha) , \tag{124}$$

$$K_t(\alpha) := \log \left[ 1 + 2q^2\alpha(\alpha - 1)(e^{\epsilon_t(2)} - 1) + \sum_{j=3}^{\alpha} \frac{2q^j \prod_{\ell=0}^{j-1}(\alpha - \ell)}{j!} e^{(j-1)\epsilon_t(j)} \right]$$

(if $\alpha = 2$ then the empty summation is interpreted to be zero), where $\epsilon_t(j)$ denotes a Rényi-DP bound for the mechanism without random subsampling. For DP-SGD we have

$$D_j(N_{\mu',\widetilde{\sigma}_t^2} \| N_{\mu,\widetilde{\sigma}_t^2}) = \frac{j\|\mu' - \mu\|^2}{2\widetilde{\sigma}_t^2} \leq \frac{2jC^2\eta_t^2}{|B|^2\widetilde{\sigma}_t^2} = \frac{2j}{\sigma_t^2} \tag{125}$$

and therefore we can use

$$\epsilon_t(j) := \frac{2j}{\sigma_t^2} . \tag{126}$$

Note that (125) uses the same bound on the means that we use in our method, and therefore (126) will yield a fair comparison. Comparing the leading order behavior of (124) with the leading order behavior of a single step of our result in Theorem 3.3, we see that our bound is smaller by a factor of 4. In practice we find similar behavior even when including the higher order terms; see Figure 2.

For non-integer $\alpha \geq 2$ they use the following bound, which follows from convexity of $K_t(\alpha)$ in $\alpha$:

$$\epsilon'_t(\alpha) \leq \frac{1 - (\alpha - \lfloor\alpha\rfloor)}{\alpha - 1} K_t(\lfloor\alpha\rfloor) + \frac{\alpha - \lfloor\alpha\rfloor}{\alpha - 1} K_t(\lfloor\alpha\rfloor + 1) , \tag{127}$$

where $\lfloor x \rfloor$ denotes the floor of $x$. For $\alpha \in (1, 2)$ the convexity bound is

$$\epsilon'_t(\alpha) \leq \frac{\alpha - \lfloor\alpha\rfloor}{\alpha - 1} K_t(2) = K_t(2) \tag{128}$$

which simplifies to

$$\epsilon'_t(\alpha) \leq \epsilon'_t(2) , \quad \alpha \in (1, 2) . \tag{129}$$

## E.2 Lower Bound from (Wang et al., 2019), Proposition 11

A general lower-bound on fixed-size subsampling enhancement was also derived in (Wang et al., 2019); we repeat it here for convenience. For integer $\alpha \geq 2$ they show

$$\epsilon_t'(\alpha) \geq \frac{\alpha}{\alpha - 1} \log(1 - q) + \frac{1}{\alpha - 1} \log \left[ 1 + \alpha \frac{q}{1 - q} + \sum_{j=2}^{\alpha} \frac{\prod_{\ell=0}^{j-1} (\alpha - \ell)}{j!} \left( \frac{q}{1 - q} \right)^j e^{(j-1)\epsilon_t(j)} \right], \tag{130}$$

where $\epsilon_t(j)$ is again given by (126). In Figure 2 we demonstrate that our result is very close to this theoretical RDP lower bound.

## E.3 Comparison with Replace-One Adjacency Bounds in Theorem 3.4

The analysis in Wang et al. (2019) is done under the replace-one adjacency definition and is therefore directly compatible to our Theorem 3.4. In Section 3.4 we showed that our replace-one and our add/remove adjacency results, Theorem 3.4 and 3.1 respectively, lead to the same RDP bounds to leading order in $q$ and $1/\sigma_t^2$, including the factor of 4 improvement over Wang et al. (2019); this confirms that the improvement made by our approach is not reliant on the adjacency definition.

In this appendix we provide a conjecture as to the origin of this difference. The leading order behavior comes from the calculations (64)-(66), where we used the constraints on the means from (57):

$$\|\Delta \mu_t(b', \widetilde{b})\| \leq r_t \,, \quad \|\Delta \mu_t'(b', \widetilde{b})\| \leq r_t \,, \quad \|\Delta \mu_t(b', \widetilde{b}) - \Delta \mu_t'(b', \widetilde{b})\| \leq r_t \,. \tag{131}$$

However, if one doesn't use the constraint on the difference and only uses the first two upper bounds on the means then the best bound on the dot product becomes

$$|\mu_t(b', \widetilde{b}) \cdot \Delta \mu_t'(b', \widetilde{b})| \leq r_t^2 \tag{132}$$

and so the bound (66) is weakened to

$$\frac{d^2}{dq^2} F_\alpha(0) \leq 2\alpha(\alpha - 1) \left( e^{4/\sigma_t^2} - e^{-4/\sigma_t^2} \right) = 16\alpha(\alpha - 1)/\sigma_t^2 + O(1/\sigma_t^4) \,, \tag{133}$$

which at leading order is a factor of 4 larger than our result (66) which was derived using all three constraints in (57). The strategies we employ differ from Wang et al. (2019) in a way that makes it difficult to pinpoint exactly what the essential differences are, however the above calculation leads us to conjecture that the difference originates in the failure to account for the constraint that couples $\Delta \mu_t$ and $\Delta \mu_t'$. See also Remark C.6.

## F Variance of Fixed-size vs. Poisson Subsampling

In this appendix we provide detailed calculations of the variance comparison that was discussed in Section 4.2. Let $a_i \in \mathbb{R}$, $i = 1, ..., |D|$ (for application to DP-SGD, one can take $a_i = \nabla_\theta \mathcal{L}(d_i) \cdot v$, the gradient of the loss at the $i$'th training sample $d_i$ in the direction $v$) and let $|B| \in \{1, ..., |D|\}$. Let $J_i$, $i = 1, ..., |D|$ be iid Bernoulli($|B|/|D|$) random variables, $B$ be a random variable that uniformly selects a subset of size $|B|$ from $\{2, ..., |D|\}$, and $R_j$, $j = 1, ..., |B|$ be iid uniformly distributed over $\{1, ..., |D|\}$. We will compare the variance of the Poisson-subsampled average,

$$Z_P := \frac{1}{|B|} \sum_{i=1}^{|D|} a_i J_i \tag{134}$$

with that of fixed-size subsampling with replacement,

$$Z_R := \frac{1}{|B|} \sum_{j=1}^{|B|} a_{R_j} \,, \tag{135}$$

and with fixed-size subsampling without replacement,

$$Z_B := \frac{1}{|B|} \sum_{i \in B} a_i \,. \tag{136}$$

They all have the same expected value, equaling the average of the samples:

$$E[Z_P] = \frac{1}{|B|} \sum_{i=1}^{|D|} a_i E[J_i] = \frac{q}{|B|} \sum_{i=1}^{|D|} a_i = \frac{1}{|D|} \sum_{i=1}^{|D|} a_i := \overline{a} \,, \tag{137}$$

$$E[Z_R] = \frac{1}{|B|} \sum_{j=1}^{|B|} E[a_{R_j}] = \frac{1}{|B|} \sum_{j=1}^{|B|} E[\sum_i a_i 1_{R_j=i}] \tag{138}$$

$$= \frac{1}{|B|} \sum_{j=1}^{|B|} \sum_{i=1}^{|D|} a_i P(R_j = i) = \frac{1}{|D|} \sum_{i=1}^{|D|} a_i = \overline{a} \,,$$

and

$$E[Z_B] = \frac{1}{|B|} \sum_{i=1}^{|D|} a_i P(i \in B) = \frac{1}{|B|} \sum_{i=1}^{|D|} a_i \frac{\binom{|D|-1}{|B|-1}}{\binom{|D|}{|B|}} = \overline{a} \,. \tag{139}$$

The variances are

$$\mathrm{Var}[Z_P] = \frac{1}{|B|^2} \sum_{i,j=1}^{|D|} a_i a_j E[J_i J_j] - \overline{a}^2 \tag{140}$$

$$= \frac{1}{|B|^2} \sum_i a_i^2 E[J_i] + \frac{1}{|B|^2} \sum_{i \neq j} a_i a_j E[J_i]E[J_j] - \overline{a}^2$$

$$= \frac{1}{|B||D|} \sum_i a_i^2 + \frac{1}{|D|^2} \sum_{i \neq j} a_i a_j - \frac{1}{|D|^2} \sum_{i,j} a_i a_j$$

$$= \left( \frac{1}{|B|} - \frac{1}{|D|} \right) \frac{1}{|D|} \sum_{i=1}^{|D|} a_i^2 \,,$$

$$\mathrm{Var}[Z_R] = \frac{1}{|B|^2} \sum_{\ell,j=1}^{|B|} E[a_{R_j} a_{R_\ell}] - \overline{a}^2$$

$$= \frac{1}{|B|^2} \sum_{j=1}^{|B|} E[a_{R_j}^2] + \frac{1}{|B|^2} \sum_{\ell \neq j} E[a_{R_j}]E[a_{R_\ell}] - \overline{a}^2$$

$$= \frac{1}{|B|^2} \sum_{j=1}^{|B|} E[a_{R_j}^2] + \frac{1}{|B|^2} \sum_{\ell \neq j} \overline{a}^2 - \overline{a}^2$$

$$= \frac{1}{|B|^2} \sum_{j=1}^{|B|} E[\sum_i 1_{R_j=i} a_i^2] + \frac{1}{|B|^2}(|B|^2 - |B|)\overline{a}^2 - \overline{a}^2$$

$$= \sum_{i=1}^{|D|} a_i^2 \frac{1}{|B|^2} \sum_{j=1}^{|B|} E[1_{R_j=i}] + (1 - 1/|B|)\overline{a}^2 - \overline{a}^2$$

$$= \sum_{i=1}^{|D|} a_i^2 \frac{1}{|B|^2} \sum_{j=1}^{|B|} P(R_j = i) - \frac{1}{|B|}\overline{a}^2$$

$$= \frac{1}{|B||D|} \sum_{i=1}^{|D|} a_i^2 - \frac{1}{|B|}\overline{a}^2$$

$$= (1 - |B|/|D|)^{-1} \mathrm{Var}[Z_P] - \frac{1}{|B|}\overline{a}^2 \,,$$

and

$$\text{Var}[Z_B] = \frac{1}{|B|^2} \sum_{i,j=1}^{|D|} a_i a_j P(i \in B, j \in B) - \bar{a}^2 \tag{141}$$

$$= \frac{1}{|B|^2} \sum_i a_i^2 P(i \in B) + \frac{1}{|B|^2} \sum_{i \neq j} a_i a_j P(i \in B, j \in B) - \bar{a}^2$$

$$= \frac{q}{|B|^2} \sum_i a_i^2 + \frac{1}{|B|^2} \sum_{i \neq j} a_i a_j \frac{\binom{|D|-2}{|B|-2}}{\binom{|D|}{|B|}} - \bar{a}^2$$

$$= \left( \frac{1}{|B|} - \frac{1}{|D|} \right) \left( \frac{1}{|D|} \sum_i a_i^2 - \frac{1}{|D|(|D|-1)} \sum_{i \neq j} a_i a_j \right)$$

$$= \frac{|D|}{|D|-1} \left( \text{Var}[Z_P] - \left( \frac{1}{|B|} - \frac{1}{|D|} \right) \bar{a}^2 \right).$$

Therefore, defining $\overline{a^2} = \frac{1}{|D|} \sum_{i=1}^{|D|} a_i^2$, when $|B| < |D|$ and $\overline{a^2} \neq 0$ we obtain the following variance relations

$$\frac{Var[Z_R]}{Var[Z_P]} = \frac{1}{1-q} \left( 1 - \frac{\bar{a}^2}{\overline{a^2}} \right), \tag{142}$$

$$\frac{\text{Var}[Z_B]}{\text{Var}[Z_P]} = \frac{1}{1-1/|D|} \left( 1 - \frac{\bar{a}^2}{\overline{a^2}} \right), \tag{143}$$

and

$$\text{Var}[Z_R] = \frac{1-1/|D|}{1-q} \text{Var}[Z_B]. \tag{144}$$

In particular, we see that fixed-size subsampling with replacement always has larger variance than fixed-size subsampling without replacement by the constant multiple $\frac{1-1/|D|}{1-q} > 1$; however, this factor is close to 1 when $q$ is small. Therefore, for practical purposes we can consider them equivalent from the perspective of the variance. When $|B| < |D|$ the ratio of fixed-size and Poisson variances is

$$\frac{\text{Var}[Z_B]}{\text{Var}[Z_P]} = \frac{1}{1-1/|D|} \left( 1 - \frac{\bar{a}^2}{\overline{a^2}} \right). \tag{145}$$

The Cauchy-Schwarz inequality implies $0 \leq \bar{a}^2/\overline{a^2} \leq 1$ and so we see that $\text{Var}[Z_B] < \text{Var}[Z_P]$ when $\bar{a} \neq 0$ and $|D|$ is sufficiently large; in DP-SGD the condition $\bar{a} \neq 0$ corresponds to being away from a (local) minimizer or critical point of the loss. This calculation suggests that fixed-sized minibatches are better to use when one is away from the optimizer, as they lead to a (potentially substantial) reduction in variance, but they can become (slightly) worse than Poisson-sampled minibatches when one is close to a local minimizer (i.e., $\bar{a} \approx 0$). However, for a typical training-set size $|D| \gg 1$ we have $(1 - 1/|D|)^{-1} \approx 1$ and so the variance advantage of Poisson subsampling when $\bar{a} \approx 0$ is very minor. In addition, due to the effect of noise, DP-SGD cannot reach $\bar{a} = 0$. Therefore, from the perspective of the variance, we conclude that fixed-size subsampling (with or without replacement) is generally preferable over Poisson subsampling.

# G Comparison of Poisson and Fixed-size DP Bounds under Add/Remove Adjacency

Using the conversion from Proposition 3 in Mironov (2017), if a mechanism satisfies $\epsilon'(\alpha)$-RDP then it satisfies $(\epsilon, \delta)$-DP with

$$\epsilon = \epsilon'(\alpha) + \frac{\log(1/\delta)}{\alpha - 1}. \tag{146}$$

Under add/remove adjacency, to leading order, both Poisson and fixed-size subsampling satisfy RDP bounds with $\epsilon'(\alpha) = \alpha R$ for some $R > 0$ (see Abadi et al. (2016); Mironov et al. (2019)) and (19). Assuming that the optimal $\alpha$ is not close to one, and hence we can approximate $\alpha$ with $\lambda := \alpha - 1$, minimizing over $\alpha$ gives

$$\epsilon = \inf_{\alpha > 1} \left\{ \epsilon'(\alpha) + \frac{\log(1/\delta)}{\alpha - 1} \right\} \approx \inf_{\lambda > 0} \left\{ \lambda R + \frac{\log(1/\delta)}{\lambda} \right\} = 2\sqrt{\log(1/\delta)R}. \tag{147}$$

For $t$ steps of fixed-size subsampling (with or without replacement), and with constant variance $\sigma$, Eq. (19) implies

$$R_{FS} \approx \frac{tq^2}{2}(e^{4/\sigma^2} - 1) \approx \frac{2tq^2}{\sigma^2}. \tag{148}$$

In contrast, for Poisson subsampling the results in Abadi et al. (2016); Mironov et al. (2019) imply

$$R_P \approx \frac{tq^2}{2}(e^{1/\sigma^2} - 1) \approx \frac{tq^2}{2\sigma^2}. \tag{149}$$

Therefore $R_{FS} \approx 4R_P$ at leading order and so (147) implies

$$\epsilon_{FS} \approx 2\sqrt{2t\log(1/\delta)}\frac{q}{\sigma} \approx 2\epsilon_P. \tag{150}$$

From a privacy perspective, this suggests that $\epsilon$ for Poisson subsampling has an intrinsic advantage over fixed-size subsampling by approximately a factor of 2 when using the add/remove adjacency relation. Noting the scaling of (150) with $\sigma$, we see that fixed-size subsampling requires noise $2\sigma$ to obtain the same privacy guarantee (at leading order) as Poisson sampling with noise $\sigma$. Tighter translation between RDP and $(\epsilon, \delta)$-DP, such as Theorem 21 in Balle et al. (2020) which is used by Meta Platforms (2024), alters this story somewhat but the general picture we have outlined here is corroborated by the empirical comparison provided in Section 4.2. We emphasize that this factor of two gap disappears when using replace-one adjacency; in that case Poisson subsampling and FS$_{woR}$ subsampling lead to the same DP guarantees, as discussed in Section 4.1.

## H  Additional Experiments

To assess the sensitivity of our proposed method to important parameters such as $\sigma$ and batch size $|B|$, we conducted a series of additional experiments to compare our proposed fixed-size method with the non-fixed size setting while varying values of $B \in \{50, 100, 150, 200\}$ and $\sigma \in \{3.0, 4.5, 6.0, 12.0\}$. Figure 10 depicts the results of these experiments.

As shown in Figure 10, the results for varying values of $\sigma$ and $|B|$ align with what we reported in Section 4.2, showing that the proposed method is not sensitive to parameter settings.

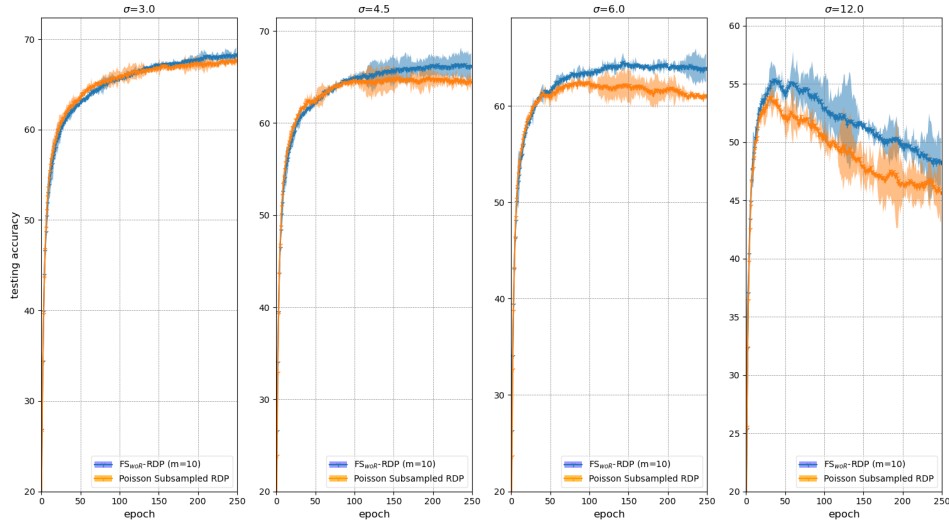

a Performance with different values of $\sigma \in \{3.0, 4.5, 6.0, 12.0\}$

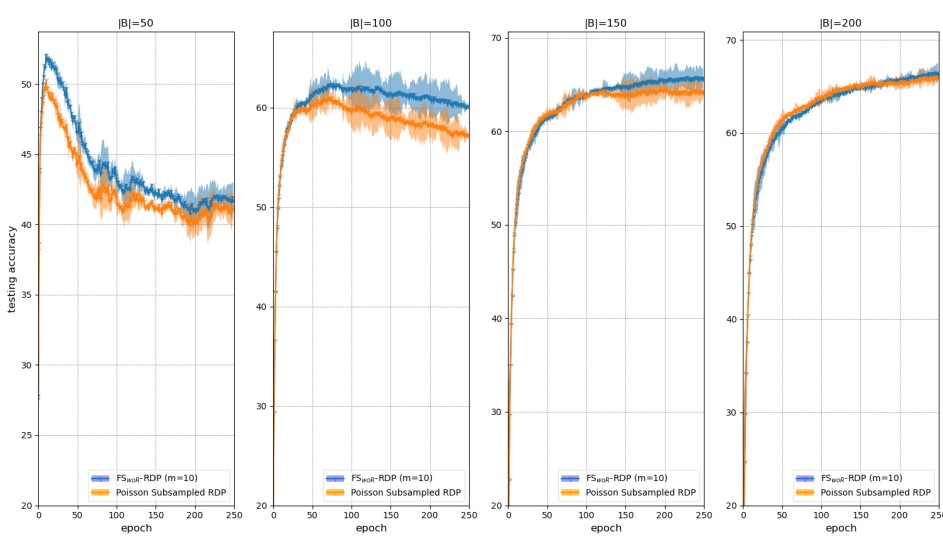

b Performance with different values of batch size $|B| \in \{50, 100, 150, 200\}$

Figure 10: Comparing Fixed and Non-Fixed Size Performance with different $\sigma$ (a) and batch sizes (b) on CIFAR10

