# OpenReview forum: "Differentially Private Stochastic Gradient Descent with Fixed-Size Minibatches: Tighter RDP Guarantees with or without Replacement"
_NeurIPS.cc/2024/Conference — NeurIPS 2024 poster_

### Official Review · Reviewer_Rv66 · 2024-06-18

**Soundness:** 3
**Presentation:** 3
**Contribution:** 1
**Rating:** 4
**Confidence:** 4

**Summary:**

The paper studies the Rényi differential privacy (RDP) guarantees of the subsamplied Gaussian mechanism with fixed-sized random minibatches, when the so called add/remove neighborhood relation of datasets is considered. It uses similar techniqes as were used in the paper

Mironov, Ilya, Kunal Talwar, and Li Zhang. "R\'enyi differential privacy of the sampled gaussian mechanism." arXiv preprint arXiv:1908.10530 (2019),

which augmented the seminar paper Deep Learning with DP (Abadi et al., 2016) and gave rigorous RDP guarantees for the Poisson subsampled DP-SGD.

The paper also studies both analytically and numerically the benefits of carrying out subsampling with fixed-sized minibatches instead of Poisson subsampling.

**Strengths:**

- The analysis looks solid, generally a well-written paper
- The variance analysis and the numerical experiments showing the benefits of fixed-size subsampling without replacement seem novel and interesting
- The analysis for the "with replacement" subsampling seems interesting though it is quite limited

**Weaknesses:**

- Clearly the biggest deficit of the paper is that it does not take into account some of the recent research in this area. Most importantly, it overlooks the work

Zhu, Yuqing, Jinshuo Dong, and Yu-Xiang Wang. "Optimal accounting of differential privacy via characteristic function." International Conference on Artificial Intelligence and Statistics. PMLR, 2022. https://proceedings.mlr.press/v151/zhu22c/zhu22c.pdf

The main result of this paper is a special case of the Thm. 11 by Zhu et al. (2022) : If $(P,Q)$ are a dominating pair of distributions for the base mechanism in substitute relation, then the fixed-size random subsampling without replacement give a dominating pair $(q P + (1-q) Q, Q)$ for removal neighbours and $(P, (1-q) P + q Q)$ gives a dominating pair for add neighbors. Then, we know the dominating pair $(P,Q)$ of the Gaussian mechanism in case of substiture relation of datasets (pair of one-dimensional Gaussians) and the result of this paper's mmain result (Thm. 3.1) follows since if a pair if dominating pair for the hockey-stick divergence, by the Blackwell theorem it dominates also for other $f$-divergences.

Since this "without replacement" result consitutes such a big part of the paper, I think this is a major deficit and there should be a major revision before accepting this paper.

This with replacement upper bound seems interesting, however my impression is that it is quite conservative (see e.g. Fig. 7 of the appendix). Only the experimental results for the lower bound in case of "with replacement" are given in the main text.

**Questions:**

Can you comment on the accuracy of the upper bound in case of "with replacement"? Is it still open/unclear, how tight those bounds are?

Could you provide "with replacement" upper bounds also for the hockey-stick divergence, i.e., do you obtain a dominating pair of distributions for the hockey-stick divergence?

**Limitations:**

Yes.

---

> ### Author Rebuttal · Authors · 2024-08-07
>
> **Relation to Zhu et al. (2022)**
> Please see the response to this comment in the general rebuttal section above. As the analysis of Zhu et al. does not investigate application to DP-SGD, the results of Theorem 11 in the DP-SGD setting are hypothetical until or unless directly demonstrated.  The reviewer provides a proof sketch that does not result in a rigorous reduction of Thm. 11 (Zhu et al.) to our Thm. 3.1. We note that DP-SGD operations in practice are much more complicated than simple Gaussian Mechanism.
>
>
> **Accuracy of the upper bound "with replacement"**
> Our results, which include both a lower and upper bound, show that sampling with replacement catastrophically increases in the Renyi bounds. This aligns with intuition as the fact that one (i.e., an adversary) can get multiple same samples in the same batch is a drastic violation of privacy, and this is what we show in our Figure 7.
>
> **With replacement upper bounds for hockey-stick divergence**
> Bounding HS divergence with replacement is a nice research question, but is not trivial and requires substantial work that warrants further study. Our study focuses on Renyi divergence.

---

> > ### Comment · Reviewer_Rv66 · 2024-08-08
> >
> > Thank you for the replies! Thm. 11 by Zhu et al. 11 is independent of any accountant used to compute the privacy profiles. As I have pointed out in my review, from that theorem follows the Blackwell dominance as well and an existence of a post-processing function and furthermore the RDP bound you state in your paper. This unfortunately reduces the relevance of that particular result I think, and therefore I plan to keep my score.

---

> > > ### Author Response · Authors · 2024-08-08
> > >
> > > Thanks for further clarification. We investigated the extension of Zhu et al.'s Thm 11 using Blackwell Thm as you suggested to validate if one can obtain our bounds in that manner.
> > >
> > > In the case of replace-one adjacency relations, we do not believe that the general Blackwell theorem ideas along with the results in Zhu et al are sufficient to reproduce our results. Theorem 11(b) in Zhu et al applies to add or remove adjacency for the subsampled mechanism. Proposition 30 of Zhu et al does address replace-one adjacency, but it does not derive a full dominating pair, as it must break the analysis into two cases based on alpha>=1 and alpha<1. Corollary 32 in Zhu et al does give general theory for constructing a dominating pair, but it is not explicit or readily implementable in practice due to the need to compute a Legendre of transform (Fenchel conjugate) as a function of $x$ for `every' $x\in[0,1]$.  Therefore to the best of our knowledge, the result in Zhu et al. cannot be directly used to reproduce our new RDP bound for replace-one adjacency as found in our equation 110.

---

> ### Comment · Reviewer_Rv66 · 2024-08-09
>
> Thanks for the answers. But why would you need a result for replace-one adjacency? You define the neighborhood relation to be the add/remove neighborhood relation. On lines 80-81:
>
> > We define datasets $D$ and $D′$ to be adjacent if one can be obtained from the other by adding or removing a single element.
>
> Also, the proof of Thm 3.1 clearly shows that $D$ and $D'$ are of different sizes.
>
> Then, I think [Zhu et al., Thm. 11b](https://proceedings.mlr.press/v151/zhu22c/zhu22c.pdf) will give your result. On the right hand side of the bound of your Thm 3.1, you exactly have the dominating pair for the Gaussian mechanism under the replacement relation, just as required by [Zhu et al., Thm. 11b](https://proceedings.mlr.press/v151/zhu22c/zhu22c.pdf). Then you can use the results of Section 3 of [Mironov et al.](https://arxiv.org/pdf/1908.10530) to show that one of the Rényi divergences dominates the other which exactly gives your Thm. 3.1 (you also seem to use those results by Mironov et al.).

---

> > ### Author Response · Authors · 2024-08-09
> >
> > Thanks for asking for clarification on this. We agree with you that the steps you outline could provide an alternative path to the add/remove result in our Theorem 3.1. What we were highlighting in our last response is that our method of proof leads to a unified method for treating both add/remove and replace-one. The replace-one result is also of significant application value as was pointed out by JRin. Originally, we had the add/remove result in Theorem 3.1 in the main text while discussing replace-one in Appendix D.4. Based on your and JRin's comment, we have changed the main text to highlight the replace-one as well. We have also pointed out that the add/remove result could alternatively be obtained via the method you outline.

---

> > > ### Comment · Reviewer_Rv66 · 2024-08-12
> > >
> > > Thank you for the replies. In fact, I think the replace one result could also be derived directly from the results by [Zhu et al.](https://proceedings.mlr.press/v151/zhu22c/zhu22c.pdf). If you use their proposition 30, and combine it with the results of Section 3 of [Mironov et al.](https://arxiv.org/pdf/1908.10530) (i.e., onlly on of the cases needs to be considered), you get the RDP bound for that case as well.

---

> > > > ### Author Response · Authors · 2024-08-12
> > > >
> > > > Thanks for the answer. It is not immediately clear to us that the calculation you propose can be carried out explicitly in that case, but either way it requires substantial analysis/computation which is not done in Zhu et al. As we noted, the result in Zhu et al, as presented, is not amenable to implementation in practice, while our results are. Specifically, a significant aspect of our proposed method is the use of Taylor series expansions to capture the leading order behavior (when $q$ is small) while providing explicit computable bounds on the error terms to prevent privacy leakage. This aspect of our work is not considered in Zhu et al or Mironov et al.

---

> ### Comment · Reviewer_Rv66 · 2024-08-13
>
> Thank you for the reply. I confused your Section C.2 with your Section D.4. There, in Section D.4, it is diffiult for me to see why that RDP bounds should hold. Namely, you have a bound with a Rényi divergence between the mixtures $q \cdot \mathcal{N}( \Delta_1, \sigma^2) + (1-q) \cdot \mathcal{N}( 0, \sigma^2)$ and $q \cdot \mathcal{N}( \Delta_2, \sigma^2) + (1-q) \cdot \mathcal{N}( 0, \sigma^2)$. The derivation of this bound is omitted. Recently, [Lebeda et al.](https://arxiv.org/pdf/2405.20769) showed that the bound for the subsampling without replacement under the substiture relation does not at least have a hockey-stick upper bound with mixtures $q \cdot \mathcal{N}( -1, \sigma^2) + (1-q) \cdot \mathcal{N}( 0, \sigma^2)$ and $q \cdot \mathcal{N}( 1, \sigma^2) + (1-q) \cdot \mathcal{N}( 0, \sigma^2)$, one has to use bounds such as Prop. 30 by [Zhu et al.](https://proceedings.mlr.press/v151/zhu22c/zhu22c.pdf). Thus, I am a bit suprised to see this result given in Section D.4, and I think a detailed derivation would be needed. Also, I don't see where the bound of Eq. (108) comes from, why is there no additional factor 2 (or 4 after squaring).
>
> I do acknowledge the contribution in making the RDP bound calculations faster, but I think the paper would benefit some polishing and by piutting the results in a context.

---

> > ### Author Response · Authors · 2024-08-13
> >
> > Thanks for the question. We want to clarify the notion of "worst-case" in this context that might have created some confusion: Our method gives "an" upper bound vs. the "tightest" bound.  Based on your comments and also those of uSpK, we have added additional details to Appendix D to make our derivation clearer.  We hope that the following outline clarifies it:
> >
> > To show that the Gaussian mixtures provide an upper bound on the worst case, we use the probabilistic Lemma A1 that decomposes the transition probabilities of mechanism.  In the replace-one case the role of $D$ and $D'$ are symmetric and so they both satisfy a decomposition of the form (31).  Then using convexity, the Renyi divergence between Eq.31(with $D$) and Eq.31(with $D'$) is bounded above by maximum over $b^\prime,\widetilde{b}$ of the Renyi divergence between Gaussian mixtures, as in Eq.102.  The only property of the divergence used in these steps is quasiconvexity, and so they apply to the hockey-stick divergence as well, though our subsequent Taylor expansion calculations that lead to explicit computable bounds are specific to Renyi. We emphasize that we only compute an upper bound and not the tightest upper bound, so there is no conflict with Lebeda et al, though we still obtain tighter bounds than the previous state-of-the-art explicit computable bound in Wang et al. The examples in Lebeda et al. seem to highlight the difficulty of obtaining the tightest possible bound on the hockey-stick divergence, which is an interesting but different theoretical problem than what we achieved in this work.
> >
> > As to your other question, the factor of 4 is contained in our definition of $r_t$; see eq. (34).

---

### Official Review · Reviewer_JRin · 2024-07-06

**Soundness:** 4
**Presentation:** 4
**Contribution:** 4
**Rating:** 7
**Confidence:** 4

**Summary:**

This paper analyzes Differentially Private-SGD with fixed batch size (with and without replacement), through the lens of Rényi-DP. The bounds without replacement have a much better constant than previous ones; the ones with replacement are brand new.

**Strengths:**

- The results are important and likely to be impactful in the niche of DP deep learning. It’s cool to see progress on DP analysis that gets closer to what practitioners actually do.
- The paper is really well written and thorough in its discussions. I really enjoyed reading it.

**Weaknesses:**

It’s a bit of a nitpick, but the paper keeps claiming that this analysis is DP-SGD specific, and that’s what enables the tight bounds. It seems like it is specific to:
- the Gaussian mechanism (with specific subsampling approaches)
- and I guess additive contributions of datapoints to the function?
Anyway, it might be good to give the specifics in a slightly more general/abstract fashion, and then discuss why it applies to well to SGD? Just so that the reader doesn’t expect tight couplings with the optimization or something, and also because it may be slightly more broadly useful.

Minor:
- several citations seem off: the “standard differential privacy (DP)” citation is CDP, poisson sampling from RDP cites the RDP paper instead of this one I assume https://arxiv.org/pdf/1908.10530.
- "even without using the convexity technique” p. 6 -> this has never been mentioned before. You need more details here for context.

**Questions:**

Since you reduce to the same intermediary quantity as https://arxiv.org/pdf/1908.10530 (in Eq. 6), does it mean that:
- the bound of that paper applies as-is to fixed batches using your proof (i.e., if one were using the Opacus accountant with fixed batches as a heuristic, it’s actually not a heuristic)?
- if not, why is that?
- if so, how does your new bound compares? (other than enabling the analysis with replacement, which is cool!)
I think that the paper would benefit from discussing those to give a bit more context. S4.2 is related, as it gives an interesting discussion of where the different factor comes from (so the quantity is not identical I guess), but not all of this, like whether you could use the previous approximation as is, and how the two numerical procedures compare.

After reading S4.2, I am wondering if the difference is a bit artificial, or at least tied to the notion of sensitivity. For instance, Poisson sampling would also pay a factor two under the change one definition (I think?) whereas your approach would mostly not change I believe (I only skimmed the appendix on sensitivity to I’m not completely sure, but intuitively it’d make sense—the section doesn’t seem to discuss the implications on Poisson vs. fixed batch). Is it the case that your approach pays a 2x factor, but also gets the stronger change-one for free, whereas Poisson gets the better guarantee under add/remove, but its equivalent under change-one? It would be an interesting thing if that’s the case I think (change-one is pretty popular in other applications, I don’t think it’s really less conventional as the paper claims).

---

> ### Author Rebuttal · Authors · 2024-08-07
>
> **DP-SGD Specific**
> That is an accurate comment, in that we do not use the fact that the additive terms involve gradients. Our method is applicable to any Gausian Mechanism with fixed-size subsampling where each sample contributes an additive term to the mean and those terms are uniformly bounded. The only aspect of DP-SGD that we do not explicitly use is the fact that each additive term is a clipped gradients. These specific structural aspects of DP-SGD are what allow us to obtain tigher bounds than Wang et al. (2019).
>
>
> **Relationship to Mironov et al. 2019**
> Thanks for pointing this out. The answer is no. The reason is that if one uses Opacus implementation with Fixed Size sampling mechanism that would lead to incorrect privacy bounds.  Our previous wording gave the false impression that the Mironov et al.'s method applies to the cases under consideration here and so we have removed that sentence.  The relation between Mironov et al. and our work is discussed in detail in Section 4.2.
>
>
> **Section 4.2: Adjacency Relations**
> Thank you for this insightful question. You are correct, the 2x factor is the same whether one considers add/remove or replace adjacency relations, as we show in Appendix D4. Therefore our approach gets change-one for free as you say, while Poisson achieves tighter bounds under add/remove but gives equivalent bounds to leading order under replace-one adjacency, as shown by a similar computation to what is currently in Appendix D4.  We have added text in the introduction as well as in Sections 4.2 and Appendix D4 to further highlight this fact.

---

> > ### Comment · Reviewer_JRin · 2024-08-09
> >
> > Thank you. My impression looking at how the theory works in your paper was that it did reduce to the same accounting at Mironov et al. 2019 (Since you reduce to the same intermediary quantity as https://arxiv.org/pdf/1908.10530 (in Eq. 6)). Could you give an intuition / place of divergence in your proof for why that is not the case? Thanks!

---

> > > ### Author Response · Authors · 2024-08-09
> > >
> > > Thanks for asking for clarification on this. We get to a similar, but not identical expression, to that of Mironov et al. The difference is in the right-hand side of our eq.6, which differs from Mironov by the factor of 1/4 in the variances. This is due to the effects of fixed-size vs. Poisson subsampling. The key difference in the derivation in our Appendix A is contained in our Lemma A.1, as compered to Theorem 4 in Mironov et al. There we construct the random variables that lead to the appropriate decomposition of the mechanism. In particular, compare the construction of our $J$, $B'$, and $\widetilde{B}$ in the paragraph about Lemma A.1 with the random variable T defined at the start of the proof of Theorem 4 in Mironov, along with (implicitly) the random variable that selects whether the additional element x is included, leading to the mixture at the bottom of page 3 of Mironov. The construction of our $J$, $B'$, and $\widetilde{B}$ are the analogues that lead to the appropriate decomposition of the mechanism in the case of fixed-size subsampling, as proven in our Lemma A.1. This difference then propagates through the derivation, eventually leading to the difference by a factor of 1/4 in the variance as noted above. The key reason for this is that our eq 34-35 shows that the mechanism in the case of fixed-size subsampling has a sensitivity of twice that in the Poisson case. This is because in Poisson subsampling when the minibatch from $D$ differs from that of $D^\prime$ it is due to the inclusion of a single additional element. However, in fixed-size subsampling, when the minibatches  are not identical then they differ by a replacement; this  contributes more to the difference in means  by a factor of $2$. Furthermore, in our analysis of fixed-size subsampling with replace-one adjacency in Appendix D.4 there are even greater differences between our analysis and that of Mironov et al.
> > >
> > > We added the text to highlight this difference with Mironov et al. and directed readers to our Appendix A and Lemma A.1 in this regard. Also, in our Appendix A, we added the intuition described above.

---

### Official Review · Reviewer_kPGW · 2024-07-08

**Soundness:** 3
**Presentation:** 2
**Contribution:** 3
**Rating:** 5
**Confidence:** 3

**Summary:**

The paper first proves new privacy bound for the subsampled Gaussian mechanism under fixed-size sampling with and without replacement, which improves over the tightest known prior results in (Wang et al. 2019) by a constant of four. The proofs rely on a careful coupling of the sampling processes on neighboring datasets (similar to the design in Wang et al. 2019) to reduce the problem of analyzing divergence between Gaussian mixtures, for which the analyses in (Mironov et al. 2019) are applicable.

- For fixed-size sampling without replacement, the paper proves integral upper bounds for the remainder term of the Taylor expansion of the privacy bound, to obtain tighter constants compared to the privacy bound in Wang et al. 2019.
- The method extends to fixed-size sampling with replacement. It allows an upper bound that is similar to the upper bound for sampling without replacement in the leading term to sampling probability q. The authors also prove a lower bound under such settings and numerically investigate its dependence on the batch size and the Renyi divergence order.
- The authors analytically showed that the empirical gradient variance is larger under Poisson sampling compared to fixed-size sampling, but the privacy bound is smaller under Poisson sampling compared to fixed-size sampling, indicating an interesting privacy-utility trade-off depending on the choice of sampling schemes, when fixing the sampling probability the same.

**Strengths:**

- An interesting way of computing DP guarantee for subsampled Gaussian mechanism, via computing integral upper bounds for the remainder term of its Taylor expansion.
- The proposed method yields a tighter RDP guarantee for subsampled Gaussian mechanism under fixed-size sampling without replacement, compared to prior results (Wang et al. 2019) by a constant factor of four.
- The method extends to fixed-size sampling with replacement. It allows an upper bound that is similar to the upper bound for sampling without replacement in the leading term to sampling probability q. The authors also prove a lower bound under such settings and numerically investigate its dependence on the batch size and the Renyi divergence order.

**Weaknesses:**

- The reason for the improved constant factor compared to (Wang et al.) is not crystal clear. Is the integral upper bound for the remainder term of Taylor expansion of the DP bound contributing to the tighter constant? Unfortunately, the Taylor expansion is not explained in much detail in the main paper (e.g., in Theorem 3.2, the crucial terms related to $A$ and $M$ are not presented nor explained).
- Although it is interesting that the paper shows tighter constant in leading order term of the DP bound (via analytical approach), the value of this contribution in practice needs more explanation. As the divergence between Gaussian mixtures could be tightly computed numerically following (Mironov 2019.), it is unclear why we need an tighter analytical bound given by integral upper bound for the remainder term of Taylor expansion of the DP bound.

**Questions:**

1. The reason for improved tightness of DP bound by a constant factor (see weakness for more details)
2. It is interesting that the paper shows that Poisson sampling enables smaller RDP bound than fixed-size sampling when keeping the sampling ratio the same. However, I wonder if it is a phenomenon that is unique to the add-or-remove-one notion of differential privacy. Could the authors comment on whether there will be a similar gap between Poisson sampling and fixed-size sampling when considering the replace-one notion of differential privacy?

---

> ### Author Rebuttal · Authors · 2024-08-07
>
> **"The reason for the improved constant factor compared to Wang et al. is not crystal clear"**
> Thanks for pointing this out. We clarified further at the end of the introduction (see also the discussion in Appendix D4) that we compute the Taylor expansion of the Renyi divergence in the sampling probability, q, with explicit upper bounds on the error terms. As q is small in practice, the error terms are small.  Therefore the fact that we exactly capture the leading-order behavior in q  leads us to have nearly optimal bounds in practice, as seen by our comparison between upper and lower bounds in Figure 2.  This is in contrast to the method of Wang et al, which does not capture the leading order behavior in q and therefore is not tight for DP-SGD.
>
> **"The value of this contribution in practice needs more explanation..."**
> The Taylor expansion is used because it allows the leading order behavior in q to be captured exactly, which provides tighter RDP  bounds.  We used explicit bounds on the error terms rather than the numerical approach of Mironov et al. because it requires computing  fewer terms.
>
> **"Could the authors comment on whether there will be a similar gap between Poisson sampling and fixed-size sampling when considering the replace-one notion of DP?"**
> Thank you for this insightful question. No, under replace-one regime, there would be no gap between the Poisson and fixed-size, to leading order.  We clarified this point in the comparison in section 4.2 as well as in Appendix D4.

---

> > ### Author Response · Authors · 2024-08-13
> >
> > We thank reviewer kPGW again for their time and useful insights.  As the discussion period comes to a close we ask that you consider our responses and whether your concerns have been adequately addressed.  We tried to pay particular attention to clarifying remarks about improvement of the constant factor and the value of the contribution.  Thanks again!

---

> > > ### Comment · Reviewer_kPGW · 2024-08-14
> > >
> > > Thanks for the clarifications. I will keep the score -- I now understand better about the efficiency value of the work (compared to numerical computation for Gaussian mixtures). However, the reason for constant improvement is still not clear to me analytically, and I encourage the authors to add more discussions of the Taylor expansion terms in the main paper.

---

### Official Review · Reviewer_uSpK · 2024-07-12

**Soundness:** 2
**Presentation:** 2
**Contribution:** 3
**Rating:** 4
**Confidence:** 3

**Summary:**

This paper studies the Renyi DP guarantees of a with- or without replacement subsampled Gaussian mechanism. Authors present a privacy analysis tailored for the subsampled Gaussian mechanism, which improves the earlier general bound for $\epsilon(\alpha)$ by Wang et al. 2019 by a factor of four. Authors show analytically, that the subsampling induced variance (i.e. the variance of the noise arising from using minibatches instead of full data) is smaller for without replacement subsampling than for the more commonly used Poisson subsampling. Authors also give a theoretical analysis of the differences between WOR and Poisson subsampling, showing that the Poisson subsampling leads to approximately half the $\epsilon$ of the WOR sampling. Authors demonstrate empirically that for fixed noise level, the WOR leads to better accuracy than Poisson subsampling in a CIFAR10 based deep learning task, suggesting that the difference is due to the lower subsampling noise. Finally, authors show that the WOR sampling leads to more stable memory usage.

**Strengths:**

The DP-SGD algorithm is by far the most widely applied tool for DP machine learning. Since WOR sampling is more commonly used in non-DP ML than Poisson subsampling, improving the privacy bounds for WOR sampling is a interesting and important contribution. The theoretical analysis based on Taylor expansion of the Renyi divergence is a novel contribution and allows stricter bounds than the general result by Wang et al. 2019. Furthermore, the fact that WOR results into smaller subsampling noise is an interesting finding as well.

The numerical results for the accounting highlight the significant improvement over the current state-of-the-art WOR sampling privacy accounting, showing over a factor of two improvement in the $\epsilon$ after conversion to approximate DP bounds. Also the empirical comparison on memory usage demonstrates the benefits of WOR sampling.

**Weaknesses:**

While the presented analysis provides important insights and improvements over the previous RDP analysis for WOR sampled Gaussian mechanism, I wonder if the problem is already solved with the modern privacy accounting tools. For example, Zhu et al. 2022 solve the WOR sampled Gaussian mechanism in their characteristic function formalism. Given that the conversion from RDP to approximate DP is lossy, I would imagine the bounds presented in this work are more loose than the bounds of Zhu et al when converted to approx DP domain. Since the approx DP bounds are more commonly used than RDP bounds, I'm not sure if the tight RDP analysis is really needed. Or is there some practical reason, e.g. an implementation difficulty, that would not allow using the characteristic function accounting for this problem?

Zhu, Yuqing, Jinshuo Dong, and Yu-Xiang Wang. "Optimal accounting of differential privacy via characteristic function." International Conference on Artificial Intelligence and Statistics. PMLR, 2022.

**Questions:**

- As you acknowledge towards the end of Section 3.5, Wang et al. 2019 use the substitute neighbourhood relation which differs from the add/remove used in this paper. Can you clarify, is this difference taken into account in Figure 3?
- Fig. 2: I'm a bit confused on the upper bounds show in this Figure. It seems that after some $\alpha$, your proposed upper bound exceeds the one from Wang et al. Does this suggest that the Wang et al. bound is tighter in some regime of $\alpha$?
-  In the Appendix D, you derive the RDP bounds for the substitute relationship using mixture of Gaussians with weight $1-q$ on Gaussian centered at $0$ and weight $q$ for the "adjacent" point. A recent work by Lebeda et al. 2024, suggest that this might not be the worst-case distribution for woR-sampling (see Section 7 in Lebeda et al.). While their analysis is tailored for approximate-DP and not RDP, I wonder if the same holds RDP.

**Typos and other minor things**
- Eq. 57, extra parenthesis in the expression for $M_{\sigma, 4}$
- I guess the NN abbreaviation is never explained. However, I don't think using it is necessary for the paper to begin with as you analysis applies to any learning task using DP-SGD.
- Fig. 2: the caption is overlapping with the axis label. Also, I think this figure is never referred to in text.
- "... even without using the convexity technique; ...": which convexity technique are you talking about?
- "$a_i = \nabla_\theta L(d_i) \cdot v$", what is the $v$ here? Also, since the gradient is multidimensional, are you talking about dimension-wise variance?
- "addtional"

Lebeda, Christian Janos, et al. "Avoiding Pitfalls for Privacy Accounting of Subsampled Mechanisms under Composition." arXiv preprint arXiv:2405.20769 (2024).

**Limitations:**

I think authors should still address whether other accounting tools can solve this problem more efficiently that their method in the approximate DP domain. Other than that I believe the limitations are well addressed.

---

> ### Author Rebuttal · Authors · 2024-08-07
>
> **Relation to Zhu et al. (2022)**
> Please see the response to this comment in the general rebuttal section above.
>
> **Lossy conversion to $(\epsilon, \delta)$ guarantees**
> We acknowledge (as the reviewer correctly states) that RDP guarantees are not lossless when converted to $(\epsilon, \delta)$ guarantees. However, for RDP accountants, regardless of the conversion that is used, with our method it is possible to significantly improve guarantees (by a factor of 4). To our knowledge this is the strongest improvement on RDP for DP-SGD. Given that RDP is one of the most widely used methods in DP libraries, using our method leads to substantial improvements in a myriad of real-world applications.
>
> **Substitute neighborhood relation of Wang et al.**
> Thanks for asking for clarification on this. We show in our Appendix D.4 that the behavior of our bounds is the same for both replace-one and add/remove adjacency notion to leading order. To be consistent across the paper, in all of our implementations, we used the add/remove adjacency notion in Figure 3 and other Figures. We clarified this in the main body of the manuscript.
>
> **Upper bounds in Fig. 2**
> The behavior of bounds for large $\alpha$ is addressed beginning on Line 196.  In practice, for any choice of $\delta$ one need only consider the bound in a range of $\alpha$'s near 1 to convert from RDP to $(\epsilon,\delta)$-DP  and so the behavior of our bounds for large $\alpha$ is irrelevant in practice.  This can be seen by Figure 3.  In addition, by combining our bounds with the convexity technique discussed in Appendix D.3 one can eliminate this large $\alpha$ issue; see the solid black line in Figure 9.
>
> **Worst-case bounds for woR-sampling**
> The worst case in Lebeda et al. 2024 holds for HS-divergence and not RDP. As these are different divergences, the worst case distributions aren't necessarily the same. For RDP, we prove in our Appendix D that the worst-case is what we presented.
>
> **Convexity technique**
> The convexity technique we refer to on p. 6 was introduced in Wang et al. (2019).  Please see Appendix D.3 for more details on application of the convexity technique to Theorem 3.1.

---

> > ### Author Response · Authors · 2024-08-13
> >
> > We thank reviewer uSpK again for their time and helpful feedback.  As the discussion period comes to a close we ask the reviewer to consider our responses and whether their concerns are adequately addressed.  In particular we have provided extensive discussion relating our work to the work of Zhu et al. in the general comments, as well as in response to reviewer Rv66.  Thanks again!

---

> > > ### Comment · Reviewer_uSpK · 2024-08-13
> > >
> > > Thanks for your response, and apologies for my very late response! I have now read the rebuttal, and authors have addressed many of my concerns.
> > >
> > > In your common response you state that "AFA is not specific to DP-SGD". This is of course true, since the particular accounting technique is not tailored for DP-SGD. However, my point was that the accounting of DP-SGD can be carried out with the results of Zhu et al. using the AFA (or any other accountant like the PRV accounting). However, I do acknowledge the possible numerical challenges these methods pose, whereas your method can give at least an upper bound for the privacy cost (when converted to approximate DP). Since the proposed bound is more tight than the one proposed by Wang et al. I believe the proposed method can provide useful insights on the privacy accounting. However, it would be very interesting to see, if the numerical accounting indeed fails for this privacy analysis, or how far it is from your bounds.
> > >
> > > One more question related to your response **Worst-case bounds for woR-sampling**. I'm not sure where in Appendix D you actually show that the worst-case is what is presented. It seems to me that you just plug in the Gaussian mixtures as the upper bound after eq. 102.

---

> > > > ### Author Response · Authors · 2024-08-13
> > > >
> > > > Thanks for the question. For FS-woR under add/remove adjacency the worst case bound is stated in eq. (6) in Theorem 3.1.
> > > > For FS-woR under replace-one the worst case bound is stated in eq (102) in appendix D4.  Both are derived by the method in Appendix A, wherein the mechanisms are decomposed into mixtures via Lemma A1: see eq (31)-(32).  Then  quasiconvexity is used to bound the mixture over $b',\widetilde{b}$, leaving only the mixture over q, which is the probability that the last sample (the one that can differ) is included.  The essential difference that the adjacency relation makes is as follows:
> > > >
> > > > - In the add/remove case the first argument of Renyi divergence is decomposed via the mixture (31) while the second via (32) (as D' has one less element).  After using quasiconvexity to bound the expectation over $b',\widetilde{b}$ by the maximum one obtains (6). where the Gaussian mixture is only in the first argument of the Renyi divergence.
> > > >
> > > > - In the replace-one case, D and D' have the same number of elements and so both arguments of the Renyi divergence are decomposed as in (31) but with different means (coming from D and D'). After using quasiconvexity, this leaves a Gaussian mixture in both arguments of the Renyi divergence.
> > > > In both cases we then go on to obtain computable worst-case bounds using Taylor expansion with remainder.
> > > >
> > > > We have added additional details to Appendix D to make this derivation clearer. Also, we agree with you that investigating the practical boundaries of numerical methods in this context is a fruitful but separate research endeavor.

---

### Author Rebuttal · Authors · 2024-08-07

We thank all reviewers for their time and very useful feedback!  We provide responses to each reviewer individually.  Please see below for responses to a common point.

**Relation to Zhu et al. (2022)**

Reviewers uSpK and Rv66 ask whether our WOR result of Theorem 3.1 is a corollary of Theorem 11 in Zhu et al. (2022).  Their work leads to the analytical Fourier accountant (AFA) for Gaussian mechanisms in general. AFA is not specific to DP-SGD, which is the main topic of our work. Our work should be viewed as an extension of Wang et al. (2019) that is specific to DP-SGD with Gaussian noise and focuses on the strong composition afforded by RDP. Theorem 11 of Zhu et al. (2022) accounts for addition and removal of neighbors in the general Gaussian mechanism case. We could not find any evidence that the analysis in Thm 11 yields a privacy bound for DP-SGD with a large number of training steps that is tighter than and as rigorous as what we have provided. Our reasons are as follows:
DP-SGD requires the consideration of Gaussian mixtures, and therefore the characteristic function required by the method of Zhu et al must be  computed numerically. We anticipate this can be difficult for a large number of compoisitions, as is typical in DP-SGD, due to the required integrations of a sum of a large number of terms, as required by Algorithm 1 in Zhu et al. In addition, there is potential privacy leakage due to lack of rigorous error bounds for the double (Gaussian) quadrature numerical integration approach as discussed in their Appendix E.1. Using Gaussian quadrature for DP-SGD presents major practical difficulties that do not seem to be addressed in Zhu et al's work. Our approach is a specialized method for DP-SGD that avoids the practical concerns associated with numerical integration.

Hayes et al. (2024) uses the results in (Zhu et al., 2022) to provide a bound specific to DP-SGD when one is only concerned with training data reconstruction attacks rather than the membership inference attacks, which leads to a relaxation of DP to provide a bound that is only applicable to defending against data reconstruction and not membership inference. Our work can be viewed as a complimentary work to Hayes et al. (2024) that is specific to DP-SGD and provides conservative bounds protecting against membership inference attacks (and thus any other type of model inversion attacks including training data reconstruction) in the RDP context. Overall, the applicability of Thm 11 in Zhu et al. (2022) to fixed-size subsampled DP-SGD is hypothetical until or unless they are explicitly demonstrated in the DP-SGD context.

We discussed both of these studies in our background and related work.

    * Zhu, Y., Dong, J. and Wang, Y.X., 2022, Optimal accounting of differential privacy via characteristic function. In International Conference on Artificial Intelligence and Statistics (pp. 4782-4817). PMLR.
    * Hayes, J., Balle, B. and Mahloujifar, S., 2024. Bounding training data reconstruction in dp-sgd. Advances in Neural Information Processing Systems, 36.

---

### Decision · Program_Chairs · 2024-09-25

**Decision:**

Accept (poster)

**Comment:**

The paper studies the privacy accounting of DP-SGD under the sampling-without-replacement model but unlike in previous work that worked with a "replace" neighbor relationship, this paper uses an "add/remove" neighbor relationship.

The main claim is that for DP-SGD, they can improve the best known Renyi-DP bound by a factor of 4 when the order $\alpha$ is smaller than a threshold and that (at least numerically) the restricted range $\alpha$ still supports stronger composition in DP-SGD.

Reviewers acknowledged the importance of the problem and the claimed improvement but found the following aspects unsatisfying.

1. the use of "add/remove" relationship in amplification by WOR sampling was studied in a prior work [ZDW'22], with a direct analysis of $(\epsilon,\delta)$-DP by analyzing the hockey-stick divergence of a dominating pair.

2. One reviewer is skeptical of the claim that the particular Gaussian mixture being a dominating pair for the WOR sampling case in the Renyi ordering, as it was shown in the recent work of Lebeda et al that it isn't in the "Blackwell ordering".  The same reviewer also comments on other issues in Appendix D.4. such as:

> "I don't see where the bound of Eq. (108) comes from, why is there no additional factor 2 (or 4 after squaring)."

3.  The techniques used are similar to those in Mironov et al. 2019, and Wang et al. 2019.

I only have a limited amount to time to look into these issues. For Point 1,  I do believe that a tighter RDP analysis has its own value even if it does not give a tight epsilon-delta DP bound as the Hockey-Stick Divergence.

- If the authors add a comparison of the RDP-converted approx-DP privacy profile to the latest FFT-based numerical account as in Lebeda et al., it should be sufficient in addressing this concern.

Point 3 was acknowledged in the paper. The particular coupling used is similar to Wang et al. 2019 and the Taylor expansion of the mixture of Gaussian into quantities one can evaluate with Gaussian CDF calls is due to Mironov et al. 2019.  The authors however clearly discussed the differences in the proof.

In the discussion period, the reviewers and the AC tried to resolve Point 2, but it requires more time than we had. However, in general, dominance in Blackwell ordering and dominance in Renyi ordering are different. It is perfectly possible that the mixture of gaussian pair does not dominate in Blackwell ordering (as was shown in prior work) but it dominates in Renyi ordering.  We did not catch any obvious error in the proof of Theorem 3.1, so it should check out!

I wanted to add that the authors also proved new results about the "replace" neighbor in appendix D.4, which the authors wrote in the rebuttal that they should have focused a bit more on it.  I agree that "replace" neighbor is more natural for random subset sampling.   The paper ran into the same issue of having mixture of two Gaussians in both the numerator and denominator as in Wang et al. 2019 when applying joint convexity directly.   Wang et al., 2019 went through a ternary-type divergence and then the binomial expansion, which resulted in a general result with more precise control of the higher order terms, but incurred a factor of 4 in the leading term.  The current paper directly applied Taylor expansion twice (first over q near q = 0, then for e^x near x = 0) in a way similar to the approach Abadi et al (2016) took. This argument ended up resulting in an expression that allows one to complete the square in Eqn (107), hence reducing the factor of 2 ( factor of 4 after squaring) in the leading term.   The reviewer's confusion of how (108) avoids the factor of 2 stems from that the sensitivity is now defined to be 2x the clipping threshold. This got clarified in the AC-reviewer discussion.

I think this is a nice step forward on the problem introduced 5 years ago by Wang et al 2019.  For this reason, I think the paper deserves to be published.

------------
Other comments the authors could consider:

1.  For the Add/Remove case, q is different for the two neighbor datasets.   |B| / n vs |B| / (n-1).  For the definition to make sense, one needs to either make n known or define the space of the dataset to be n > certain public threshold, then state the results according to that pubic threshold.  This was discussed in [ZDW'22].

2. Because of the above, it is more natural to use the Replace neighbors for WOR, and the result hidden in Appendix D.4. could be emphasized and moved to the main paper?

3. Results in Appendix D.4 are worth developing a bit further so that the higher-order terms in the Taylor expansions are replaced with valid upper bounds.  I also suspect the bound to be weaker than that of Wang et al 2019 when we are outside the regime of \sigma being large and q being very small.

4. As I mentioned before, numerically adding comparison to the FFT-based numerical accountants that skips RDP would make it clear the practical impact of the result here.